# Sail into the Headwind: Alignment via Robust Rewards and Dynamic Labels against Reward Hacking

**Paria Rashidinejad, Yuandong Tian**
Fundamental AI Research (FAIR) @ Meta AI
`{pariard,yuandong}@meta.com`

## Abstract

Aligning AI systems with human preferences typically suffers from the infamous *reward hacking* problem, where optimization of an imperfect reward model leads to undesired behaviors. In this paper, we investigate reward hacking in offline preference optimization, which aims to improve an initial model using a preference dataset. We identify two types of reward hacking stemming from statistical fluctuations in the dataset: Type I Reward Hacking due to subpar choices appearing more favorable, and Type II Reward Hacking due to decent choices appearing less favorable. We prove that many (mainstream or theoretical) preference optimization methods suffer from both types of reward hacking. To mitigate Type I Reward Hacking, we propose POWER, a new preference optimization method that combines Guiaşu's weighted entropy with a robust reward maximization objective. POWER enjoys finite-sample guarantees under general function approximation, competing with the best covered policy in the data. To mitigate Type II Reward Hacking, we analyze the learning dynamics of preference optimization and develop a novel technique that dynamically updates preference labels toward certain "stationary labels", resulting in diminishing gradients for untrustworthy samples. Empirically, POWER with dynamic labels (POWER-DL) consistently outperforms state-of-the-art methods on alignment benchmarks, achieving improvements of up to **13.0** points on AlpacaEval 2.0 and **11.5** points on Arena-Hard over DPO, while also improving or maintaining performance on downstream tasks such as mathematical reasoning. Strong theoretical guarantees and empirical results demonstrate the promise of POWER-DL in mitigating reward hacking.

## 1 Introduction

Aligning AI systems with human values is a core problem in artificial intelligence (Russell, 2022). After training on vast datasets through self-supervised learning, large language models (LLMs) typically undergo an alignment phase to elicit desired behaviors aligned with human values (Ouyang et al., 2022). A main alignment paradigm involves leveraging datasets of human preferences, with techniques like reinforcement learning from human feedback (Christiano et al., 2017) or preference optimization (Rafailov et al., 2024b). These methods learn an (implicit or explicit) reward model from human preferences, which guides the decision-making process of the AI system. This paradigm has been instrumental in today's powerful chat models (Achiam et al., 2023; Dubey et al., 2024).

However, these alignment techniques are observed to suffer from the notorious *reward hacking* problem (Amodei et al., 2016; Tien et al., 2022; Gao et al., 2023; Casper et al., 2023), where optimizing imperfect learned reward leads to poor performance under the true reward—assuming an underlying true reward exists (Skalse et al., 2022). One primary cause of the discrepancy between the learned and true rewards arises because preference data do not encompass *all* conceivable choices, making the learned reward model vulnerable to significant statistical fluctuations in areas with sparse data. Consequently, the AI system might be swayed toward choices that only appear favorable under the learned reward but are, in reality, subpar, or the system might be deterred from truly desirable choices that do not seem favorable according to the learned rewards.

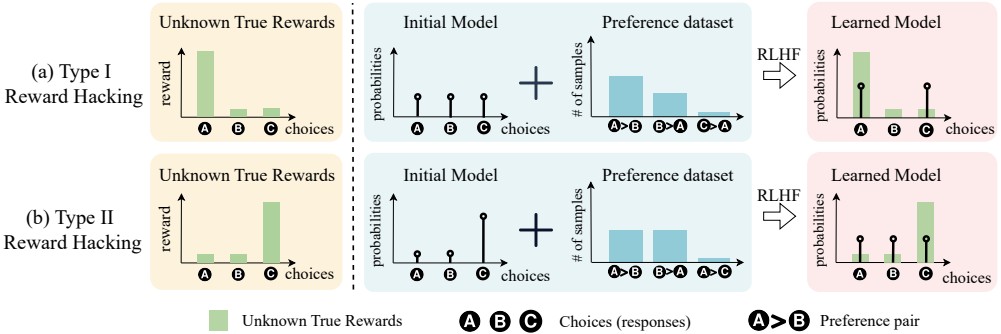

Figure 1: (a) Example of Type I Reward Hacking. The initial model has a uniform distribution over choices while the dataset has a high coverage on the high-reward choice and low coverage on a low-reward choice. With a decent chance, the poorly-covered, low-reward choice is labeled as *preferred*, causing PO methods to erroneously assign a high weight to it (Proposition 1). (b) Example of Type II Reward Hacking. The initial model is aligned with the true rewards while dataset has a low coverage on the high-reward choice. With a decent chance, the poorly-covered, high-reward choice is labeled as *rejected*, leading to deterioration of the model post alignment (Proposition 2).

In this paper, we investigate reward hacking in offline preference optimization (PO), in which we are provided with an initial AI system (initial model) and a preference dataset. We do not assume that the preference dataset is necessarily constructed through sampling from the initial model, allowing to leverage existing datasets collected from other models. Our objective is to dissect the roots of reward hacking from a statistical standpoint, analyze current methods, and introduce theoretically sound and practically strong methods to mitigate reward hacking. Our contributions are as follows.

**Types of reward hacking.** We describe two types of reward hacking in preference optimization that stem from high statistical fluctuations in regions with sparse data; see Figure 1 for an illustration. Type I Reward Hacking manifests when poorly covered, subpar choices appear more favorable than they truly are, leading the model to assign high weights to these subpar choices. Type II Reward Hacking arises when decent choices with insufficient coverage appear worse than their true value and that leads to deterioration of the initial model. While reward hacking in offline preference optimization is related to the challenge partial coverage in offline RL, the setting we consider here faces two sources of distribution shift: between the learned model and data, and between the initial model and data. This differs from offline RL, which typically considers access to an offline dataset alone (Levine et al., 2020) and is concerned with a single source of distribution shift.

**PO methods provably suffer from reward hacking.** We prove that several PO methods suffer from both types of reward hacking (Propositions 1 and 2). A common countermeasure against reward hacking is keeping the learned model close to the initial model through minimization of divergence measures (Rafailov et al., 2024b; Azar et al., 2024; Huang et al., 2024). Yet, our analysis reveals that divergence minimization does not induce sufficient pessimism to prevent Type I Reward Hacking, nor does it mitigate deterioration of the initial model caused by Type II Reward Hacking. Notably, reward hacking can occur even when divergence from the initial model is small.

**POWER-DL: Against Type I and Type II Reward Hacking.** To mitigate reward hacking, we integrate a robust reward maximization framework with Guiaşu's weighted entropy (Guiaşu, 1971). We transform this objective into a single-step optimization problem (Proposition 3) leading to Preference Optimization via Weighted Entropy Robust Rewards (POWER). We prove that POWER enjoys finite-sample guarantees for general function approximation, improving over the best covered policy and mitigating Type I Reward Hacking (Theorem 1). Due to the weighted entropy, POWER effectively learns from well-covered choices in the dataset, even those with a large divergence against the initial model, countering potential underoptimization in divergence-based methods. We next develop Dynamic Labels to mitigate Type II Reward hacking, whereby preference labels are updated in a way that diminishes gradients for untrustworthy data (Theorem 2). Our final algorithm combines POWER with Dynamic Labels (POWER-DL), which interpolates robust rewards with maintaining closeness to the initial model, allowing to trade off between reward hacking types.

**POWER-DL consistently outperforms other methods across various settings.** For aligning LLMs, we implement POWER-DL and compare it against other preference optimization methods

across different datasets and two scenarios: one using an existing preference dataset and another with preference data generated through sampling from the initial model. POWER-DL consistently outperforms state-of-the-art methods in alignment benchmarks, achieving improvements over DPO of up to **13.0** points on AlpacaEval 2.0 and **11.5** points on Arena-Hard. Additionally, POWER-DL improves or maintains performance on downstream tasks such as truthfulness, mathematical reasoning, and instruction-following, demonstrating robustness against reward hacking and achieving a more favorable bias-variance trade-off compared to other methods.

## 2 BACKGROUND AND PROBLEM FORMULATION

### 2.1 LEARNING FROM HUMAN PREFERENCE

**Contextual bandit formulation.** We adopt the contextual bandits formulation described by a tuple $(\mathcal{X}, \mathcal{Y}, r)$, where $\mathcal{X}$ is the space of contexts (e.g., prompts), $\mathcal{Y}$ is the space of actions (e.g., responses), and $r : \mathcal{X} \times \mathcal{Y} \to \mathbb{R}$ is a scalar reward function. A stochastic policy (e.g., model or language model) $\pi : \mathcal{X} \to \Delta(\mathcal{Y})$ takes in a context $x \in \mathcal{X}$ and outputs an action according to $y \sim \pi(\cdot|x)$. We denote the set of all stochastic policies by $\Pi := \{\pi : \mathcal{X} \to \Delta(\mathcal{Y})\}$.

**Performance metric.** We assume that there exists an underlying (unknown) true reward function $r^\star : \mathcal{X} \times \mathcal{Y} \to \mathbb{R}$. Given the true reward function $r^\star$ and a target distribution over contexts $x \sim \rho(\cdot)$, performance of a policy $\pi$ is the expected true reward over contexts and actions

$$J(\pi) := \mathbb{E}_{x \sim \rho, y \sim \pi(\cdot|x)} \left[ r^\star(x, y) \right]. \tag{1}$$

**The Bradley-Terry model of human preferences.** Consider a prompt $x \in \mathcal{X}$ and a pair of responses $y^0, y^1 \in \mathcal{Y}$. For any reward function $r$, the Bradley-Terry (BT) model characterizes the probability of preferring $y^1$ over $y^0$, denoted by $l = 1$, according to:

$$\mathbb{P}_r(l = 1 \mid x, y^1, y^0) = \sigma \left( r(x, y^1) - r(x, y^0) \right), \tag{2}$$

where $\sigma(z) := 1/(1 + \exp(-z))$ is the sigmoid function.

**Offline preference optimization.** We consider an offline learning setup, where we start from an initial reference policy (model), denoted by $\pi_{\theta_0} = \pi_{\text{ref}}$, and an offline pairwise preference dataset $\mathcal{D} = \{(x, y^0, y^1, l)\}$, comprising of $N$ iid samples. Prompt and response pairs are sampled according to a data distribution: $x, y^0, y^1 \sim \mu$, and preferences label is sampled according to the BT model corresponding to true rewards: $l \sim \mathbb{P}_{r^\star}(\cdot|x, y^1, y^0)$. Importantly, we do *not* assume that the preference dataset is necessarily constructed through sampling from the initial model. To simplify notation, we define $y^+ = ly^1 + (1-l)y^0$ and $y^- = (1-l)y^1 + ly^0$ to denote the chosen and rejected responses in the dataset, respectively. Appendix A presents additional notation.

### 2.2 DIRECT PREFERENCE OPTIMIZATION

A classical approach to learning from human preferences involves learning a reward model from dataset, followed by finding a policy through maximizing the learned reward typically regularized with a (reverse) KL-divergence to keep the learned policy closed to initial policy:

$$\begin{aligned} \hat{r} &\in \arg\min_r L_{\text{BT}}(r) := -\mathbb{E}_{\mathcal{D}} \left[ \log \sigma \left( r(x, y^+) - r(x, y^-) \right) \right] \\ \hat{\pi} &\in \arg\max_\pi \mathbb{E}_{x \sim \rho, y \sim \pi}[\hat{r}(x, y)] - \beta D_{\text{KL}}[\pi \| \pi_{\text{ref}}], \end{aligned} \tag{3}$$

Here, $D_{\text{KL}}[\pi \| \pi_{\text{ref}}] := \mathbb{E}_{x \sim \rho} \left[ D_{\text{KL}}[\pi(\cdot|x) \| \pi_{\text{ref}}(\cdot|x)] \right]$ and $L_{\text{BT}}(r)$ is the negative log-likelihood according to the BT model. Rafailov et al. (2024b) observed that the policy maximization step in (3) can be computed in closed form and thus simplified the two-step process into a single minimization objective. This method is called direct preference optimization (DPO) and has inspired a series of works; see Tables 1 and 3 for several examples.

Some representative variants of DPO that we theoretically analyze are IPO (Azar et al., 2024), which applies a nonlinear transformation to preferences to reduce overfitting, and SimPO (Meng et al., 2024), which removes the reference policy from the DPO objective. We also analyze two recent theoretical methods that come with finite-sample guarantees and aim at mitigating overoptimization: $\chi$PO (Huang et al., 2024), which replaces the KL divergence in DPO with a stronger $\chi^2$+KL

Table 1: Preference optimization objectives given data $\mathcal{D} = \{(x, y^+, y^-)\}$ and initial model $\pi_{\text{ref}}$.

| Method | Objective |
|---|---|
| DPO (Rafailov et al., 2024b) | $\hat{\pi}_{\text{DPO}} \in \arg\min_\theta -\mathbb{E}_{\mathcal{D}} \left[ \log \sigma \left( \beta \left( \log \frac{\pi_\theta(y^+\|x)}{\pi_{\text{ref}}(y^+\|x)} - \log \frac{\pi_\theta(y^-\|x)}{\pi_{\text{ref}}(y^-\|x)} \right) \right) \right]$ |
| DPO+SFT (Liu et al., 2024) | $\hat{\pi}_{\text{DPO+SFT}} \in \arg\min_\theta -\mathbb{E}_{\mathcal{D}} \left[ \log \sigma \left( \beta \left( \log \frac{\pi_\theta(y^+\|x)}{\pi_{\text{ref}}(y^+\|x)} - \log \frac{\pi_\theta(y^-\|x)}{\pi_{\text{ref}}(y^-\|x)} \right) \right) \right] - \mathbb{E}_{\mathcal{D}} \left[ \log \pi_\theta(y^+\|x) \right]$ |
| IPO (Azar et al., 2024) | $\hat{\pi}_{\text{IPO}} \in \arg\min_\theta \mathbb{E}_{\mathcal{D}} \left[ \left( \log \frac{\pi_\theta(y^+\|x)}{\pi_{\text{ref}}(y^+\|x)} - \log \frac{\pi_\theta(y^-\|x)}{\pi_{\text{ref}}(y^-\|x)} - \frac{1}{2\tau} \right)^2 \right]$ |
| SimPO (Meng et al., 2024) | $\hat{\pi}_{\text{SimPO}} \in \arg\min_\theta -\mathbb{E}_{\mathcal{D}} \left[ \log \sigma \left( \beta \left( \frac{1}{\|y^+\|} \log \pi_\theta(y^+\|x) - \frac{1}{\|y^-\|} \log \pi_\theta(y^-\|x) \right) - \gamma \right) \right]$ |
| $\chi$PO (Huang et al., 2024) | $\hat{\pi}_{\chi\text{PO}} \in \arg\min_\theta -\mathbb{E}_{\mathcal{D}} \left[ \log \sigma \left( \text{clip}_{2R} \left[ \beta \left( \phi \left( \frac{\pi_\theta(y^+\|x)}{\pi_{\text{ref}}(y^+\|x)} \right) - \phi \left( \frac{\pi_\theta(y^-\|x)}{\pi_{\text{ref}}(y^-\|x)} \right) \right) \right] \right) \right]; \ \phi(z) := z + \log(z)$ |

divergence, and DPO+SFT (Liu et al., 2024; Cen et al., 2024), which adds a supervised finetuning term that increases log-likelihood of chosen responses in the preference dataset.

## 3 REWARD HACKING IN PREFERENCE OPTIMIZATION

In this section, we investigate reward hacking in preference optimization. One driver of reward hacking is statistical errors present in the dataset. Typically, preference datasets suffer from *partial coverage*, lacking extensive samples across all possible options. As a result, preferences for poorly covered choices are subject to high levels of statistical fluctuations, given the fact that preference labels are Bernoulli random variables with probabilities described by the Bradley-Terry model (2). Subsequently, we describe two types of reward hacking, both originating from the presence of poorly covered choices (actions) in the dataset.

### 3.1 TYPE I REWARD HACKING

Type I Reward Hacking occurs when poorly covered, *subpar* choices in the dataset appear more favorable due to statistical errors, and that leads to a learned policy $\hat{\pi}$ with a low expected true reward $J(\hat{\pi})$. In the following proposition, we prove that even in the favorable scenario that the high-reward actions are well-covered in the dataset, the existence of a single sample on a low-reward action can overwhelm many preference optimization algorithms, causing them to learn highly suboptimal policies. Figure 1(a) provides an illustration of one of the failure instances analyzed in this proposition and proof is presented in Appendix C.1.

**Proposition 1 (Type I Reward Hacking in $\star$PO).** *Consider multi-armed bandits with bounded rewards $r^\star(a) \in [0, 1]$ and the softmax policy class, defined as*

$$\Pi_\theta := \left\{ \pi_\theta(y) = \exp(\theta(y))/Z_\theta \Big| Z_\theta = \sum_y \exp(\theta(y)), \theta(y) \in [0, 1] \right\}. \tag{4}$$

*Define the best-in-class policy $\pi_{\theta^\star} = \max_{\pi \in \Pi_\theta} J(\pi)$. There exist three-armed bandit instances with $\Pi_\theta$ parameterization, high coverage of the optimal arms $\mu(a \in \arg\max_a r^\star(a)) > 1/2$, and bounded KL-divergence $D_{KL}(\pi_{\theta^\star} \mid \pi_{ref})$, such that for any $N \geq 2$, $\beta > 0$, $\gamma, \tau > 0$, policy $\hat{\pi} \in \{\hat{\pi}_{DPO}, \hat{\pi}_{IPO}, \hat{\pi}_{SimPO}\}$ or $\hat{\pi} = \hat{\pi}_{\chi PO}$ for $0 < \beta \leq 1/3$, suffers from a constant suboptimality $J(\pi_{\theta^\star}) - J(\hat{\pi}) > 0.15$ with a constant probability of at least $(e(1 + e))^{-1}$.*

Type I Reward Hacking and the failure result in Proposition 1 are closely connected to the challenge of partial data coverage in offline RL (Levine et al., 2020), which can be robustly addressed through the principle of pessimism in the face of uncertainty. Pessimism can be applied in various ways such as reducing the rewards (values) of poorly covered actions (Kumar et al., 2020; Cheng et al., 2022) or keeping the learned policy close to data collection policy (Nachum et al., 2019). Although divergence-based methods such as DPO and IPO aim at keeping the learned policy close to the initial policy, Proposition 1 shows that maintaining a small KL divergence from initial model does not induce a sufficient amount of pessimism to prevent Type I Reward Hacking.[1]

---

[1]Proposition 1 does not contradict guarantees of Huang et al. (2024) as this work assumes that preference data are collected from the initial policy. However, this assumption is restrictive, as it prevents using existing

**Remark 1.** *Failure result in Proposition 1 is rigorous and constructed under a realistic setting close to practice: the policy class is a softmax with bounded rewards and the KL divergence between initial and best-in-class policy is bounded. This makes Proposition 1 stronger than prior arguments on overoptimization in DPO, which rely on unbounded rewards (Azar et al., 2024), updates to model parameters despite receiving no samples (hence, conclusion breaking in gradient-based optimization) (Huang et al., 2024), or events with probabilities approaching zero (Song et al., 2024).*

## 3.2 TYPE II REWARD HACKING

Type II Reward Hacking can occur when poorly covered, *good* choices in the dataset appear to be less favorable than their true value due to statistical errors, leading to the deterioration of the initial model after preference optimization. An example of this type of reward hacking is illustrated in Figure 1(b), where the initial policy has a high probability on the high-reward choice. Yet, due to the low coverage of the high-reward choice in the data, by mere chance this choice can appear unfavorable, resulting in the deterioration of the initial model. In the following proposition, we prove that many preference optimization methods are susceptible to Type II Reward Hacking. Proof of this proposition can be found in Appendix C.2.

**Proposition 2 (Type II Reward Hacking in $\star$PO).** *Consider the multi-armed bandits setting with the softmax policy class $\Pi_\theta$, as defined in (4). Let $\pi_{\theta^\star} = \max_{\pi \in \Pi_\theta} J(\pi)$ represent the best-in-class policy. There exists a three-armed bandit problem with $\Pi_\theta$ parameterization and $\pi_{\theta_0} = \pi_{\theta^\star}$, such that for any $N \geq 3$, $\beta > 0, \eta \geq 0, \gamma$ and policy $\hat{\pi} \in \{\hat{\pi}_{DPO}, \hat{\pi}_{DPO+SFT}, \hat{\pi}_{SimPO}\}$ or $\hat{\pi} \in \{\hat{\pi}_{\chi PO}, \hat{\pi}_{IPO}\}$ for $0 < \beta, \tau \leq 1$, the following holds with a constant probability of at least $(e(1+e))^{-1}$:*

$$J(\pi_{\theta^\star}) - J(\hat{\pi}) > 0.1.$$

Proposition 2 states that even with a strong initial model, a preference dataset that poorly covers high-reward actions can lead to substantial deterioration of the initial model in existing approaches, even in methods such as DPO, IPO, and $\chi$PO that incorporate divergence-minimization. We note that the above setting is beyond the guarantees of traditional pessimistic offline RL, as these techniques typically do not consider access to an initial model and guarantee competing with the best covered policy in the data. Despite this, as we see in Section 5, there may be hope to mitigate the degradation of the initial model and better control the trade-off between Type I and Type II Reward Hacking.

# 4 AGAINST TYPE I REWARD HACKING: WEIGHTED ENTROPY ROBUST REWARD MAXIMIZATION

## 4.1 WEIGHTED ENTROPY REWARD MAXIMIZATION

We demonstrated that approaches involving divergence minimization remain vulnerable to reward hacking. Moreover, maintaining a small divergence can inadvertently lead to *underoptimizing* the preference dataset, as it may risk overlooking policies that, although well-covered, deviate significantly from the initial policy.[2] These reasons motivate us to explore an alternative route and consider regularizing the reward maximization objective with the concept of *weighted entropy*.

**Definition 1 (Weighted Entropy; Guiaşu (1971)).** *The weighted entropy of a (discrete) distribution $p(\cdot)$ with non-negative weights $w(y)$ is defined as $H_w(p) := -\sum_y w(y)p(y)\log p(y)$.*

Weighted entropy extends Shannon's entropy by incorporating weights associated with each outcome, reflecting attributes such as favorableness or utility toward a specific goal (Guiaşu, 1971). Building on this, we consider a weighted-entropy reward (WER) maximization objective:

$$\max_{\pi \in \Pi} \mathbb{E}_{x \sim \rho, y \sim \pi}\left[r(x,y)\right] + \beta H_w(\pi), \tag{5}$$

where $H_w(\pi) := \mathbb{E}_{x \sim \rho}[H_w(\pi(\cdot|x))]$. This objective expresses the principle of maximum (weighted) entropy (Jaynes, 1957; Guiasu & Shenitzer, 1985), promoting the selection of policies with maximum entropy—thus favoring the most uniform or unbiased policies—among those compatible with

---

preference datasets collected from other models, which is a common approach in practical pipelines such as Wang et al. (2024e) and Tunstall et al. (2023).

[2]Simply reducing $\beta$ to alleviate underoptimization may not always be viable. For example, reducing $\beta$ may reduce underoptimization in one state while inadvertently amplify overoptimization in another state.

the constraints, such as achieving high rewards. Objective (5) extends the well-established maximum entropy framework in RL, used in various settings such as exploration (Haarnoja et al., 2018), inverse RL (Ziebart et al., 2008), and robust RL (Eysenbach & Levine, 2022).

## 4.2 POWER: PREFERENCE OPTIMIZATION WITH WEIGHTED ENTROPY ROBUST REWARDS

To mitigate reward hacking, we integrate the WER objective (5) with a robust (adversarial) reward framework, inspired by favorable theoretical guarantees (Liu et al., 2024; Cen et al., 2024; Fisch et al., 2024). Specifically, we find a policy that maximizes WER (5) against an adversarial reward, which seeks to minimize WER while fitting the preference dataset:

$$\max_{\pi \in \Pi} \min_{r \in \mathcal{R}} \underbrace{L_{\text{BT}}(r)}_{\text{negative log-likelihood}} + \eta \underbrace{\Big( \mathbb{E}_{x \sim \rho, y \sim \pi}\left[r(x,y)\right] - \mathbb{E}_{x \sim \rho, y' \sim \pi'}\left[r(x,y')\right] + \beta H_w(\pi) \Big)}_{\text{WER minus baseline}}, \quad (6)$$

where $\eta \geq 0$, $\mathcal{R}$ is a reward function class, and $L_{\text{BT}}(r)$ is the negative log-likelihood of the BT model. We subtracted a baseline $\mathbb{E}_{x \sim \rho, y' \sim \pi'}\left[r(x,y')\right]$ that computes the average reward with respect to some policy $\pi'$, as the preference data under the BT model only reveal information on reward differences (Zhan et al., 2023a). This baseline plays a crucial role in simplifying the objective and establishing finite-sample guarantees, and we subsequently discuss reasonable choices for $\pi'$.

Under some regularity conditions (detailed in Appendix D.2), the objective (6) can be equivalently expressed as a minimax problem, which leads to a single-step preference objective presented in the proposition below; see Appendix D.3 for the derivation and proof.

**Proposition 3 (POWER Objective).** *Let $w(y) > 0$ denote the weights in the weighted entropy $H_w(\pi)$ and $\pi_r$ denote the policy that maximizes the objective (5). Under certain regularity conditions on $\mathcal{R}$ (Assumption 1) and for any $\beta > 0$, solving the maximin objective (6) is equivalent to solving the following optimization problem:*

$$\min_{r \in \mathcal{R}} L_{BT}\Big( \beta\big( w(y) \log \pi_r(y|x) + w(y) \big) \Big) - \eta \mathbb{E}_{x \sim \rho, y' \sim \pi'}\Big[ \beta w(y') \log \pi_r(y'|x) \Big]. \quad (7)$$

We call the above objective preference optimization via weighted entropy robust rewards, or POWER. The first term in the above objective is the Bradley-Terry loss with rewards set to $w(y) \log \pi_r(y|x) + w(y)$, resulting in a reward gap expressed as a weighted difference of log probabilities of chosen and rejected responses. The second expectation is a *weighted* negative log-likelihood (a.k.a. supervised fine-tuning, or SFT) regularizer over the baseline policy $\pi'$.

**Remark 2.** *Liu et al. (2024) propose an adversarial objective similar to (6) that uses KL divergence instead of weighted entropy. From a theoretical perspective, our approach with weighted entropy improves over this work, such as through mitigating underoptimization; see Section 4.3 for details. Moreover, our final algorithm presented in Algorithm 1 is considerably different from the DPO+SFT objective (Liu et al., 2024; Pal et al., 2024) and significantly outperforms empirically (Section 6).*

## 4.3 FINITE-SAMPLE GUARANTEES AND THEORETICAL BENEFITS OF POWER

The following theorem shows that POWER enjoys finite-sample guarantees on performance.

**Theorem 1 (Finite-Sample Performance Guarantees of POWER).** *Given a competing policy $\pi \in \Pi$, assume a bounded concentrability coefficient $C_\mu^\pi(\mathcal{R}, \pi') < \infty$ as defined in Definition 2. Furthermore, assume realizability of the true reward function $r^\star \in \mathcal{R}$, boundedness of rewards $\forall r \in \mathcal{R} : r(x,y) \in [0, R]$, and that the reward function class has a finite $\epsilon$-covering number $\mathcal{N}_\epsilon$ under the infinity norm. Define $\tilde{R} := 1 + \exp(R), \epsilon \asymp (\tilde{R}N)^{-1}, \iota = \sqrt{\log(\mathcal{N}_\epsilon)/\delta}$. Set $\eta \asymp \iota/(\tilde{R}^2\sqrt{N})$ and $\beta = 1/\sqrt{N}$ in the objective (6). Then with a probability of at least $1 - \delta$, policy $\hat{\pi}_{POWER}$ that solves (6) satisfies the following*

$$J(\hat{\pi}_{POWER}) \gtrsim J(\pi) - \frac{1}{\sqrt{N}} \left( \left( [C_\mu^\pi(\mathcal{R}, \pi')]^2 + 1 \right) \tilde{R}^2 \iota + H_w(\pi) \right).$$

*Furthermore, let $L$ denote the maximum response length. Selecting $w(y) = 1/|y|$ to be the inverse response length, one has*

$$J(\hat{\pi}_{POWER}) \gtrsim J(\pi) - \frac{1}{\sqrt{N}} \left( \left( [C_\mu^\pi(\mathcal{R}, \pi')]^2 + 1 \right) \tilde{R}^2 \iota + \log(L|\mathcal{V}|) \right).$$

Proof of the above theorem can be found in Appendix E.2. Below, we discuss the implications of the above theorem and the guidelines it offers for practical choices.

**Guarantees against Type I Reward Hacking.** Theorem 1 shows that the policy learned by POWER competes with the best policy *covered* in the dataset, where the notion of coverage is characterized by single-policy concentrability, considered the gold standard in offline RL (Rashidinejad et al., 2021; Xie et al., 2021; Zhan et al., 2023a). This implies that as long as a favorable policy is covered in the preference data, POWER is robust to existence of poorly covered, subpar policies and thus mitigates Type I Reward Hacking. Moreover, as Theorem 1 does not impose a parametric form on the class $\mathcal{R}$, guarantees hold for general function classes.

**Benefits of weighted entropy and choice of weights.** A non-zero weighted entropy term ($\beta > 0$) is essential in obtaining the one-step optimization problem in (7) and establishing its equivalence to the maximin problem, as this term induces strict concavity in the objective (5). Moreover, using KL-regularization leads to rates that grow with the divergence of competing policy and initial policy (Liu et al., 2024), which can be large or even unbounded. However, weighted entropy ensures bounded rates and thus mitigates underoptimization. Theorem 1 suggests that a particularly appealing choice for weights is the inverse response length $w(y) = 1/|y|$, which intuitively discourages learning policies that generate lengthy responses. Theoretically, while using Shannon entropy ($w(y) = 1$) results in a convergence rate that grows linearly with response length, with weights $w(y) = 1/|y|$ convergence rate only depends on the logarithm of the vocabulary (token) size and logarithm of response length. Other choices for weights include response preference scores and per-sample importance weights.

**Choice of the baseline policy.** The rate in Theorem 1 is influenced by the concentrability coefficient $C_\mu^\pi(\mathcal{R}, \pi')$, which is impacted by the baseline policy $\pi'$. Inspecting the definition of concentrability coefficient in Definition 2, a reasonable choice to make this coefficient small is selecting the distribution of chosen responses in the dataset (Zhan et al., 2023a; Liu et al., 2024).

With the above choices applied to objective 7, a practically appealing version of POWER becomes:

$$\max_\theta \mathbb{E}_\mathcal{D}\left[\log\sigma\left[\beta\left[\frac{\log\pi_\theta(y^+|x)}{|y^+|} - \frac{\log\pi_\theta(y^-|x)}{|y^-|} + \frac{1}{|y^+|} - \frac{1}{|y^-|}\right]\right]\right] + \eta\beta\mathbb{E}_\mathcal{D}\left[\frac{\log\pi_\theta(y^+|x)}{|y^+|}\right] \quad (8)$$

**Remark 3.** *Objective (8) shares similarities to SimPO (Meng et al., 2024) but has important differences. First, objective (8) includes a length-normalized SFT term, which is key in mitigating Type I Reward Hacking (Theorem 1, Proposition 4), from which SimPO suffers (Proposition 1). Second, our approach analytically leads to the margin $1/|y^+| - 1/|y^-|$ while SimPO uses a fixed hyperparameter. Lastly, our objective is rooted in theory and enjoys finite-sample guarantees.*

### 4.4 POWER FACED WITH HARD INSTANCES AND THE ROLE OF PARTITION FUNCTION

In the following proposition, we analyze the POWER objective (8) in the hard reward hacking instances of Proposition 1 and Proposition 2. Proof is presented in Appendix C.3.

**Proposition 4.** *(I) Consider the three-armed bandit instance in Proposition 1. Then, for any $\beta > 0$ and $\eta > \frac{(2+e)}{N-(2+e)}$, POWER policy $\hat{\pi}_{POWER}$ that solves the objective (8) is the best-in-class policy: $\hat{\pi}_{POWER} = \pi_{\theta^\star}$. (II) Consider the three-armed bandit instance in Proposition 2. Then, for any $\beta > 0, \eta \geq 0$, policy $\hat{\pi}_{POWER}$ that solves the objective (8) suffers from a constant suboptimality $J(\pi_{\theta^\star}) - J(\hat{\pi}_{POWER}) > 0.2$.*

Proposition 4 confirms that POWER robustly (for any $\beta > 0$ and $\eta \gtrsim 1/N$) prevents Type I Reward Hacking in the hard instance of Proposition 1, where other preference optimization algorithms DPO, SimPO, IPO, and $\chi$PO fail. Yet, the above proposition shows that POWER suffers from Type II Reward Hacking, which the design dynamic labels in the following section.

## 5 AGAINST TYPE II REWARD HACKING: DYNAMIC LABELS

We now turn our focus to mitigating Type II Reward Hacking, based on the following intuition: keeping the model's internal preferences close to initialization in the low-coverage regions (trust the

preference labels less) while learning from the data in the high-coverage regions (trust the preference labels more).

For this purpose, we analyze the learning dynamics of preference optimization in the bandits setting with softmax parameterization $\pi_\theta(y) \propto \exp(\theta(y))$. We denote the dataset by $\mathcal{D} = \{(y^0, y^1, l)\}$ with labels $l \sim \mathbb{P}_{r^\star}(\cdot | y^0, y^1)$. We use $\hat{\mu}_{0,1}$ to indicate the empirical probability of comparing $y^0$ with $y^1$, and $\hat{\mu}_{1 \succ 0}$ the empirical probability of preferring $y^1$ over $y^0$. To simplify presentation, we consider POWER with $w(y) = 1, \eta = 0, \beta = 1$; a similar analysis can be extended to other objectives.

**Reverse engineering label updates based on learning dynamics.** Rather than using static preference labels, we allow the labels $l_t$ to evolve across gradient updates. Denote the parameter gap corresponding to two actions $y^0, y^1$ by $d_{\theta_t}(y^1, y^0) := \theta_t(y^1) - \theta_t(y^0)$. We show in Appendix F.1 that isolated (batch) gradient updates on $y^0$ and $y^1$ is:

$$d_{\theta_{t+1}}(y^1, y^0) = d_{\theta_t}(y^1, y^0) + \alpha\hat{\mu}_{0,1}\Big[(\hat{\mu}_{1 \succ 0} - \hat{\mu}_{0 \succ 1})l_t - \big(\sigma\big(d_{\theta_t}(y^1, y^0)\big) - \hat{\mu}_{0 \succ 1}\big)\Big] \quad (9)$$

We design labels $l_t$ so that gradient updates are rapidly diminished for poorly covered preference pairs, ensuring that preferences for such pairs remain close to initialization. To achieve this, we first directly set the gradient in (9) to zero and derive a "stationary" label $\bar{l}_t$:

$$(\hat{\mu}_{1 \succ 0} - \hat{\mu}_{0 \succ 1})l_t - (\sigma(d_{\theta_t}(y^1, y^0)) - \hat{\mu}_{0 \succ 1}) = 0 \quad \Rightarrow \quad \bar{l}_t = \frac{\sigma\big(d_{\theta_t}(y^1, y^0)\big) - \hat{\mu}_{0 \succ 1}}{\hat{\mu}_{1 \succ 0} - \hat{\mu}_{0 \succ 1}}. \quad (10)$$

$\bar{l}_t$ represents the ratio between a *learned* preference gap and the *empirical* preference gap, and we have $\bar{l}_t = 1$ when learned and empirical preferences are equal. To implement dynamic preference labels, we employ the following update rule for $l_t$, where $\gamma$ ranges between 0 and 1:

$$l_{t+1} = (1 - \gamma)l_t + \gamma\bar{l}_t, \quad l_0 = 1. \quad (11)$$

In the following theorem, we analyze the coupled dynamical systems described by equations (11) and (9); see Appendix F.2 for the proof.

**Theorem 2** (**Learning Dynamics with Label Updates**). *Consider the following set of differential equations with initial values $l_0 = 1$ and any $d_0$:*

$$\begin{aligned}
\dot{d}_t &= \alpha\hat{\mu}_{0,1}\Big(((\hat{\mu}_{1 \succ 0} - \hat{\mu}_{0 \succ 1})l_t - (\sigma(d_t) - \hat{\mu}_{0 \succ 1}))\Big) \\
\dot{l}_t &= -\frac{\gamma}{\hat{\mu}_{1 \succ 0} - \hat{\mu}_{0 \succ 1}}\Big(((\hat{\mu}_{1 \succ 0} - \hat{\mu}_{0 \succ 1})l_t - (\sigma(d_t) - \hat{\mu}_{0 \succ 1}))\Big)
\end{aligned} \quad (12)$$

*Assume $\hat{\mu}_{1 \succ 0} > 1/2$ and let $c = \min\{\sigma(d_0)(1 - \sigma(d_0)), \hat{\mu}_{1 \succ 0}\hat{\mu}_{0 \succ 1}\}$. For any $\epsilon_l \ll 1$, fix $\mu_l, \mu_h, T, \gamma, \hat{\mu}_{0,1}, \alpha$ such that $\alpha\mu_l/\epsilon_l \leq \gamma \leq 1/2 \exp(-1/4)\alpha\mu_h \leq 1$ and $\alpha\hat{\mu}_{1 \succ 0}T \geq 1$.*

1. *(Low Coverage Case) When $\hat{\mu}_{0,1} \leq \mu_l$, we have $|\sigma(d_T) - \sigma(d_0)| \leq |d_T - d_0| \leq \epsilon_l$.*

2. *(High Coverage Case) When $\hat{\mu}_{0,1} \geq \mu_h$, we have $(\sigma(d_T) - \hat{\mu}_{1 \succ 0})^2 \leq \exp\big(-\alpha c\hat{\mu}_{0,1}T\big)$.*

The above theorem shows that for poorly covered pairs (small $\hat{\mu}_{1,0}$), learned preferences $\sigma(d_T)$ remain close to initialization, while for high coverage pairs (large $\hat{\mu}_{1,0}$), learned preferences converge to empirical preferences. In a sense, $\gamma$ determines the level of *conservatism*, adjusting the threshold of what considered poor coverage. Moreover, the convergence rate in the high-coverage case is impacted by empirical preferences through $c$. In the case of nearly equal preferences $\hat{\mu}_{1 \succ 0} \approx 1/2$, the convergence rate is faster, whereas in the case of strong preference with $\hat{\mu}_{1 \succ 0} \to 1$, the convergence rate is slower suggesting that more updates are required to further distinguish the two choices.

**Remark 4** (Related work on soft labels in RLHF). *In preference optimization, Mitchell (2024) considers noisy preference labels and incorporates constant soft labels through linear interpolation. Concurrent work by Furuta et al. (2024) develop a geometric averaging approach, in which samples are weighted according to the preference gap using scores from a reward model. In the context of reward learning, Zhu et al. (2024) propose iterative data smoothing that updates labels toward learned preferences. In contrast to these methods, our approach is rooted in updating labels to shrink gradients of poorly covered pairs via a general recipe whereby dynamic labels are smoothly updated toward labels that set the gradient to zero. This approach goes beyond constant soft labels and does not require scores from an extra reward or preference model. Moreover, label updates in prior works do not guarantee remaining close to a (non-uniform) initial model in the low coverage areas, which aims at mitigating Type II Reward Hacking in the offline alignment setting.*

**POWER with Dynamic Labels.** Our final algorithm POWER-DL (Algorithm 1) integrates the POWER objective with dynamic labels against reward hacking. In untrustworthy regions, POWER-DL interpolates between the initial model and robust rewards, allowing to trade off the two types of reward hacking through adjusting conservatism parameters $\eta$ and $\gamma$, reflecting relative quality of the initial model compared to preference data and up to removing conservatism completely by setting $\eta = \gamma = 0$. We highlight the fact that divergence-based methods aim at keeping the learned model close to the initial model wherever the learned model has a decent probability, regardless of data coverage. In contrast, the dynamic label procedure aims at keeping the learned model close to the initialization only in the untrustworthy regions while learning from the data in high coverage region, which can alleviate potential over-pessimism. All these factors can lead to a better performance, as supported by our empirical evaluations in Section 6. See Appendix B.3 for further discussion.

## 6    EXPERIMENTS

### 6.1    EXPERIMENTAL SETUP

We conduct experiments to assess different PO methods on aligning LLMs across four settings, varying in dataset size and level of distribution shift between the initial model and data. We follow two pipelines: Helpsteer2 (Wang et al., 2024e), which employs smaller datasets, and Zephyr (Tunstall et al., 2023) with significantly larger datasets. We implement two distinct setups similar to Meng et al. (2024): the *base* setup that uses an existing preference dataset and the *instruct* setup that constructs a preference dataset by sampling from the initial model. These two setups allow evaluating across different levels of *distribution shift* between the initial model and preference data.

**Helpsteer2 setups.** In the base setup, we train Llama-3-8B on the OpenAssistant2 dataset (Köpf et al., 2024) to create the initial model. We conduct preference optimization using the Helpsteer2 dataset (Wang et al., 2024e), selecting responses based on helpfulness scores and discarding ties, yielding about 7K samples. In the instruct setup, we use Llama-3-8B-Instruct as the initial model and generate a preference dataset from Helpsteer2 prompts. Following Wang et al. (2024e), we generate 10 responses per prompt with temperature 0.7. We then score them with Armo reward model (Wang et al., 2024c) and select the highest and lowest score responses as $y^+$ and $y^-$, respectively.

**Zephyr setups.** In the base setup, we obtain the initial model by training Llama-3-8B base model on the UltraChat-200K dataset (Ding et al., 2023). We then perform preference optimization on the UltraFeedback dataset (Cui et al., 2024), comprising approximately 61K samples. In the instruct setup and following Meng et al. (2024), we start from Llama-3-8B-Instruct and generate 5 responses with temperature 0.8 per prompt in the UltraFeedback dataset. As before, the highest and lowest score responses are selected as preference response pairs.

**Evaluation benchmarks.** We primarily assess preference methods by evaluating the trained models on standard instruction-following benchmarks: AlpacaEval 2.0 (Li et al., 2023a; Dubois et al., 2024) and Arena-Hard (Li et al., 2024), which evaluate the quality of the model responses. Following standard guidelines, for Arena-Hard, we report the win rate (WR) of the model's responses against responses from GPT-4-Turbo. For AlpacaEval 2.0, in addition to the WR against GPT-4-Turbo, we report the length-controlled (LC) win rate, designed to mitigate bias toward verbosity. We further evaluate the performance of models on MT-Bench (Zheng et al., 2023) and downstream tasks such as mathematics, reasoning, truthfulness, and instruction-following (Beeching et al., 2023).

**Preference optimization methods.** We compare POWER-DL against various baselines; see Appendix H.1 for details. These include divergence-base methods DPO (Rafailov et al., 2024b), IPO (Azar et al., 2024), offline SPPO (Wu et al., 2024b), and $\chi$PO (Huang et al., 2024), along with robust variants such as conservative DPO (cDPO) (Mitchell, 2024), robust preference optimization (ROPO) (Liang et al., 2024), R-DPO (Park et al., 2024), and DPO+SFT (Pal et al., 2024; Liu et al., 2024). We also evaluate against reference-free methods CPO (Xu et al., 2024a), SLiC-HF (Zhao et al., 2023), RRHF (Yuan et al., 2024a), ORPO (Hong et al., 2024), and SimPO (Meng et al., 2024).

### 6.2    BENCHMARK RESULTS

**POWER-DL outperforms SoTA methods on alignment benchmarks.** Table 2 presents the results on alignment benchmarks. POWER-DL consistently outperforms other methods in both Helpsteer2

Table 2: AlpacaEval 2 and Arena-Hard results on Helpsteer2 and Zephyr settings.

| Method | Helpsteer2 | | | | | | Zephyr | | | | | |
| --- | --- | --- | --- | --- | --- | --- | --- | --- | --- | --- | --- | --- |
| | Llama3-8B-Base | | | Llama3-8B-Instruct | | | Llama3-8B-Base | | | Llama3-8B-Instruct | | |
| | AlpacaEval | | Arena-Hard | AlpacaEval | | Arena-Hard | AlpacaEval | | Arena-Hard | AlpacaEval | | Arena-Hard |
| | LC(%) | WR(%) | WR(%) | LC(%) | WR(%) | WR(%) | LC(%) | WR(%) | WR(%) | LC(%) | WR(%) | WR(%) |
| Initial Model | 8.02 | 5.42 | 2.4 | 33.41 | 32.40 | 23.0 | 4.76 | 2.83 | 2.0 | 33.41 | 32.40 | 23.0 |
| DPO | 18.52 | 14.99 | 10.0 | 40.87 | 39.05 | 29.6 | 22.53 | 17.84 | 13.3 | 44.20 | 43.63 | 38.4 |
| DPO+SFT | 18.33 | 12.93 | 7.9 | 39.85 | 37.51 | 27.0 | 19.11 | 14.69 | 9.5 | 45.98 | 44.07 | 39.0 |
| cDPO | 19.06 | 14.65 | 8.5 | 42.27 | 40.36 | 34.4 | 21.06 | 16.33 | 11.4 | 44.96 | 44.37 | 39.5 |
| R-DPO | 11.03 | 15.20 | 8.3 | 33.67 | 33.89 | 25.7 | 18.66 | 17.88 | 9.5 | 44.13 | 44.94 | 37.5 |
| IPO | 20.11 | 14.60 | 9.4 | 42.95 | 40.76 | 30.8 | 10.55 | 8.04 | 7.2 | 36.63 | 35.30 | 24.5 |
| χPO | 11.06 | 7.67 | 5.1 | 42.10 | 39.65 | **35.8** | 13.16 | 10.87 | 8.9 | 44.25 | 42.41 | 34.7 |
| SPPO | 26.23 | 18.12 | 11.8 | 42.01 | 39.46 | 29.5 | 16.08 | 15.52 | 9.1 | 42.64 | 39.68 | 35.9 |
| CPO | 15.07 | 16.78 | 8.3 | 35.90 | 35.20 | 26.8 | 7.01 | 6.84 | 3.0 | 36.39 | 35.40 | 22.8 |
| RRHF | 8.25 | 7.15 | 5.8 | 35.15 | 34.07 | 25.7 | 6.61 | 6.39 | 3.0 | 35.56 | 34.56 | 23.1 |
| SLiC-HF | 15.19 | 18.77 | 10.1 | 37.76 | 39.68 | 32.2 | 19.35 | 21.81 | 11.2 | 41.74 | 45.05 | 38.2 |
| ORPO | 23.99 | 16.91 | 11.2 | 43.01 | 35.68 | 27.1 | 23.20 | 19.43 | 14.7 | 45.51 | 40.95 | 33.3 |
| SimPO | 25.35 | 19.30 | 13.7 | 43.23 | 36.89 | 32.6 | 24.38 | 21.21 | 16.4 | 43.24 | 37.34 | 26.8 |
| ROPO | 21.24 | 17.66 | 9.5 | 41.03 | 36.32 | 31.5 | 22.91 | 19.67 | 10.9 | 45.55 | **45.58** | 33.7 |
| POWER-DL | **31.52** | **31.44** | **21.5** | **47.16** | **43.08** | 34.8 | **27.00** | **22.57** | **17.3** | **48.97** | 43.75 | **41.5** |
| POWER | 29.57 | 30.00 | 19.0 | 43.52 | 40.19 | 31.5 | 23.72 | 21.26 | 16.0 | 46.93 | 42.02 | 38.0 |

and Zephyr pipelines and across base and instruct settings. These improvements can largely be attributed to the integration of weighted entropy, which effectively counters underoptimization, and mitigation of reward hacking. Notably, POWER-DL surpasses other robust methods such as cDPO and ROPO demonstrating its efficacy in handling poorly covered samples. Additionally, POWER-DL improvements are more pronounced in the base setting, which is more susceptible to reward hacking due to higher levels of distribution shift. Comparing POWER-DL with POWER shows that incorporating dynamic labels further improves performance. See Appendix I and Appendix J for results on the Mistral family, MT-Bench, sample responses, and hyperparameter robustness.

**POWER-DL improves or maintains performance on downstream tasks.** One of the challenges of the alignment step is possible degradation of performance on downstream tasks, which can be attributed to reward hacking (Xu et al., 2024b). We evaluate the trained models on the LLM Leaderboard (Beeching et al., 2023), which encompass a variety of tasks, including language understanding and knowledge benchmarks MMLU (Hendrycks et al., 2020), MMLU-PRO (Wang et al., 2024d), and ARC-Challenge (Clark et al., 2018), commonsense reasoning assessments like HellaSwag (Zellers et al., 2019) and Winogrande (Sakaguchi et al., 2021), factual accuracy evaluations on TruthfulQA (Lin et al., 2022), instruction-following capabilities measured on IFEval (Zhou et al., 2023), and mathematical reasoning evaluated on the GSM8K dataset (Cobbe et al., 2021).

Tables 4 and 5 present the downstream task results. POWER-DL consistently improves or maintains performance across all tasks, effectively mitigating reward hacking. Notably, while PO methods vary in results on the GSM8K benchmark, with some like SimPO significantly degrading the initial model, POWER-DL consistently maintains or enhances performance, achieving up to a **7.0** point gain. Other tasks with notable variation include IFEval and TruthfulQA benchmarks. In TruthfulQA, POWER-DL significantly outperforms DPO, with up to a **12.8** point improvement over initial model. In the IFEval, methods like DPO and SLiC-HF sometimes degrade performance of the initial model, whereas POWER-DL consistently maintains or improves it by up to **11.7** points.

## 7 DISCUSSION

We studied reward hacking in offline preference optimization. We identified two types of reward hacking stemming from statistical fluctuations in preference data. We demonstrated that many existing methods are vulnerable to both types of reward hacking, despite maintaining a small divergence from the initial model. To mitigate reward hacking, we introduced POWER-DL, a practical algorithm based on a weighted entropy robust reward framework augmented with dynamic preference labels. POWER-DL enjoys theoretical guarantees and achieves strong empirical performance. Future research directions include applications of dynamic labels to out-of-distribution robustness and investigating the interplay between statistical errors and reward misspecification in reward hacking.

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

# A    ADDITIONAL NOTATION

We use calligraphy letters to denote sets, e.g., $\mathcal{X}, \mathcal{Y}$. Given a response $y$, we write $|y|$ to denote the length of the response in the number of tokens. We denote by $\mathcal{V}$ the vocabulary set and write $|\mathcal{V}|$ to denote the cardinality of the token space. We write $x \lesssim y$ when there exists a constant $c$ such that $x \leq cy$ and similarly, write $x \asymp y$ when there exists a constant $c$ such that $x = cy$. We write $y^1 \succ y^0$ denoting that $y^1$ is preferred over $y^0$ in the dataset. For any two discrete probability distributions $\pi$ and $\pi'$ over $\mathcal{Y}$, we define the KL divergence $D_{\mathrm{KL}}(\pi \| \pi') \coloneqq \mathbb{E}_{y \sim \pi}[\log \frac{\pi(y)}{\pi'(y)}]$. The probability simplex over a set $\mathcal{X}$ is denoted by $\Delta(\mathcal{X})$. We write $\mathbb{1}\{x = c\}$ to denote the indicator function, which is equal to 1 when $x = c$ and zero otherwise. We write $\mathbb{E}_{\mathcal{D}}$ to denote the empirical average over data.

# B    RELATED WORK

## B.1    RLHF AND PREFERENCE OPTIMIZATION

Earlier works on reinforcement learning from human preferences mainly focused on the continuous control domain (Wirth et al., 2017) such as Atari games (Christiano et al., 2017). Recently, RLHF has been extensively applied in the natural language domain (Ziegler et al., 2019) to improve alignment of LLMs with human preferences in various areas such as summarization (Stiennon et al., 2020; Wu et al., 2021), information accuracy (Menick et al., 2022), and instruction following (Ouyang et al., 2022).

Classical RLHF pipeline includes two steps of reward learning and policy optimization using RL, commonly using variants of proximal policy optimization (PPO) algorithm (Schulman et al., 2017) that involves on-policy sampling. Direct preference optimization (Rafailov et al., 2024b) simplifies the two-step process into a single-step offline optimization of the policy, reducing computational burden and training instabilities of PPO. DPO has inspired development of new preference optimization objectives from a practical perspective such as IPO (Azar et al., 2024), RRHF (Yuan et al., 2024a), SLiC-HF (Zhao et al., 2023), CPO (Xu et al., 2024a), ORPO (Hong et al., 2024), R-DPO (Park et al., 2024), SimPO (Meng et al., 2024), and general preference optimization (Tang et al., 2024) and theoretical perspective such as $\chi$PO (Huang et al., 2024) and RPO (DPO+SFT) (Liu et al., 2024). Our approach also falls under the category of offline preference optimization. We theoretically analyzed several of the mentioned methods and demonstrated theoretical benefits offered by our approach POWER-DL. We also showed that POWER-DL outperforms prior methods empirically across a variety of settings.

Going beyond the Bradley-Terry model of human preferences, some works consider general preference models (Munos et al., 2023; Swamy et al., 2024; Rosset et al., 2024; Choi et al., 2024; Wu et al., 2024b), and develop algorithms aiming at finding the Nash equilibrium. Recently Huang et al. (2024) showed that this generalization comes at a cost of an information-theoretic limit, where no statistically efficient algorithm exists to solve RLHF with general preferences under single-policy concentrability. Another line of work focuses on iterative, online improvement of language models through self-play (Chen et al., 2024b; Wu et al., 2024b; Xu et al., 2023; Yuan et al., 2024b).

## B.2    UNDERSTANDING REWARD HACKING

The phenomenon of reward hacking in training AI models has been observed in a variety of domains, ranging from games (Ibarz et al., 2018) to natural language (Paulus et al., 2018) to autonomous driving (Knox et al., 2023). In the language modeling domain, existing RLHF algorithms are observed to be susceptible to reward hacking (Gao et al., 2023; Casper et al., 2023; Amodei et al., 2016; Lambert & Calandra, 2023). Reward hacking in LLMs manifests in different ways such as verbosity (Shen et al., 2023; Singhal et al., 2023; Wang et al., 2023a), refusing to follow instructions (Röttger et al., 2024), lazy generations (Lambert & Calandra, 2023), emergence of language (Lewis et al., 2017), degradation of performance on downstream tasks such as reasoning (Xu et al., 2024b), and other problems such as hedging and self-doubt (Schulman, 2023).

**Origins of reward hacking.** In RL/RLHF, reward hacking can originate from various factors such as reward misspecification (Amodei et al., 2016; Hadfield-Menell et al., 2017; Knox et al., 2023),

diversity and inconsistencies in human preferences (Chakraborty et al., 2024), labeling noise (Wang et al., 2024a), human labeler bias (Bansal et al., 2024), and statistical errors (Liu et al., 2024). Pan et al. (2022) study reward hacking due to human misspecification of the reward model and empirically assess the impact of model size, optimization, and training on reward hacking, given synthetic misspecified reward models. Wei et al. (2024) attribute failure modes of LLM safety training to conflicts between model's capabilities and safety goals. Bansal et al. (2024) study mismatch arising from annotator bias in different types of human rating data. Peng et al. (2023) explain that variations of reward distribution across different tasks can lead to reward hacking. Rame et al. (2024) attribute reward hacking in classical RLHF to distribution shift and human preference inconsistencies. Tien et al. (2023) conduct an empirical study, revealing that non-causal distractor features, human bias and noise, and partial observability exacerbate reward misidentification.

Lambert & Calandra (2023) argue that objective mismatch in RLHF originates from learning reward model, policy training, and evaluation, and links between each pair, and suggest further research is needed to understand objective mismatch in preference optimization due to entanglement of policy and reward. Rafailov et al. (2024a) conduct an empirical study of reward hacking in direct preference optimization methods, showing that in larger KL regimes, preference optimization methods suffer from degradations reminiscent of overoptimization in RLHF. In contrast to the above works, we focus on reward hacking in offline preference optimization that originates from statistical fluctuations due to partial data coverage. Furthermore, the mentioned works conduct empirical studies, whereas here we present statistical learning theory characterizations of reward hacking.

### B.3 Reward Hacking Types and Comparison with Pessimism in Offline RL

We now highlight the differences between the setting considered in this paper and conventional offline RL, explaining usefulness of defining two types of reward hacking. In the practice of RLHF fine-tuning of LLMs, we typically have access an initial model, which already has decent performance on many downstream tasks, and a previously-collected preference data, which may not have been sampled from initial model (Xu et al., 2024b; Wang et al., 2024e). In this setting, we face two sources of distribution shift: one between the final model and data distribution, and the other between the initial model and data distribution. This setting is different from conventional offline RL, which considers access to an offline dataset (with possibly known data collection policy) and is only concerned with distribution shift between final model and data distribution (Levine et al., 2020).

Due to the existence of two sources of distribution shift, we find it useful to define Type I and Type II Reward Hacking. These definitions motivate the design of our algorithm that achieves strong empirical performance. Furthermore, our empirical results removing dynamic labels (POWER vs. POWER-DL) as well as removing the SFT term (Appendix I.4) show that the two components contribute achieving the best empirical performance. Liu et al. (2024) and Huang et al. (2024) also consider reward hacking due to partial data coverage. However, Huang et al. (2024) assume preference data are collected from the initial model, which eliminates the distribution shift between initial model and data distribution. Liu et al. (2024) propose DPO+SFT to handle the distribution shift between the final model and data distribution, yet; reward hacking due to degradation of the initial model is not considered. In this paper, we present separate analysis for POWER (Theorem 1) and dynamic labels (Theorem 2). Combining these two results into a unified analysis of POWER-DL is challenging due to extending the analysis of dynamic labels to general function approximation, which we leave for future work.

### B.4 Mitigating Reward Hacking

Various approaches have been proposed to mitigate reward hacking from applied and theoretical perspectives. Michaud et al. (2020) propose using interpretability techniques for probing whether learned rewards are aligned with human preferences. To mitigate reward hacking, several methods leverage multiple reward models. Moskovitz et al. (2024); Xu et al. (2024b) develop constrained policy optimization frameworks that leverage multiple reward models and assign weights to each of the reward models to mitigate reward hacking. Reward model ensembles (Coste et al., 2024; Zhai et al., 2023) aim to characterize uncertainty and can alleviate reward hacking; however, empirical investigations observe that they are not sufficient (Eisenstein et al., 2023). Rame et al. (2024)

propose averaging the weights of multiple trained reward models instead of ensembles to improve efficiency and performance. In contrast to these methods, our approach does not require access to or training multiple reward models. To reduce the computational costs of ensemble methods, Zhang et al. (2024) construct lightweight uncertainty estimation via linear approximation that yields a pessimistic reward model. However, such uncertainty quantification requires restrictive neural tangent kernel and approximately linear assumptions while the guarantees for our approach hold under general function approximation.

Other works use additional data to reduce reward hacking. Rita et al. (2024) leverage additional human demonstrations to calibrate the reward model and Shen et al. (2024) propose methods that leverage data augmentation to improve consistency of the reward model. RaFT (Dong et al., 2023) reduces instabilities of RLHF through iterative supervised finetuning that only keeps the highest ranked responses from a reward model. Peng et al. (2023) propose using advantage models instead of values. We propose theoretically-founded methods to mitigate reward hacking, focusing on the offline setting. Our approaches are implemented with simple modifications to objective of DPO and directly leverage previously-collected dataset; without requiring training any additional models, generating or augmenting data, or computationally expensive operation.

Among preference optimization methods, the most common approach is divergence regularization that aims at keeping the learned model close to the initial model, through metrics such as KL divergence (Rafailov et al., 2024b), $f$-divergence (Wang et al., 2024b), or Chi-squared divergence (Huang et al., 2024). $\beta$DPO (Wu et al., 2024a) calibrates $\beta$, which is the strength of divergence minimization, based on the implicit reward gap and dynamically subsamples each batch to increase robustness with respect to outliers. Other methods such as conservative DPO (Mitchell, 2024) and ROPO (Liang et al., 2024) design robust variants of DPO. We proved that many divergence-based methods still suffer from reward hacking and showed that our proposed methods outperform divergence-based and prior robust methods empirically.

## B.5 MITIGATING SPECIFIC MANIFESTATIONS OF REWARD HACKING

Several works focus on mitigating specific artifacts of reward hacking such as verbosity through various designs. Design of length-controlled winrate Alpaca-Eval (Dubois et al., 2024) aims at making the evaluation more robust against length exploitation, through estimation of controlled direct effect (VanderWeele, 2011). ODIN (Chen et al., 2024a) enforces disentanglement of preference estimation and response length by using two linear heads. In preference optimization, length-normalization (Meng et al., 2024; Grinsztajn et al., 2024; Yuan et al., 2024a) and length regularization (Park et al., 2024) are used to mitigate length exploitation.

In this paper, we consider reward hacking due to partial data coverage that can manifest in many different ways and not just length. Our weighted entropy approach provides a general framework for handling specific manifestations of reward hacking by selecting weights $w(y)$ that are smaller for undesirable response properties, such as inverse response length or preference scores. Furthermore, our approach provides a theoretically-sound way of incorporating such weights into preference optimization objective (Proposition 3), that is different from previous methods. For example, compared to SLiC-HF and SimPO, our approach results in a weight gap in the preference optimization objective and a weighted SFT term. In our practical implementation, we used inverse response length as weights, which we show in Theorem 1 prevents sample complexity to depend linearly on the response length.

Another potential benefit of our weighted entropy approach is disentangling entropy (controlled through $\beta$) and conservatism components (controlled through $\eta$ and $\gamma$). Adjusting the level of stochasticity of final learned model through $\beta$ may result in alleviating the notorious *overconfidence* challenge, in which RLHF-finetuned models become overconfident and have sharpened output probability (Leng et al., 2024; Kadavath et al., 2022). In contrast, in DPO, $\beta$ is the coefficient of the KL divergence, which impacts both pessimism and stochasticity of the learned model.

## B.6 THEORY OF RLHF AND PREFERENCE OPTIMIZATION

A series of works study theoretical foundations for RLHF and preference optimization under different settings (Zhu et al., 2023; Xiong et al., 2023; Zhu et al., 2024; Liu et al., 2024; Huang et al.,

2024; Song et al., 2024; Fisch et al., 2024). Xiong et al. (2023); Zhu et al. (2023) propose provable pessimistic offline RLHF algorithms either through confidence regions or lower confidence bounds, but are restricted to the linear family of models. Zhu et al. (2023) show that maximum entropy IRL is similar to the maximum likelihood under the Plackett-Luce models; however, entropy in maximum entropy IRL is different from our use of weighted entropy in objective (6), which goes beyond optimizing the Bradley-Terry loss. Other works that develop provable algorithms with general function approximation (Chen et al., 2022; Zhan et al., 2023a;b; Wang et al., 2023b; Li et al., 2023b) involve intractable computation. Exceptions include $\chi$PO (Huang et al., 2024) and DPO+SFT (Liu et al., 2024; Cen et al., 2024), which we have compared with POWER-DL from theoretical and empirical fronts.

## C    PROOFS FOR REWARD HACKING

### C.1    TYPE I REWARD HACKING: PROOF OF PROPOSITION 1

We first construct two multi-armed bandit (MAB) instances and then analyze each algorithm.

#### C.1.1    MAB INSTANCES FOR TYPE I REWARD HACKING

We construct a three-armed bandit problem, with true rewards $r^\star(1) = 1$, $r^\star(3) = 0$, and all actions having length one $|y| = 1$. We consider a preference data distribution that has high coverage on the high reward arms, where the probability of comparing arms 1 and 2 is $\mu_{1,2} = 1 - 1/N$ and the probability of comparing arms 1 and 3 is $\mu_{1,3} = 1/N$. In this scenario, there is a constant probability that arm 3 is compared with arm 1 exactly once. To demonstrate this, let $N(i, j)$ denote the number of comparisons between arms $i$ and $j$. We have $\mathbb{P}(N(1,3) = 1) = N(1 - \mu_{1,3})^{N-1}\mu_{1,3} = (1 - 1/N)^{N-1}$. For any $N \geq 2$, the above probability is bounded below according to

$$\mathbb{P}(N(1,3) = 1) = (1 - 1/N)^{N-1} \geq 1/e. \tag{13}$$

Conditioned on the event $N(1, 3) = 1$, there is a constant probability that arm 3 is preferred over arm 1 according to the Bradley-Terry model:

$$\mathbb{P}_{r^\star}(3 \succ 1) = \sigma(r^\star(3) - r^\star(1)) = \sigma(-1) = 1/(1 + e). \tag{14}$$

Throughout the rest of the proof, we condition on the event $\mathcal{E} = \{N(1, 3) = 1 \text{ and } 3 \succ 1\}$ which occurs with a probability of at least $1/e(1 + e)$. We further consider two special instances of the above MAB problem, with the following specifications for the initial model parameters and the reward of the second arm:

- **Instance 1:** True reward of the second arm is $r^\star(2) = 0$ and initial parameters are $\theta_0(1) = \theta_0(2) = \theta_0(3) = 1$. In this case, the best-in-class softmax policy has parameters $\theta^\star(1) = 1, \theta^\star(2) = \theta^\star(3) = 0$.
- **Instance 2:** True reward of the second arm is $r^\star(2) = 1$ and initial parameters are $\theta_0(1) = \theta_0(2) = 1, \theta_0(3) = 0$. In this case, the best-in-class softmax policy has parameters $\theta^\star(1) = \theta^\star(2) = 1, \theta^\star(3) = 0$.

We additionally consider a favorable scenario, in which an oracle reveals the best-in-class values of the first two arms $\theta^\star(1), \theta^\star(2)$. This simplifies the preference optimization objectives as it remains for the preference optimization algorithm to find $\theta(3)$.

#### C.1.2    ANALYSIS OF $\star$PO METHODS IN THE MAB INSTANCES IN SECTION C.1.1

We first record the following expression for the difference of the log probabilities of any two arms in the softmax policy class:

$$\log \pi_\theta(y) - \log \pi_\theta(y') = \log \exp(\theta(y)) - \log Z_\theta - \log \exp(\theta(y')) + \log Z_\theta$$
$$= \theta(y) - \theta(y'). \tag{15}$$

**Suboptimality of DPO.** We show that DPO fails in both instances constructed in Section C.1.1. Since parameters $\theta(1)$ and $\theta(2)$ are revealed by the oracle, the optimization problem solved by DPO

simplifies to

$$\max_{\theta(3)\in[0,1]} \frac{1}{N} \left[ \log \sigma \left( \beta \left( \log \frac{\pi_\theta(3)}{\pi_{\theta_0}(3)} - \log \frac{\pi_\theta(1)}{\pi_{\theta_0}(1)} \right) \right) \right]$$

$$= \max_{\theta(3)\in[0,1]} \log \sigma \left( \beta \left( \theta(3) - \theta_0(3) - \theta(1) - \theta_0(1) \right) \right)$$

$$= \max_{\theta(3)\in[0,1]} \log \sigma \left( \beta \left( \theta(3) - \theta_0(3) \right) \right)$$

The first equality is due to (15) and the second equality is because $\theta_0(1) = \theta(1) = 1$. For $\beta > 0$ and regardless of $\theta_0(3)$, the above function is increasing in $\theta(3)$, and thus the maximum occurs at $\theta(3) = 1$. As a result, DPO fails in both instances constructed in Section C.1.1, learning a policy with constant suboptimality:

$$\text{Instance 1:} \quad J(\pi^\star) - J(\hat{\pi}_{\text{DPO}}) = \mathbb{E}_{y\sim\pi_{\theta^\star}}[r^\star(y)] - \mathbb{E}_{y\sim\hat{\pi}_{\text{DPO}}}[r^\star(y)] = \frac{e}{2+e} - \frac{e}{1+2e} > 0.15$$

$$\text{Instance 2:} \quad J(\pi^\star) - J(\hat{\pi}_{\text{DPO}}) = \mathbb{E}_{y\sim\pi_{\theta^\star}}[r^\star(y)] - \mathbb{E}_{y\sim\hat{\pi}_{\text{DPO}}}[r^\star(y)] = \frac{2e}{1+2e} - \frac{2}{3} > 0.15$$

**Suboptimality of IPO.** We show that IPO fails in Instance 1 constructed in Section C.1.1. Leveraging uniform initialization and the logit gap expression (15), the IPO objective can be simplified as follows:

$$\min_\theta \mathbb{E}_{\mathcal{D}} \left[ \left( \log \frac{\pi_\theta(y^+)}{\pi_{\theta_0}(y^+)} - \log \frac{\pi_\theta(y^-)}{\pi_{\theta_0}(y^-)} - \frac{1}{2\tau} \right)^2 \right] = \min_\theta \mathbb{E}_{\mathcal{D}} \left[ \left( \theta(y^+) - \theta(y^-) - \frac{1}{2\tau} \right)^2 \right]$$

Since $\theta(1) = \theta(2)$ are revealed by an oracle, the IPO objective can be further simplified to

$$\min_{\theta(3)\in[0,1]} \frac{1}{N} \left( \theta(3) - \theta(1) - \frac{1}{2\tau} \right)^2 = \min_{\theta(3)\in[0,1]} \left( \theta(3) - 1 - \frac{1}{2\tau} \right)^2$$

Over the interval of $\theta(3) \in [0,1]$ and for any $\tau > 0$, this objective is decreasing in $\theta(3)$ and therefore the optimum is found at $\theta(3) = 1$. Thus, the policy found by IPO suffers from the following subpoptimality:

$$J(\pi_{\theta^\star}) - J(\hat{\pi}_{\text{IPO}}) = \mathbb{E}_{y\sim\pi_{\theta^\star}}[r^\star(y)] - \mathbb{E}_{y\sim\hat{\pi}_{\text{IPO}}}[r^\star(y)] = \frac{e}{2+e} - \frac{e}{1+2e} > 0.15$$

**Suboptimality of SimPO.** We show that SimPO suffers from Type I Reward Hacking in both instances detailed in Section C.1.1. Following the same steps as in our analysis of DPO and since $\theta(1) = \theta(2) = 1$ are assumed to be revealed by an oracle, SimPO objective simplifies to

$$\max_{\theta(3)\in[0,1]} \frac{1}{N} \log \sigma \left( \beta \left( \theta(3) - \theta(1) - \gamma \right) \right).$$

Regardless of the value of $\gamma$ and for any $\beta > 0$, the function $\log \sigma \left( \beta \left( \theta(3) - \theta(1) - \gamma \right) \right)$ is increasing in $\theta(3)$ and thus optimizing this objective over $\theta(3) \in [0,1]$ finds $\theta(3) = 1$. Therefore, policies found by SimPO in both instances in Section C.1.1 suffer from constant suboptimality:

$$\text{Instance 1:} \quad J(\pi^\star) - J(\hat{\pi}_{\text{SimPO}}) = \mathbb{E}_{y\sim\pi_{\theta^\star}}[r^\star(y)] - \mathbb{E}_{y\sim\hat{\pi}_{\text{SimPO}}}[r^\star(y)] = \frac{e}{2+e} - \frac{e}{1+2e} > 0.15$$

$$\text{Instance 2:} \quad J(\pi^\star) - J(\hat{\pi}_{\text{SimPO}}) = \mathbb{E}_{y\sim\pi_{\theta^\star}}[r^\star(y)] - \mathbb{E}_{y\sim\hat{\pi}_{\text{SimPO}}}[r^\star(y)] = \frac{2e}{1+2e} - \frac{2}{3} > 0.15$$

**Suboptimality of $\chi$PO.** We analyze $\chi$PO for Instance 2 constructed in Section C.1.1. The $\chi$PO objective is given by

$$\max_{\theta(3)\in[0,1]} \left(1 - \frac{1}{N}\right) \left[ \hat{\mu}_{1\succ2} \log \left( \sigma \left( \text{clip}_2 \left[ \beta \left( \log \frac{\pi_\theta(1)}{\pi_{\theta_0}(1)} - \log \frac{\pi_\theta(2)}{\pi_{\theta_0}(2)} + \frac{\pi_\theta(1)}{\pi_{\theta_0}(1)} - \frac{\pi_\theta(2)}{\pi_{\theta_0}(2)} \right) \right] \right) \right) \right.$$

$$+ \hat{\mu}_{2\succ1} \log \left( \sigma \left( \text{clip}_2 \left[ \beta \left( \log \frac{\pi_\theta(2)}{\pi_{\theta_0}(2)} - \log \frac{\pi_\theta(1)}{\pi_{\theta_0}(1)} + \frac{\pi_\theta(2)}{\pi_{\theta_0}(2)} - \frac{\pi_\theta(1)}{\pi_{\theta_0}(1)} \right) \right] \right) \right) \right]$$

$$+ \frac{1}{N} \left[ \log \left( \sigma \left( \text{clip}_2 \left[ \beta \left( \log \frac{\pi_\theta(3)}{\pi_{\theta_0}(3)} - \log \frac{\pi_\theta(1)}{\pi_{\theta_0}(1)} + \frac{\pi_\theta(3)}{\pi_{\theta_0}(3)} - \frac{\pi_\theta(1)}{\pi_{\theta_0}(1)} \right) \right] \right) \right) \right]$$

$$\tag{16}$$

By construction, $\theta_0(1) = \theta_0(2), \pi_{\theta_0}(1) = \pi_{\theta_0}(2)$, and $\theta(1) = \theta(2) = 1$ are known, and therefore the first two terms in (16) are equal to zero:

$$\log \frac{\pi_\theta(1)}{\pi_{\theta_0}(1)} - \log \frac{\pi_\theta(2)}{\pi_{\theta_0}(2)} + \frac{\pi_\theta(1)}{\pi_{\theta_0}(1)} - \frac{\pi_\theta(2)}{\pi_{\theta_0}(2)}$$
$$= \theta(1) - \theta_0(1) - \theta(2) + \theta_0(2) + \frac{1}{\pi_{\theta_0}(1)}\left(\frac{\exp(\theta(1))}{Z_\theta} - \frac{\exp(\theta(2))}{Z_\theta}\right) = 0.$$

Thus the maximization problem in (16) is simplified to

$$\max_{\theta(3)\in[0,1]} \log \sigma \mathsf{clip}_2 \beta \left(\log \frac{\pi_\theta(3)}{\pi_{\theta_0}(3)} - \log \frac{\pi_\theta(1)}{\pi_{\theta_0}(1)} + \frac{\pi_\theta(3)}{\pi_{\theta_0}(3)} - \frac{\pi_\theta(1)}{\pi_{\theta_0}(1)}\right)$$
$$= \max_{\theta(3)\in[0,1]} \log \sigma \mathsf{clip}_2 \beta \left(\theta(3) - \theta_0(3) - \theta(1) + \theta_0(1) + (1+2e)\left(\frac{\exp(\theta(3))}{Z_\theta} - \frac{\exp(\theta(1))}{eZ_\theta}\right)\right)$$
$$= \max_{\theta(3)\in[0,1]} \log \sigma \mathsf{clip}_2 \beta \left(\theta(3) + (1+2e)\left(\frac{\exp(\theta(3)) - 1/e}{\exp(\theta(3)) + 2e}\right)\right)$$

It is straightforward to check that the above function is strictly increasing over $\theta(3) \in [0, 1]$ and the maximization step finds $\theta(3) = 1$ when $0 < \beta \le 1/3$. This results in $\chi$PO finding the uniform policy $\theta(1) = \theta(2) = \theta(3) = 1$ and thus suffering from the following suboptimality:

$$J(\pi_{\theta^\star}) - J(\hat{\pi}_{\chi\text{PO}}) = \mathbb{E}_{y\sim\pi_{\theta^\star}}[r^\star(y)] - \mathbb{E}_{y\sim\hat{\pi}_{\chi\text{PO}}}[r^\star(y)] = \frac{2e}{2e+1} - \frac{2}{3} > 0.15.$$

## C.2 TYPE II REWARD HACKING: PROOF OF PROPOSITION 2

### C.2.1 MAB INSTANCE FOR TYPE II REWARD HACKING

We construct a three-armed bandit problem with the following true reward structure: $r^\star(1) = r^\star(2) = 0, r^\star(3) = 1$. This reward function implies the following parameters for the best-in-class policy: $\theta^\star(1) = \theta^\star(2) = 0, \theta^\star(3) = 1$, leading to the following policy:

$$\pi_{\theta^\star}(1) = \pi_{\theta^\star}(2) = \frac{1}{2+e}, \pi_{\theta^\star}(3) = \frac{e}{2+e}.$$

The performance of the above policy is $J(\pi_{\theta^\star}) = r^\star(3)\pi_{\theta^\star}(3) = e/(2+e)$. We further consider the following initialization: $\theta_0(1) = \theta_0(2) = 0, \theta_0(3) = 1$. Suppose that the comparison probabilities between the arms are $\mu_{1,2} = 1 - 1/N$ and $\mu_{1,3} = 1/N$. Following the same argument as in Section C.1.1, there is a constant probability that arm 3 is compared with arm 1 exactly once and that arm 1 is preferred to arm 3. Throughout the rest of the proof, we condition on the event $\{N(1,3) = 1, 1 \succ 3\}$. We further consider a favorable case where the optimal parameters corresponding to arms 1 and 2 are revealed by an oracle $\theta(1) = \theta(2) = 0$, leaving only $\theta(3)$ to be estimated.

### C.2.2 ANALYSIS OF $\star$PO METHODS IN THE MAB INSTANCE C.2.1

**Suboptimality of the DPO+SFT policy.** We show that DPO+SFT suffers from reward hacking for any $\eta \ge 0$, and hence the argument also shows reward hacking in DPO as a special case. The objective of DPO+SFT is given by

$$\max_\theta \mathbb{E}\left[\log \sigma\left(\beta\left(\log \frac{\pi_\theta(y^+)}{\pi_{\theta_0}(y^+)} - \log \frac{\pi_\theta(y^-)}{\pi_{\theta_0}(y^-)}\right)\right)\right] + \eta\beta\mathbb{E}\left[\log \pi_\theta(y^+)\right]$$

Since $\theta(1) = \theta(2) = 0$ are revealed by an oracle, we focus on the terms that involve $\theta(3)$ in the objective:

$$\max_{\theta(3)\in[0,1]} \frac{1}{N}\left[\log \sigma\left(\beta\left(\log \frac{\pi_\theta(1)}{\pi_{\theta_0}(1)} - \log \frac{\pi_\theta(3)}{\pi_{\theta_0}(3)}\right)\right)\right] := T_1$$
$$+ \eta\beta\left(1 - \frac{1}{N}\right)[\hat{\mu}_{1\succ2}\log \pi_\theta(1) + \hat{\mu}_{2\succ1}\log \pi_\theta(2)] + \eta\frac{1}{N}\log \pi_\theta(3) := T_2$$

Applying the softmax policy parameterization and using the fact that $\theta(1) = \theta_0(1) = 0$ and $\theta_0(3) = 1$, term $T_1$ simplifies to

$$T_1 = \frac{1}{N} \log\left(\sigma\left(\beta\left(\theta(1) - \theta_0(1) - \theta(3) + \theta_0(3)\right)\right)\right) = \frac{1}{N} \log\left(\sigma\left(\beta\left(1 - \theta(3)\right)\right)\right)$$

The term $T_2$ can also be simplified by substituting values $\theta(1) = \theta(2) = 0$:

$$T_2 = \eta\beta\left(1 - \frac{1}{N}\right)\left[\hat{\mu}_{1 \succ 2}(\theta(1) - \log Z_\theta) + \hat{\mu}_{2 \succ 1}(\theta(2) - \log Z_\theta)\right] + \eta\beta\frac{1}{N}(\theta(3) - \log Z_\theta)$$

$$= \eta\beta\left(\frac{1}{N}\theta(3) - \log Z_\theta\right)$$

$$= \eta\beta\left(\frac{1}{N}\theta(3) - \log(\exp(\theta(1)) + \exp(\theta(2)) + \exp(\theta(3)))\right)$$

$$= \eta\beta\left(\frac{1}{N}\theta(3) - \log(\exp(\theta(3)) + 2)\right)$$

Combining terms $T_1$ and $T_2$, the objective becomes

$$\max_{\theta(3) \in [0,1]} \frac{1}{N} \log\left(\sigma\left(\beta\left(1 - \theta(3)\right)\right)\right) + \eta\beta\left(\frac{1}{N}\theta(3) - \log(\exp(\theta(3)) + 2)\right)$$

We show that for any $N > 3$ the above function is decreasing in $\theta(3)$ for any $\beta, \eta$ and $\theta(3) \in [0,1]$. Since the first function $\frac{1}{N} \log\left(\sigma\left(\beta\left(1 - \theta(3)\right)\right)\right)$ is decreasing in $\theta(3)$, it is sufficient to show $\frac{1}{N}\theta(3) - \log(\exp(\theta(3)) + 2)$ is decreasing in $\theta(3)$. Derivative of this function is over $\theta(3) \in [0,1]$ and $N \geq 3$ is bounded by

$$\frac{1}{N} - \frac{\exp(\theta(3))}{\exp(\theta(3)) + 2} \leq \frac{1}{N} - \frac{1}{3} < 0.$$

Therefore, optimizing over $\theta(3) \in [0,1]$ finds $\theta(3) = 0$. This leads DPO+SFT to find a uniform policy, which suffers from a constant suboptimality:

$$J(\pi_{\theta^\star}) - J(\hat{\pi}_{\text{DPO+SFT}}) = \frac{e}{2 + e} - \frac{1}{3} > 0.2.$$

**Suboptimality of the IPO policy.** Similar to the analysis of DPO+SFT, for the IPO objective, we only focus on the terms that include $\theta(3)$:

$$\min_{\theta(3) \in [0,1]} \frac{1}{N}\left(\log\frac{\pi_\theta(1)}{\pi_{\theta_0}(1)} - \log\frac{\pi_\theta(3)}{\pi_{\theta_0}(3)} - \frac{1}{2\tau}\right)^2 = \min_{\theta(3) \in [0,1]}\left(1 - \frac{1}{2\tau} - \theta(3)\right)^2$$

Since $\tau > 0$, solution to the above minimization is

$$\theta(3) = \max\left\{0, 1 - \frac{1}{2\tau}\right\}$$

Therefore, the suboptimality of the IPO policy is given by

$$J(\pi_{\theta^\star}) - J(\hat{\pi}_{\text{IPO}}) = \frac{e}{2 + e} - \frac{\exp(\max\{0, 1 - 1/(2\tau)\})}{2 + \exp(\max\{0, 1 - 1/(2\tau)\})}$$

And for the regime with $\tau < 1$, we have $J(\pi_{\theta^\star}) - J(\hat{\pi}_{\text{IPO}}) > 0.1$.

**Suboptimality of the SimPO policy.** The objective optimized by SimPO over the term that involve $\theta(3)$ is given by

$$\max_{\theta(3) \in [0,1]} \frac{1}{N} \log\sigma\left(\beta\left(\log\pi_\theta(1) - \log\pi_\theta(3) - \gamma\right)\right)$$

$$= \max_{\theta(3) \in [0,1]} \log\sigma\left(\beta\left(-\theta(3) - \gamma\right)\right)$$

The above function is decreasing in $\theta(3)$ therefore SimPO finds $\theta(3) = 0$ and suffers from the followingg suboptimality

$$J(\pi_{\theta^\star}) - J(\hat{\pi}_{\text{SimPO}}) = \frac{e}{2+e} - \frac{1}{3} > 0.2.$$

**Suboptimality of the $\chi$PO policy.** The objective optimized by $\chi$PO is given by

$$
\max_{\theta(3)\in[0,1]} \left(1 - \frac{1}{N}\right) \left[ \hat{\mu}_{1\succ2} \log\left(\sigma\left(\text{clip}_2\left[\beta\left(\log\frac{\pi_\theta(1)}{\pi_{\theta_0}(1)} - \log\frac{\pi_\theta(2)}{\pi_{\theta_0}(2)} + \frac{\pi_\theta(1)}{\pi_{\theta_0}(1)} - \frac{\pi_\theta(2)}{\pi_{\theta_0}(2)}\right)\right]\right)\right) \right.
$$

$$
+ \hat{\mu}_{2\succ1} \log\left(\sigma\left(\text{clip}_2\left[\beta\left(\log\frac{\pi_\theta(2)}{\pi_{\theta_0}(2)} - \log\frac{\pi_\theta(1)}{\pi_{\theta_0}(1)} + \frac{\pi_\theta(2)}{\pi_{\theta_0}(2)} - \frac{\pi_\theta(1)}{\pi_{\theta_0}(1)}\right)\right]\right)\right) \right]
$$

$$
+ \frac{1}{N}\left[\log\left(\sigma\left(\text{clip}_2\left[\beta\left(\log\frac{\pi_\theta(1)}{\pi_{\theta_0}(1)} - \log\frac{\pi_\theta(3)}{\pi_{\theta_0}(3)} + \frac{\pi_\theta(1)}{\pi_{\theta_0}(1)} - \frac{\pi_\theta(3)}{\pi_{\theta_0}(3)}\right)\right]\right)\right)\right]
\tag{17}
$$

By construction, $\pi_{\theta_0}(1) = \pi_{\theta_0}(2) = 1/(2 + e)$, and $\theta(1) = \theta(2) = 0$ revealed by an oracle, and therefore the first two lines in (16) are equal to zero:

$$
\log\frac{\pi_\theta(1)}{\pi_{\theta_0}(1)} - \log\frac{\pi_\theta(2)}{\pi_{\theta_0}(2)} + \frac{\pi_\theta(1)}{\pi_{\theta_0}(1)} - \frac{\pi_\theta(2)}{\pi_{\theta_0}(2)}
$$

$$
= \theta(1) - \theta_0(1) - \theta(2) + \theta_0(2) + (2 + e)\left(\frac{\exp(\theta(1))}{Z_\theta} - \frac{\exp(\theta(2))}{Z_\theta}\right) = 0.
$$

The objective (17) can therefore be simplified to

$$
\max_{\theta(3)\in[0,1]} \frac{1}{N}\left[\log\left(\sigma\left(\text{clip}_2\left[\beta\left(\log\frac{\pi_\theta(1)}{\pi_{\theta_0}(1)} - \log\frac{\pi_\theta(3)}{\pi_{\theta_0}(3)} + \frac{\pi_\theta(1)}{\pi_{\theta_0}(1)} - \frac{\pi_\theta(3)}{\pi_{\theta_0}(3)}\right)\right]\right)\right)\right]
$$

$$
= \max_{\theta(3)\in[0,1]} \log\left(\sigma\left(\text{clip}_2\left[\beta\left(\theta(1) - \theta_0(1) - \theta(3) + \theta_0(3) + \frac{\pi_\theta(1)}{\pi_{\theta_0}(1)} - \frac{\pi_\theta(3)}{\pi_{\theta_0}(3)}\right)\right]\right)\right)
$$

$$
= \max_{\theta(3)\in[0,1]} \log\left(\sigma\left(\text{clip}_2\left[\beta\left(1 - \theta(3) + \frac{\pi_\theta(1)}{\pi_{\theta_0}(1)} - \frac{\pi_\theta(3)}{\pi_{\theta_0}(3)}\right)\right]\right)\right)
$$

$$
= \max_{\theta(3)\in[0,1]} \log\sigma\left(\text{clip}_2\left[\beta\left(1 - \theta(3) + \frac{2+e}{2+\exp(\theta(3))}(1 - \exp(\theta(3) - 1))\right)\right]\right)
$$

The last equation applies the definition of $\pi_\theta$ and substitutes values for $\theta_0(1), \theta_0(3), \theta(1)$. The function

$$
\beta\left(1 - \theta(3) + \frac{2+e}{2+\exp(\theta(3))}(1 - \exp(\theta(3) - 1))\right)
$$

is decreasing in $\theta(3)$ for $0 < \beta \leq 1$ and that for $\theta(3) \in [0,1]$, the above function remains between 0 and 2. As a result, the maximization problem leads to $\theta(3) = 0$. Thus, $\chi$PO finds the uniform policy $\hat{\pi}_{\chi\text{PO}} = \text{Unif}(\{1,2,3\})$, which suffers from a constant suboptimality:

$$J(\pi_{\theta^\star}) - J(\hat{\pi}_{\chi\text{PO}}) = \frac{e}{2+e} - \frac{1}{3} > 0.2.$$

### C.3    PROOF OF PROPOSITION 4

**Policy learned by POWER in the MAB instances in C.1.1.** We show that when faced with the Type I Reward Hacking MAB instances of C.1.1, a more general variant of POWER mitigates reward hacking. Let $g(x)$ be an increasing function with bounded derivative: $g'(x - 1) \leq B_g$ for any $x \in [0,1]$. We consider the following objective:

$$\max_\theta \mathbb{E}_{\mathcal{D}}\left[g\left(w(y^+)\log\pi_\theta(y^+|x) - w(y^-)\log\pi_\theta(y^-|x) + w(y^+) - w(y^-)\right)\right] + \eta\beta\mathbb{E}_{\mathcal{D}}\left[w(y)\log\pi_\theta(y)\right]$$

The POWER objective is a special case of the above objective with $g(\cdot) = \log\sigma(\cdot)$. Note that we have $g'(x - 1) = \sigma(-(x - 1)) \leq 1$, and therefore the bounded derivative assumption is satisfied.

For the MAB instances in C.1.1, the optimization problem simplifies to

$$\max_{\theta(3)\in[0,1]} \frac{1}{N}\left(g(\theta(3) - 1) + \eta\theta(3)\right) - \eta\log\left(\exp(\theta(3)) + e + 1\right).$$

We show that for any $\eta > \frac{B_g(2+e)}{N-(2+e)}$ the derivative of above function is negative. This is because for any $\theta(3) \in [0, 1]$, we have

$$\begin{aligned}
\frac{1}{N}(g'(\theta(3) - 1) + \eta) - \eta\frac{\exp(\theta(3))}{\exp(\theta(3)) + e + 1} &\leq \frac{1}{N}(B_g + \eta) - \frac{\eta}{2+e} \\
&= \frac{B_g}{N} - \eta\left(\frac{1}{2+e} - N\right) \\
&< \frac{B_g}{N} - \frac{B_g(2+e)}{N-(2+e)}\left(\frac{1}{2+e} - N\right) \\
&\leq 0.
\end{aligned}$$

Because the function is decreasing in $\theta(3)$ the optimum is at $\theta(3) = 0$ and thus the algorithm finds $\hat{\pi} = \pi_{\theta^\star}$. Finally, the conclusion also holds for POWER as a special case and thus $\hat{\pi}_{\text{POWER}} = \pi_{\theta^\star}$.

**Policy learned by POWER in the MAB instance in C.2.1.** The POWER objective with response lengths equal to one is given by

$$\max_{\theta} \mathbb{E}\left[\log\sigma\left(\beta\left(\log\pi_\theta(y^+) - \log\pi_\theta(y^-)\right)\right)\right] + \eta\beta\mathbb{E}\left[\log\pi_\theta(y^+)\right]$$

With a similar argument as in our analysis of DPO+SFT, we obtain the following objective:

$$\max_{\theta(3)\in[0,1]} \frac{1}{N}\log\left(\sigma\left(\beta\left(-\theta(3)\right)\right)\right) + \eta\beta\left(\frac{1}{N}\theta(3) - \log(\exp(\theta(3)) + 2)\right)$$

It is easy to check that the above function is decreasing in $\theta(3)$. Therefore, optimization leads to $\theta(3) = 0$ and the policy learned by POWER suffers from a constant suboptimality

$$J(\pi_{\theta^\star}) - J(\hat{\pi}_{\text{POWER}}) = \frac{e}{2+e} - \frac{1}{3} > 0.2.$$

# D  POWER OBJECTIVE DERIVATION

This section is organized as follows. In Section D.1 we record a useful proposition that captures properties of the optimal policy to the WER objective. This result comes in handy for deriving our preference optimization objective—which relies on the equivalence between minimax and maximin objectives as well as finding a closed-form solution for the inner maximization problem. In Section D.2, we prove that under certain regularity conditions on reward class $\mathcal{R}$, the maximization and minimization steps in objective (6) can be interchanged. With these two results at hand, we prove Proposition 5 in Section D.3, which gives the POWER objective.

## D.1  OPTIMAL POLICY FOR WEIGHTED ENTROPY REWARD MAXIMIZATION

For the WER objective, we have the following proposition which shows the uniqueness of the optimal WER policy on the support of prompt distribution, and connects this policy to the reward gap.

**Proposition 5** (**WER Policy**). *For any $\beta > 0$, reward function $r$, any $x \in \mathcal{X}$ with $\rho(x) > 0$, and any action pairs $y, y' \in \mathcal{Y}$, the policy $\pi_r$ that maximizes the WER objective (5) satisfies the following statements:*

1. *For any $\beta > 0$, policy $\pi_r$ is unique on the support of $\rho$.*

2. *Policy $\pi_r$ satisfies the following equation:*

$$r(x, y) - r(x, y') = \beta\Big(w(y)\log\pi_r(y|x) - w(y')\log\pi_r(y'|x) + (w(y) - w(y'))\Big) \quad (18)$$

*Proof of Proposition 5.* The WER objective solves the following optimization problem:

$$\max_\pi \mathbb{E}_{x\sim\rho}\left[\sum_y \pi(y|x)\left[r(x,y) - \beta w(y)\log\pi(y|x)\right]\right]$$

$$\sum_y \pi(y|x) = 1 \qquad \forall x \in \mathcal{X} \tag{19}$$

We find the optimal policy for each context in the support of $\rho$ independently. For any such $x$, we rewrite the constrained optimization problem using Lagrange multipliers:

$$\sum_y \pi(y|x)\left[r(x,y) - \beta w(y)\log\pi(y|x)\right] - \lambda_x\left(\sum_y \pi(y|x) - 1\right)$$

Notice that the above function is concave in $\pi(y|x)$ due to $w(y), \beta > 0$ and thus the solution is unique on the support of $\rho$ and the stationary point is the maximizer. Taking the derivative with respect to $\pi(y|x)$ and setting it to zero finds an equation governing the optimal policy $\pi_r$

$$r(x,y) = \beta w(y)\log\pi_r(y|x) + \beta w(y) + \lambda_x$$

Thus for any $y, y'$, one has

$$r(x,y) - r(x,y') = \beta\Big(w(y)\log\pi_r(y|x) + w(y) - w(y')\log\pi_r(y'|x) - w(y')\Big),$$

which concludes the proof. $\qquad\qquad\square$

### D.2 Minimax Objective Equivalence to Maximin Objective

In this section, we show that that the maximin objective (6) can be written as a minimax objective under certain regularity conditions. Define the following notation to denote the weighted-entropy robust reward objective for any $\pi \in \Pi$ and $r \in \mathcal{R}$:

$$\phi(\pi, r) \coloneqq L_{\mathrm{BT}}(r) + \eta\Big(\mathbb{E}_{x\sim\rho, y\sim\pi}\left[r(x,y)\right] - \mathbb{E}_{x\sim\rho, y'\sim\pi'}\left[r(x,y')\right] + \beta H_w(\pi)\Big) \tag{20}$$

We follow the approach of Liu et al. (2024) and impose regularity conditions on the class $\mathcal{R}$, which is contingent upon our definition of $\phi$, to show the maximin and minimax equivalence. Formally, we make the following assumption.

**Assumption 1** (**Regularity of the Reward Class**)**.** *We assume that class $\mathcal{R}$ satisfies the following:*

1. *The space $\mathcal{R}$ is a non-empty compact topological space;*

2. *The function $\phi$ defined in (20) is convex-like in $\mathcal{R}$; that is, for any $r_1, r_2 \in \mathcal{R}, \pi \in \Pi$, and $\alpha \in [0,1]$, there exists $r_3 \in \mathcal{R}$ such that*

$$\phi(\pi, r_3) \le \alpha\phi(\pi, r_1) + (1-\alpha)\phi(\pi, r_2).$$

The above condition is satisfied in several special cases. For example, it is satisfied when $\mathcal{R}$ is convex such as a linear class (Xiong et al., 2023; Fisch et al., 2024). As a more general case, if $\mathcal{R}$ is a Lipschitz continuous class, we can conclude function $\phi(\pi, \cdot)$ to be convex over $\mathcal{R}$ as $\phi(\pi, \cdot)$ is a sum of a linear term in $r$ and a convex term $L_{\mathrm{BT}}(r)$.

Under this assumption, we have the following proposition showing the equivalence between maximin and minimax objectives.

**Proposition 6** (**Equivalence of Maximin and Minimax Algorithms**)**.** *For the policy class $\Pi = \{\pi : \mathcal{X} \to \Delta(\mathcal{Y})\}$ and reward class $\mathcal{R}$ satisfying Assumption 1, define policy $\pi_{\hat{r}}$ to be the optimal policy corresponding to the minimax reward function, i.e.,*

$$\pi_{\hat{r}} \in \arg\max_{\pi\in\Pi}\phi(\pi, \hat{r}) \quad where \quad \hat{r} \in \arg\min_{r\in\mathcal{R}}\max_{\pi\in\Pi}\phi(\pi, r)$$

*Then, policy $\pi_{\hat{r}}$ is also the optimal solution to the maximin objective, i.e.*

$$\pi_{\hat{r}} \in \arg\max_{\pi\in\Pi}\min_{r\in\mathcal{R}}\phi(\pi, r).$$

*Proof.* We begin by recording the following lemma that shows the equivalence of the maximin and minimax objectives under the assumptions on $\mathcal{R}$. Proof of this lemma is deferred to the end of this section.

**Lemma 1** (Equivalence of Maximin and Minimax Objectives). *Given the policy class* $\Pi : \{\pi : \mathcal{X} \to \Delta(\mathcal{Y})\}$ *and reward class* $\mathcal{R}$ *satisfying Assumption 1, the following statement holds for* $\phi$ *defined in (20):*

$$\max_{\pi \in \Pi} \min_{r \in \mathcal{R}} \phi(\pi, r) = \min_{r \in \mathcal{R}} \max_{\pi \in \Pi} \phi(\pi, r).$$

Denote the policy solving the maximin problem by $\hat{\pi} \in \arg\max_{\pi \in \Pi} \min_{r \in \mathcal{R}} \phi(\pi, r)$. The duality gap of $\hat{r}, \hat{\pi}$ is given by

$$
\begin{aligned}
\text{dual}(\hat{r}, \hat{\pi}) &:= \max_{\pi \in \Pi} \phi(\pi, \hat{r}) - \min_{r \in \mathcal{R}} \phi(\hat{\pi}, r) \\
&= \max_{\pi \in \Pi} \phi(\pi, \hat{r}) - \min_{r \in \mathcal{R}} \max_{\pi \in \Pi} \phi(\pi, r) + \min_{r \in \mathcal{R}} \max_{\pi \in \Pi} \phi(\pi, r) - \min_{r \in \mathcal{R}} \phi(\hat{\pi}, r) \\
&= \max_{\pi \in \Pi} \phi(\pi, \hat{r}) - \min_{r \in \mathcal{R}} \max_{\pi \in \Pi} \phi(\pi, r) + \max_{\pi \in \Pi} \min_{r \in \mathcal{R}} \phi(\pi, r) - \min_{r \in \mathcal{R}} \phi(\hat{\pi}, r) \\
&= 0
\end{aligned}
\tag{21}
$$

In the penultimate equation, we applied Lemma 1 and the last equation uses the definition of $\hat{r}$ and $\hat{\pi}$. The duality gap is also equal to

$$\text{dual}(\hat{r}, \hat{\pi}) = \max_{\pi \in \Pi} \phi(\pi, \hat{r}) - \phi(\hat{\pi}, \hat{r}) + \phi(\hat{\pi}, \hat{r}) - \min_{r \in \mathcal{R}} \phi(\hat{\pi}, r) \tag{22}$$

Comparing (21) and (22), we conclude that $\max_{\pi} \phi(\pi, \hat{r}) = \phi(\hat{\pi}, \hat{r})$ which means $\hat{\pi} \in \arg\max_{\pi \in \Pi} \phi(\hat{r}, \pi)$. Recall that by definition, we also have $\pi_{\hat{r}} \in \arg\max_{\pi \in \Pi} \phi(\hat{r}, \pi)$. By the uniqueness of the WER optimal policy over $\rho$ as established in Proposition 5, we conclude that $\pi_{\hat{r}}(\cdot|x) = \hat{\pi}(\cdot|x)$ for any $x$ with $\rho(x) > 0$. Since $\phi(\pi, r)$ depends on $\pi$ only through its value on the support of $\rho$, we conclude that $\pi_{\hat{r}} \in \arg\max_{\pi \in \Pi} \min_{r \in \mathcal{R}} \phi(\pi, r)$, which completes the proof. $\square$

*Proof of Lemma 1.* This result relies on a minimax theorem by Fan (1953) presented in Lemma 2. We prove that all the requirements of this theorem are satisfied. First, by definition, policy class $\Pi$ is a non-empty convex set and by Assumption 1, the reward class $\mathcal{R}$ is a non-empty compact topological space. Second, function $\phi(\pi, r)$ is concave on $\Pi$ because it is a sum of a linear function in $\pi$ and (weighted) entropy of $\pi$. Lastly, by Assumption 1, function $\phi(\pi, r)$ is continuous and convex-like on $\mathcal{R}$. Therefore, we apply Lemma 2 to conclude the equivalence of maximin and minimax problems on $\phi$. $\square$

### D.3 PROOF OF PROPOSITION 3

We start by deriving the objective (8) by changing the order of maximization and minimization in the maximin objective (6), which is valid on the account of Proposition 6. Writing the minimax objective and rearranging some terms yields

$$\min_r L_{\text{BT}}(r) + \eta \max_{\pi} \left( \mathbb{E}_{x \sim \rho, y \sim \pi, y' \sim \pi'} \left[ r(x, y) - r(x, y') \right] + \beta H_w(\pi) \right) \tag{23}$$

The inner maximization problem over $\pi$ is the same as the weighted entropy reward maximization objective (5) minus a baseline term, which is independent of $\pi$.

We apply the reward gap expression provided by Proposition 5 that governs the maximizer policy $\pi_r$ as well as the definition of weighted entropy in Definition 1 to find the maximum value of the inner optimization problem:

$$
\begin{aligned}
&\max_{\pi} \mathbb{E}_{x \sim \rho, y \sim \pi, y' \sim \pi'} \left[ r(x, y) - r(x, y') \right] + \beta H_w(\pi) \\
&= \beta \mathbb{E}_{x \sim \rho, y \sim \pi, y' \sim \pi'} \left[ w(y) \log \pi_r(y|x) - w(y') \log \pi_r(y'|x) + (w(y) - w(y')) - w(y) \log \pi_r(y|x) \right] \\
&= -\beta \mathbb{E}_{x \sim \rho, y \sim \pi, y' \sim \pi'} \left[ w(y') \log \pi_r(y'|x) - (w(y) - w(y')) \right]
\end{aligned}
$$

We substitute the above expression back in the minimax objective (23):

$$\min_r L_{\text{BT}}(r) - \eta \beta \mathbb{E}_{x \sim \rho, y \sim \pi, y' \sim \pi'} \left[ w(y') \log \pi_r(y'|x) - (w(y) - w(y')) \right] \tag{24}$$

$$= \min_r L_{\text{BT}}(r) - \eta \beta \mathbb{E}_{x \sim \rho, y' \sim \pi'} \left[ w(y') \log \pi_r(y'|x) \right] \tag{25}$$

The above equation uses the fact that $(w(y) - w(y'))$ is independent of $r$. To obtain the final objective, we replace the reward gap expression from Proposition 5 in $L_{\text{BT}}(r)$, which cocludes the proof.

### D.4 AUXILIARY LEMMAS

**Lemma 2** (**Minimax Theorem;** Fan (1953)). *Let $\mathcal{X}$ be a nonempty (not necessarily topologized) set and $\mathcal{Y}$ be a nonempty compact topological space. Let $f : \mathcal{X} \times \mathcal{Y} \to \mathbb{R}$ be lower semicontinuous on $Y$. Suppose that $f$ is concave-like on $\mathcal{X}$ and convex-like on $\mathcal{Y}$, i.e., for any $x_1, x_2 \in \mathcal{X}, y \in \mathcal{Y}$, $\alpha \in [0, 1]$, there exists $x_3 \in \mathcal{X}$ such that*

$$f(x_3, y) \geq \alpha \cdot f(x_1, y) + (1 - \alpha) \cdot f(x_2, y),$$

*and for any $y_1, y_2 \in \mathcal{Y}, x \in \mathcal{X}$, $\beta \in [0, 1]$, there exists $y_3 \in Y$ such that*

$$f(x, y_3) \leq \beta \cdot f(x, y_1) + (1 - \beta) \cdot f(x, y_2).$$

*Then the following holds:*

$$\max_{x \in \mathcal{X}} \min_{y \in \mathcal{Y}} f(x, y) = \min_{y \in \mathcal{Y}} \max_{x \in \mathcal{X}} f(x, y).$$

## E FINITE-SAMPLE ANALYSIS OF POWER

This section is organized as follows. We begin by presenting the definition of single-policy concentrability for offline preference optimization, which characterizes the coverage of the competing policy in the dataset, in Section E.1. In Section E.2, we prove the finite-sample guarantees for POWER.

### E.1 SINGLE-POLICY CONCENTRABILITY IN PREFERENCE OPTIMIZATION

**Definition 2** (**Single-Policy Concentrability;** Zhan et al. (2023a)). *Given a policy $\pi$ and ground truth reward $r^\star$, the concentrability coefficient of offline data distribution $\mu$ with respect to the reward model class $\mathcal{R}$ and the baseline policy $\pi'$ is defined as*

$$C_\mu^\pi(\mathcal{R}, \pi') := \max \left\{ 0, \sup_{r \in \mathcal{R}} \frac{\mathbb{E}_{x \sim \rho, y \sim \pi, y' \sim \pi'} \left[ r^\star(x, y) - r^\star(x, y') - (r(x, y) - r(x, y')) \right]}{\sqrt{\mathbb{E}_{x, y, y' \sim \mu} \left[ (r^\star(x, y) - r^\star(x, y') - (r(x, y) - r(x, y')))^2 \right]}} \right\}. \tag{26}$$

Single-policy concentrability coefficient in offline RL quantifies the extent to which a target competing policy $\pi$ is covered by an offline data collection distribution $\mu$. In the offline RLHF setting, single-policy concentrability as defined in the work Zhan et al. (2023a) also depends on a baseline policy $\pi'$.

### E.2 PROOF OF THEOREM 1

To prove finite-sample guarantees, we use a similar argument to Liu et al. (2024), adapted to the weighted-entropy objective and combined with the bounds on weighted entropy and the special weights as inverse response lengths. Suboptimality of the learned policy $\hat{\pi} := \hat{\pi}_{\text{POWER}}$ with respect to a competing policy $\pi$ can be decomposed into three terms:

$$J(\pi) - J(\hat{\pi}) = \mathbb{E}_{x \sim \rho, y \sim \pi} \left[ r^\star(x, y) \right] - \mathbb{E}_{x \sim \rho, y \sim \hat{\pi}} \left[ r^\star(x, y) \right] = T_1 + T_2 + T_3.$$

where $T_1$ is defined as

$$\begin{aligned}
T_1 := \; &\mathbb{E}_{x \sim \rho, y \sim \pi, y' \sim \pi'} \left[ r^\star(x, y) - r^\star(x, y') - \beta H_w(\pi) \right] \\
&- \eta^{-1} \min_{r \in \mathcal{R}} \left\{ \eta \mathbb{E}_{x \sim \rho, y \sim \hat{\pi}, y' \sim \pi'} \left[ r(x, y) - r(x, y') - \beta H_w(\pi) \right] + L_{\text{BT}}(r) \right\},
\end{aligned} \tag{27}$$

$T_2$ is defined as

$$\begin{aligned}
T_2 := \; &\eta^{-1} \min_{r \in \mathcal{R}} \left\{ \eta \mathbb{E}_{x \sim \rho, y \sim \hat{\pi}, y' \sim \pi'} \left[ r(x, y) - r(x, y') - \beta H_w(\pi) \right] + L_{\text{BT}}(r) \right\} \\
&- \mathbb{E}_{x \sim \rho, y \sim \hat{\pi}, y' \sim \pi'} \left[ r^\star(x, y) - r^\star(x, y') - \beta H_w(\pi) \right],
\end{aligned} \tag{28}$$

and $T_3$ is defined as

$$T_3 := \beta \left[ H_w(\pi) - H_w(\hat\pi) \right]. \tag{29}$$

We will prove in the subsequent section that for weighted entropy $H_w(\pi)$ with general weights, terms $T_1 + T_2$ and $T_3$ are bounded according to:

$$T_1 + T_2 \lesssim \frac{(C_\mu^\pi(\mathcal{R}, \pi') + 1)\tilde{R}^2 \iota}{\sqrt{N}}, \tag{30}$$

$$T_3 \lesssim \frac{H_w(\pi)}{\sqrt{N}}. \tag{31}$$

Summing the above bounds, we conclude the first claim:

$$J(\pi) - J(\hat\pi) \lesssim \frac{1}{\sqrt{N}} \left( \left( [C_\mu^\pi(\mathcal{R}, \pi')]^2 + 1 \right) \tilde{R}^2 \iota + H_w(\pi) \right).$$

Furthermore, in the special case of $w(y) = 1/|y|$, we have the following bound on $T_3$:

$$T_3 \lesssim \frac{\log|\mathcal{V}|}{\sqrt{N}}, \tag{32}$$

The above bound combined with (30) leads to the following rate

$$J(\pi) - J(\hat\pi) \lesssim \frac{1}{\sqrt{N}} \left( \left( [C_\mu^\pi(\mathcal{R}, \pi')]^2 + 1 \right) \tilde{R}^2 \iota + \log|\mathcal{V}| \right).$$

### E.2.1  PROOF OF THE BOUND (30) ON $T_1 + T_2$

**Bounding $T_1$.** $\hat\pi$ is the maximizer to the following objective

$$\hat\pi \in \arg\max_{\pi \in \Pi} \min_{r \in \mathcal{R}} \eta \mathbb{E}_{x \sim \rho, y \sim \hat\pi, y' \sim \pi'} \left[ r(x, y) - r(x, y') - \beta H_w(\pi) \right] + L_{\mathrm{BT}}(r)$$

We use this fact to bound the term $T_1$ according to

$$T_1 \le \mathbb{E}_{x \sim \rho, y \sim \pi, y' \sim \pi'} \left[ r^\star(x, y) - r^\star(x, y') - \beta H_w(\pi) \right]$$
$$- \eta^{-1} \min_{r \in \mathcal{R}} \left\{ \eta \mathbb{E}_{x \sim \rho, y \sim \pi, y' \sim \pi'} \left[ r(x, y) - r(x, y') - \beta H_w(\pi) \right] + L_{\mathrm{BT}}(r) \right\},$$
$$= \max_{r \in \mathcal{R}} \left\{ \mathbb{E}_{x \sim \rho, y \sim \pi, y' \sim \pi'} \left[ r^\star(x, y) - r^\star(x, y') - (r(x, y) - r(x, y')) \right] - \eta^{-1} L_{\mathrm{BT}}(r) \right\} \tag{33}$$

**Bounding $T_2$.** By realizability of the true reward function $r^\star \in \mathcal{R}$ we bound the term $T_2$:

$$T_2 \le \mathbb{E}_{x \sim \rho, y \sim \pi, y' \sim \pi'} \left[ r^\star(x, y) - r^\star(x, y') - \beta H_w(\pi) \right] + \eta^{-1} L_{\mathrm{BT}}(r^\star)$$
$$- \mathbb{E}_{x \sim \rho, y \sim \hat\pi, y' \sim \pi'} \left[ r^\star(x, y) - r^\star(x, y') - \beta H_w(\pi) \right]$$
$$= \eta^{-1} L_{\mathrm{BT}}(r^\star) \tag{34}$$

**Bounding $T_1 + T_2$.** Combing the bound (33) on $T_1$ and to bound (34) on $T_2$, it remains the bound the following:

$$T_1 + T_2 \le \max_{r \in \mathcal{R}} \left\{ \mathbb{E}_{x \sim \rho, y \sim \pi, y' \sim \pi'} \left[ r^\star(x, y) - r^\star(x, y') - (r(x, y) - r(x, y')) \right] := T_{1,1} \right.$$
$$\left. + \eta^{-1} \left( L_{\mathrm{BT}}(r^\star) - L_{\mathrm{BT}}(r) \right) := T_{1,2} \right\} \tag{35}$$

Define the following notation:

$$\Delta_r := \sqrt{ \mathbb{E}_{x, y, y' \sim \mu} \left[ (r^\star(x, y) - r^\star(x, y')) - (r(x, y) - r(x, y'))^2 \right] } \tag{36}$$

Term $T_{1,1}$ is directly bounded by $C_\mu^\pi(\mathcal{R}, \pi')\Delta_r$ based on the Definition 2 of single-policy concentrability, and we subsequently prove a bound on $T_{1,2}$ according to:

$$T_{1,1} \le C_\mu^\pi(\mathcal{R}, \pi')\Delta_r \tag{37}$$

$$T_{1,2} \le -2\frac{\Delta_r^2}{\eta\tilde{R}^2} + \frac{3\iota}{\eta N}, \tag{38}$$

where $\tilde{R} = 1 + \exp(R)$ and $\iota = \sqrt{\log(\mathcal{N}_\epsilon)/\delta}$. Adding the bounds on $T_{1,1}$ and $T_{1,2}$ and taking the maximum over $r$, the bound on $T_1 + T_2$:

$$T_1 + T_2 \leq \max_r \left\{ C_\mu^\pi(\mathcal{R}, \pi') \Delta_r - 2\frac{\Delta_r^2}{\eta \tilde{R}^2} \right\} + \frac{3\iota}{\eta N} \tag{39}$$

$$\leq \frac{\left[ C_\mu^\pi(\mathcal{R}, \pi') \right]^2 \eta \tilde{R}^2}{8} + \frac{3\iota}{\eta N} \tag{40}$$

The last inequality uses the fact that $az - bz^2 \leq a^2/4b$ for any $z \in \mathbb{R}$. By the choice of $\eta = \sqrt{6}\iota/(\tilde{R}^2\sqrt{N})$, the above bound becomes:

$$T_1 + T_2 \lesssim \frac{(\left[ C_\mu^\pi(\mathcal{R}, \pi') \right]^2 + 1)\tilde{R}^2\iota}{\sqrt{N}}.$$

*Proof of the bound (38) on $T_{1,2}$.* In the view of the uniform concentration result in Liu et al. (2024, Lemma A.1), with probability at least $1 - \delta$ setting $\epsilon = (6\tilde{R}N)^{-1}$, the following bound holds for any $r \in \mathcal{R}$

$$L_{\text{BT}}(r^\star) - L_{\text{BT}}(r) \leq -2\mathbb{E}_{x,y,y'\sim\mu} \left[ D_{\text{Hellinger}}^2 \left( \mathbb{P}_{r^\star}(\cdot|x,y,y') \| \mathbb{P}_r(\cdot|x,y,y') \right) \right] + \frac{3\iota}{N}, \tag{41}$$

where $\mathbb{P}_r(\cdot|x,y,y')$ is the Bradley-Terry preference probability given a reward model $r$ as defined in (2). The Hellinger distance can be bounded by total variation (TV) distance according to

$$\begin{aligned}
&D_{\text{Hellinger}}^2 \left( \mathbb{P}_{r^\star}(\cdot|x,y,y') \| \mathbb{P}_r(\cdot|x,y,y') \right) \\
&\geq D_{\text{TV}}^2 \left( \mathbb{P}_{r^\star}(\cdot|x,y,y') \| \mathbb{P}_r(\cdot|x,y,y') \right) \\
&= \frac{1}{2} \left| \sigma(r^\star(x,y) - r^\star(x,y')) - \sigma(r(x,y) - r(x,y')) \right| \\
&\quad + \frac{1}{2} \left| \sigma(r^\star(x,y') - r^\star(x,y)) - \sigma(r(x,y') - r(x,y)) \right| \\
&= \left| \sigma(r^\star(x,y) - r^\star(x,y')) - \sigma(r(x,y) - r(x,y')) \right| \\
&\geq \frac{1}{\tilde{R}^2} \left| (r^\star(x,y) - r^\star(x,y')) - (r(x,y) - r(x,y')) \right|
\end{aligned} \tag{42}$$

The penultimate equation uses the fact that $\sigma(-x) = 1 - \sigma(x)$ and the last inequality is due to bi-Lipschitz continuity of the sigmoid function over $[-R, R]$; see e.g., Liu et al. (2024, Lemma A.2). Applying the bound in (42) to (41), we have

$$L_{\text{BT}}(r^\star) - L_{\text{BT}}(r) \leq -2\mathbb{E}_{x,y,y'\sim\mu} \left[ \left| (r^\star(x,y) - r^\star(x,y')) - (r(x,y) - r(x,y')) \right|^2 \right] + \frac{3\iota}{N}$$

$$= -\frac{2\Delta_r}{\tilde{R}^2} + \frac{3\iota}{N}.$$

where the last equation uses the definition of $\Delta_r$ provided in (36), completing the proof. $\square$

### E.2.2 PROOF OF THE BOUNDS (31) AND (32) ON $T_3$

In this section, we prove the bounds on $T_3$ as delineated in inequalities (31) and (32) through bounding weighted entropy. The key bounds are encapsulated in the following lemma which asserts that weighted entropy is non-negative and for special case of weights $w(y) = 1/|y|$ it can be bounded from above. The proof of this lemma is presented at the end of this section.

**Lemma 3** (Bounds on Weighted Entropy). *For any weight function $w(y) \geq 0$ and any probability distribution $p(y)$, the weighted entropy satisfies $H_w(p) \geq 0$. Furthermore, when weights are assigned according to $w(y) = 1/|y|$, with $|y|$ denoting the response length, the weighted entropy is bounded by $H_w(p) \leq \log|\mathcal{V}|$, where $|\mathcal{V}|$ is the size of the vocabulary.*

Based on Lemma 3, weighted entropy is nonnegative. Setting $\beta \asymp 1/\sqrt{N}$ immediately gives the bound (31) on $T_3$:

$$T_3 = \beta\left[H_w(\pi) - H_w(\hat{\pi})\right] \leq \frac{H_w(\pi)}{\sqrt{N}} \tag{43}$$

Moreover, when $w(y) = 1/|y|$ by Lemma 3, we have

$$T_3 \leq \frac{H_w(\pi)}{\sqrt{N}} \leq \frac{\log|\mathcal{V}|}{\sqrt{N}}.$$

*Proof of Lemma 3.* First consider the case for any general non-negative weight function $w(y) > 0$. This ensures that the weighted entropy is non-negative because:

$$H_w(p) = -\sum_y w(y)p(y)\log p(y) = \sum_y w(y)p(y)\log\frac{1}{p(y)} \geq 0.$$

Next, we provide an upper bound on the weighted entropy when $w(y) = 1/|y|$. Define $z$ to be a random variable denoting the length of a response. The weighted entropy can be decomposed as follows

$$-\sum_y \frac{1}{|y|}p(y)\log p(y) = -\sum_y \frac{1}{|y|}p_z(z=|y|)p(y \mid |y|=z)\log p_z(z=|y|)p(y \mid |y|=z)$$

$$= -\sum_y \frac{1}{|y|}p_z(z=|y|)p(y \mid |y|=z)\Big[\log p_z(z=|y|) + \log p(y \mid |y|=z)\Big]$$

$$= -\sum_z \frac{1}{|y|}p_z(z=|y|)\sum_{y \text{ s.t. } |y|=z} p(y \mid |y|=z)\log p(y \mid |y|=z) := T_{3,1}$$

$$- \sum_z \frac{1}{|y|}p_z(z=|y|)\log p_z(z=|y|)\sum_{y \text{ s.t. } |y|=z} p(y \mid |y|=z) := T_{3,2}$$

For the term $T_{3,1}$, we have

$$T_{3,1} = \sum_z p_z(z=|y|)\frac{1}{|y|} \cdot -\sum_{y \text{ s.t. } |y|=z} p_{y|z}(y \mid z=|y|)\log p_{y|z}(y \mid z=|y|)$$

The sum $-\sum_y p_{y|z}(y \mid z=|y|)\log p_{y|z}(y \mid z=|y|)$ is the Shannon entropy of a conditional distribution, which reaches its maximum when the distribution is uniform. Consequently, the maximum of this conditional entropy for a fixed length $|y|$ is given by $\log|\mathcal{V}|^{|y|} = |y|\log|\mathcal{V}|$. Substituting this bound back to $T_1$ gives:

$$T_{3,1} \leq \sum_z p_z(z=|y|)\frac{1}{|y|}|y|\log|\mathcal{V}| = \log|\mathcal{V}|$$

For the term $T_{3,2}$, first note that $\sum_{y||y|=z} p(y \mid |y|=z) = 1$. Therefore, with the maximum response length denoted by $L$ and since $|y| \geq 1$, we have

$$T_{3,2} = \sum_z \frac{1}{|y|}p_z(z=|y|)\log p_z(z=|y|) \leq \sum_z p_z(z=|y|)\log p_z(z=|y|) \leq \log L.$$

Combining the bounds on $T_{3,1}$ and $T_{3,2}$, the upper bound on weighted entropy when using inverse response length as weights is $\log|\mathcal{V}| + \log L$, which concludes the proof. $\qquad\square$

## F  DERIVATIONS AND PROOFS FOR DYNAMIC LABELS

### F.1  DERIVATION OF THE LEARNING DYNAMICS

In this section we compute the learning dynamics of POWER over preference dataset $\mathcal{D} = \{(x, y^0, y^1, l)\}$ with the label notation defined in 2.1. Here, $l = 0$ indicates that $y^0$ was preferred and

$l = 1$ indicates that $l = 1$ was preferred. With this notation, the POWER objective from Proposition (3) is given by

$$
\max_{\pi} \mathbb{E}_{\mathcal{D}} \Bigg[ l \log \sigma \bigg( \beta \Big[ w(y^1) \log \pi_\theta(y^1|x) - w(y^0) \log \pi_\theta(y^0|x) + \Big( w(y^1) - w(y^0) \Big) \Big] \bigg)
$$
$$
(1 - l) \log \sigma \bigg( \beta \Big[ w(y^0) \log \pi_\theta(y^0|x) - w(y^1) \log \pi_\theta(y^1|x) + \Big( w(y^0) - w(y^1) \Big) \Big] \bigg) \Bigg]
$$
$$
+ \eta \beta \mathbb{E}_{\mathcal{D}} \Big[ l w(y^1) \log \pi_\theta(y^1|x) + (1 - l) w(y^0) \log \pi_\theta(y^0|x) \Big]
$$

To simplify presentation, we consider the softmax MAB setting with $\beta = 1, w(y) = 1, \eta = 0$. In this case, the objective simplifies to

$$
\min_\theta - \mathbb{E}_{\mathcal{D}} \Big[ l \log \sigma(\log \pi_\theta(y^1) - \log \pi_\theta(y^0)) + (1 - l) \log \sigma(\log \pi_\theta(y^0) - \log \pi_\theta(y^1)) \Big]
$$
$$
= \min_\theta - \mathbb{E}_{\mathcal{D}} \Big[ l \log \sigma(\log \exp(\theta(y^1)) - \log Z_\theta - \log \exp(\theta(y^0)) + \log Z_\theta)
$$
$$
+ (1 - l) \log \sigma(\log \exp(\theta(y^0)) - \log Z_\theta - \log \exp(\theta(y^1)) + \log Z_\theta) \Big]
$$
$$
= \min_\theta - \mathbb{E}_{\mathcal{D}} \Big[ l \log \sigma(\theta(y^1) - \theta(y^0)) + (1 - l) \log \sigma(\theta(y^0) - \theta(y^1)) \Big].
$$

To understand the updates to the model parameters, consider the empirical probabilities derived from dataset comparisons: $\hat\mu_{0,1}$ is the empirical probability of comparing $y^0$ and $y^1$, and $\hat\mu_{1 \succ 0}$ is the empirical probability of preferring $y^1$ over $y^0$, conditioned on their comparison. Isolate the updates to the parameters $\theta(y^1)$ and $\theta(y^0)$ based on comparisons between $y^0$ and $y^1$ and through batch gradient descent. We allow the preference labels $l_t$ to change across gradient steps and thus updates to the parameter gap at step $t$ and with a learning rate $\alpha$ is given by

$$
\theta_{t+1}(y^1) - \theta_{t+1}(y^0)
$$
$$
= \theta_t(y^1) - \theta_t(y^0) + \alpha \hat\mu_{0,1} \Bigg[ \Big( \hat\mu_{1 \succ 0} l_t + \hat\mu_{0 \succ 1}(1 - l_t) \Big) \sigma(\theta_t(y^0) - \theta_t(y^1))
$$
$$
- \Big( \hat\mu_{0 \succ 1} l_t + \hat\mu_{1 \succ 0}(1 - l_t) \Big) \sigma(\theta_t(y^1) - \theta_t(y^0)) \Bigg].
$$

We used the fact that $\partial(\log \sigma(x))/\partial x = \sigma(-x)$. Since $\sigma(x) = 1 - \sigma(-x)$ and $\hat\mu_{1 \succ 0} + \hat\mu_{0 \succ 1} = 1$, the learning dynamics simplify to:

$$
\theta_{t+1}(y^1) - \theta_{t+1}(y^0)
$$
$$
= \theta_t(y^1) - \theta_t(y^0) + \alpha \hat\mu_{0,1} \Big[ (\hat\mu_{1 \succ 0} - \hat\mu_{0 \succ 1}) l - \Big( \sigma \big( \theta_t(y^1) - \theta_t(y^0) \big) - \hat\mu_{0 \succ 1} \Big) \Big].
$$

### F.2    Proof of Theorem 2

The proof is organized as follows. We start by establishing a lower bound on the dynamic labels. We subsequently use this lower bound to prove bounds on the parameter gap in low-coverage and high-coverage cases separately.

**Lower bound on dynamic labels.** The dynamics of labels are described by the following equation:

$$
\dot{l}_t = \gamma \Big( \frac{\sigma(d_t) - (1 - \hat\mu_{1 \succ 0})}{\hat\mu_{1 \succ 0} - (1 - \hat\mu_{1 \succ 0})} - l_t \Big) \tag{44}
$$

Without loss of generality, we assumed that $\hat\mu_{1 \succ 0} > 1/2$. This condition is easily met by appropriately ordering the responses. Define:

$$
\kappa := \frac{1 - \hat\mu_{1 \succ 0}}{\hat\mu_{1 \succ 0} - (1 - \hat\mu_{1 \succ 0})} \tag{45}
$$

Given that $0 \leq \sigma(d_t) \leq 1$ and the assumption $\hat{\mu}_{1 \succ 0} - (1 - \hat{\mu}_{1 \succ 0}) = 2\hat{\mu}_{1 \succ 0} - 1 > 0$, we find the following lower bound on $\dot{l}_t$:

$$\dot{l}_t = \gamma \left( \frac{\sigma(d_t) - (1 - \hat{\mu}_{1 \succ 0})}{\hat{\mu}_{1 \succ 0} - (1 - \hat{\mu}_{1 \succ 0})} - l_t \right) \geq \gamma \left( \frac{-(1 - \hat{\mu}_{1 \succ 0})}{\hat{\mu}_{1 \succ 0} - (1 - \hat{\mu}_{1 \succ 0})} - l_t \right) = \gamma \left( -\kappa - l_t \right)$$

The subsequent lemma establishes a lower bound on $l_t$ using Grönwall's inequality, with its proof provided at the end of this section.

**Lemma 4.** *Suppose that $l_t$ satisfies the following inequality $\dot{l}_t \geq \gamma(-\kappa - l_t)$ with initial value $l_0 = 1$. Then, we have the following lower bound $l_t \geq -\kappa + (\kappa + 1) \exp(-\gamma t)$.*

We proceed by separately analyzing the scenarios of low coverage and high coverage.

**Low coverage case.** The coupled dynamical system in (12) satisfies the following equation:

$$\frac{\gamma}{2\hat{\mu}_{1 \succ 0} - 1} \dot{d}_t + \alpha \hat{\mu}_{0,1} \dot{l}_t = 0$$

Upon integrating the equation above and considering the initial condition $l_0 = 1$, it follows that

$$\frac{\gamma}{2\hat{\mu}_{1 \succ 0} - 1} (d_t - d_0) = \alpha \hat{\mu}_{0,1} (1 - l_t)$$

Applying the lower bound from Lemma 4 yields:

$$\frac{\gamma}{2\hat{\mu}_{1 \succ 0} - 1} (d_t - d_0) = \alpha \hat{\mu}_{0,1} (1 - l_t)$$
$$\leq \alpha \hat{\mu}_{0,1}(1 + \kappa - (1 + \kappa) \exp(-\gamma t))$$

Consequently, we conclude that

$$|d_t - d_0| \leq \frac{\alpha \hat{\mu}_{0,1}(2\hat{\mu}_{1 \succ 0} - 1)}{\gamma} \left( 1 + \kappa - (1 + \kappa) \exp(-\gamma t) \right) \leq \frac{\alpha \hat{\mu}_{0,1} \hat{\mu}_{1 \succ 0}}{\gamma} \leq \frac{\alpha \mu_l}{\gamma} \leq \epsilon_l,$$

where we used the definition of $\kappa$ and the fact that by assumption $\alpha \mu_l / \epsilon_l \leq \gamma$.

**High coverage case.** We extend the argument by Zhu et al. (2024) for a general initialization $d_0$ and establish the final convergence rate for proper choices of hyperparameters. Consider a Lyapunov function $V_t = (\sigma(d_t) - \hat{\mu}_{1 \succ 0})^2$. Derivative of $V_t$ is given by

$$\dot{V}_t = 2 \left( \sigma(d_t) - \hat{\mu}_{1 \succ 0} \right) \sigma(d_t) \sigma(-d_t) \dot{d}_t$$
$$= 2\alpha \hat{\mu}_{0,1} \left( \sigma(d_t) - \hat{\mu}_{1 \succ 0} \right) \sigma(d_t) \sigma(-d_t) \left( (2\hat{\mu}_{1 \succ 0} - 1)l_t + 1 - \hat{\mu}_{1 \succ 0} - \sigma(d_t) \right)$$
$$= -2\alpha \hat{\mu}_{0,1} \sigma(d_t) \sigma(-d_t) \left( \sigma(d_t) - \hat{\mu}_{1 \succ 0} \right)^2 + 2\alpha n \sigma(d_t) \sigma(-d_t) \left( \sigma(d_t) - \hat{\mu}_{1 \succ 0} \right) (2\hat{\mu}_{1 \succ 0} - 1)(l_t - 1)$$
$$= 2\alpha \hat{\mu}_{0,1} \sigma(d_t) \sigma(-d_t) \left( V_t - (\sigma(d_t) - \hat{\mu}_{1 \succ 0})(2\hat{\mu}_{1 \succ 0} - 1)(l_t - 1) \right)$$

Let $\epsilon = \epsilon_0 \alpha \hat{\mu}_{0,1} T \geq \gamma T$. We find an upper bound on $\dot{V}_t$ by applying the bound on $l_t$ given in Lemma 4 and using the fact that $\hat{\mu}_{1 \succ 0}, \sigma \in [0, 1]$:

$$\dot{V}_t \leq \alpha \hat{\mu}_{0,1} \sigma(d_t) \sigma(-d_t) \left( -V_t + (\kappa + 1)(1 - \exp(-\gamma t)) \right) \tag{46}$$
$$\leq \alpha \hat{\mu}_{0,1} \sigma(d_t) \sigma(-d_t) \left( -V_t + (\kappa + 1)(1 - \exp(-\epsilon)) \right) \tag{47}$$

Now consider two cases. The first case is that for any $0 \leq t \leq T$, we have $V_t \geq 2(\kappa + 1)(1 - \exp(-\epsilon))$. In such a scenario, $V_t$ is a non-increasing function because

$$\dot{V}_t \leq 2\alpha \hat{\mu}_{0,1} \sigma(d_t) \sigma(-d_t) \left( -V_t + (\kappa + 1)(1 - \exp(-\epsilon)) \right) \leq -\alpha \hat{\mu}_{0,1} \sigma(d_t) \sigma(-d_t) V_t \leq 0. \tag{48}$$

Next, we analyze the term $\sigma(d_t) \sigma(-d_t)$. We establish a bound on $\sigma(d_t) \sigma(-d_t)$ for the case of $\sigma(d_0) \leq \hat{\mu}_{1 \succ 0}$; the case of $\sigma(d_0) \geq \hat{\mu}_{1 \succ 0}$ can be proved with a similar argument.

We prove that when $\sigma(d_0) \leq \hat{\mu}_{1 \succ 0}$, then we must have $\sigma(d_t) \leq \hat{\mu}_{1 \succ 0}$ for any $t$. Assume otherwise that there exists some $t_0$ where $\sigma(d_{t_0}) > \hat{\mu}_{1 \succ 0}$. By continuity of $\sigma(d_t)$ and since $\sigma(d_0) \leq \hat{\mu}_{1 \succ 0}$, there exists $t_1 \leq t_0$ such that $\sigma(d_{t_1}) = \hat{\mu}_{1 \succ 0}$. However, this results in $V_{t_1} = 0 < V_{t_0}$, which contradicts the fact that $V_t$ is non-increasing. Therefore, we have $\sigma(d_t) \leq \hat{\mu}_{1 \succ 0}$. Moreover, since $V_t$ is non-increasing, we also have $\sigma(d_0) \leq \sigma(d_t)$. Thus, we have $\sigma(d_0) \leq \sigma(d_t) \leq \hat{\mu}_{1 \succ 0}$ and we use this fact to find the following bound:

$$\sigma(d_t)\sigma(-d_t) = \sigma(d_t)(1 - \sigma(d_t)) \geq \min\left\{\sigma(d_0)(1 - \sigma(d_0)), \hat{\mu}_{1 \succ 0}(1 - \hat{\mu}_{1 \succ 0})\right\} = c. \quad (49)$$

Applying bound (49) to (48) and integrating over $t$ on both sides, we have

$$V_t \leq \exp(-c\alpha\hat{\mu}_{0,1}t)V_0 \leq \exp(-c\alpha\hat{\mu}_{0,1}t). \quad (50)$$

We now consider the case where for some $t_0$, we have $V_{t_0} < 2(\kappa + 1)(1 - \exp(-\epsilon))$. We show that in this case we must have $V_T \leq 2(\kappa + 1)(1 - \exp(-\epsilon))$. Assume otherwise that $V_T > 2(\kappa + 1)(1 - \exp(-\epsilon))$, then by continuity there must be $t_1$ such that $V_{t_1} = 2(\kappa + 1)(1 - \exp(-\epsilon))$. However, inequality (48) implies that for any $t \in [t_1, T]$, $V_t$ is non-increasing, leading to $V_T \leq 2(\kappa + 1)(1 - \exp(-\epsilon))$, which contradicts our assumption. Therefore, we know that

$$V_T \leq 2(\kappa + 1)(1 - \exp(-\epsilon)) \quad (51)$$

Combining the two bounds (50) and (51) on $V_T$, we have:

$$V_T \leq \max\left\{2(\kappa + 1)(1 - \exp(-\epsilon)), \exp(-c\alpha\hat{\mu}_{0,1}T)\right\}. \quad (52)$$

Subsequently, we show that $2(\kappa+1)(1-\exp(-\epsilon)) \leq \exp(-c\alpha\hat{\mu}_{0,1}T)$ provided that $\epsilon_0 \leq \frac{\exp(-1/4)}{2(\kappa+1)}$. Utilizing the inequality $\exp(-x) \geq 1 - x$, we have

$$\exp(-\epsilon_0) \geq 1 - \epsilon_0 \geq 1 - \frac{\exp(-1/4)}{2(\kappa + 1)}\exp(-1/4)$$

Using the fact that $c \leq 1/4, \kappa \geq 0$ and that $\hat{\mu}_{0,1}\alpha T \geq 1$, we conclude

$$\begin{aligned}
2(\kappa + 1) &\leq 2(\kappa + 1)\exp(-\epsilon_0) + \exp(-1/4) \\
&\leq 2(\kappa + 1)\exp(-\epsilon_0) + \exp(-c) \\
&\leq 2(\kappa + 1)\exp(-\epsilon_0\alpha\hat{\mu}_{0,1}T) + \exp(-c\alpha\hat{\mu}_{0,1}T) \\
&\leq 2(\kappa + 1)\exp(-\epsilon_0\alpha\hat{\mu}_{0,1}T) + \exp(-c\alpha\hat{\mu}_{0,1}T).
\end{aligned}$$

This leads to the bound $V_T \leq \exp(-c\alpha\hat{\mu}_{0,1}T)$.

*Proof of Lemma 4.* The result can be shown using Grönwall's inequality. Rewriting the inequality as $\dot{l}_t + \gamma l_t \geq -\gamma\kappa$ and multiplying both sides by $\exp(\gamma t)$ gives:

$$\exp(\gamma t)\dot{l}_t + \gamma\exp(\gamma t)l_t \geq -\gamma\kappa\exp(\gamma t).$$

Notice that the left-hand side is the derivative of $l_t \exp(\gamma t)$. This yields

$$\frac{\partial}{\partial t}(l_t \exp(\gamma t)) \geq -\gamma\kappa\exp(\gamma t) \Rightarrow \int_0^t \frac{\partial}{\partial t}(l_t \exp(\gamma t)) \geq \int_0^t -\gamma\kappa\exp(\gamma t). \quad (53)$$

The left-hand side and using the fact that $l_0 = 1$ is given by

$$\int_0^t \frac{\partial}{\partial t}(l_t \exp(\gamma t)) = l_t \exp(\gamma t) - l_0 \exp(\gamma \cdot 0) = l_t \exp(\gamma t) - 1.$$

The right-hand side is

$$\int_0^t -\gamma\kappa\exp(\gamma t) = -\kappa(\exp(\gamma t) - 1). \quad (54)$$

Substituting equations (54) and (54) in (53), we have

$$l_t \exp(\gamma t) - 1 \geq -\kappa(\exp(\gamma t) - 1) \Rightarrow l_t \exp(\gamma t) \geq -\kappa\exp(\gamma t) + \kappa + 1. \quad (55)$$

Multiplying both sides with $\exp(-\gamma t)$ gives the final lower bound on $l_t$. $\qquad\square$

## G    POWER-DL Pseudocode

---

**Algorithm 1** POWER with Dynamic Labels

---

1: **Inputs:** Dataset $\mathcal{D} = \{(x_i, y_i^+, y_i^-)\}_{i=1}^n$, hyperparameters $\eta, \beta, \gamma$, learning rate $\alpha$, weight function $w(y)$ (e.g., $w(y) = 1/|y|$), initial model $\pi_{\theta_0}$
2: Initialize $l_i^0 = 0$ for all $(x_i, y_i^+, y_i^-)$.
3: **for** $t \in \{1, \ldots, T\}$ **do**
4:    Objective function:

$$L_{\text{POWER-DL}}(l^t; \theta) = \mathbb{E}_{\mathcal{D}}\left[l_i^t L_{\text{POWER}}(x, y_i^+, y_i^-) + (1 - l_i^t)L_{\text{POWER}}(x, y_i^-, y_i^+)\right]$$

$$L_{\text{POWER}}(x, y^+, y^-) := \log\left(\sigma\Big(\beta[w(y^+)\log \pi_\theta(y^+|x) - w(y^-)\log \pi_\theta(y^-|x)\right.$$

$$\left. + w(y^+) - w(y^-)]\Big) + \eta\beta w(y^+)\log \pi(y^+|x)\right)$$

5:    Update policy with stop gradient on labels:
6:    $\theta_t \leftarrow \theta_{t-1} + \alpha \nabla_{\text{sg}(l^t)} L_{\text{POWER-DL}}(l^t; \theta)$
7:    Update dynamic labels:

$$l_i^t = (1 - \gamma)l_i^{t-1} + \gamma \frac{\sigma(w(y_i^+)\log \pi_\theta(y^+|x) - w(y_i^-)\log \pi_\theta(y^-|x) + w(y_i^+) - w(y_i^-)) - \hat{\mu}_{y_i^- \succ y_i^+}}{\hat{\mu}_{y_i^+ \succ y_i^-} - \hat{\mu}_{y_i^- \succ y_i^+}}$$

8: **Return:** $\pi_{\theta_T}$

---

## H    Experimental Details

### H.1    Preference Optimization Objectives

We compare POWER with a variety of preference optimization methods as baselines, summarized in Table 3. Several of these methods include divergence minimization (typically KL divergence) against the reference models, such as DPO (Rafailov et al., 2024b), IPO (Azar et al., 2024), which considers a different loss function on preferences, and $\chi$PO (Huang et al., 2024), which combines Chi-Squared with KL divergence for stronger pessimism. We also implement an offline variant of SPPO (Wu et al., 2024b), derived based on a self-play mechanism to improve upon the initial model.

We additionally consider several objectives that do not include the reference model. These include CPO (Xu et al., 2024a), which considers DPO with a uniform initial model; SLiC-HF (Zhao et al., 2023), which uses a hinge loss; RRHF (Yuan et al., 2024a), which applies a hinge loss with length-normalization on the contrastive term; and ORPO (Hong et al., 2024), which proposes odd ratio terms without initial models to contrast the chosen and rejected responses. Finally, SimPO (Meng et al., 2024) that removes the reference model from DPO and adds length-normalization and a margin. We implement POWER-DL by combining the objective in (8) with dynamic labels (11), where we estimate the empirical preferences with $\hat{\mu}_{1 \succ 0} = l$. We also compare against POWER, which corresponds to $\gamma = 0$, removing the dynamic labels.

### H.2    Instruction-Following Benchmarks

**Benchmark details.** We select the default choices in benchmark as baselines. In particular, for AlpacaEval 2.0 benchmark, we use GPT-4-Preview-1106 as comparison baseline model, and GPT-4 as the judge model. AlpacaEval 2.0 then compares responses generated by our PO trained models with responses from the baseline model, and length-controlled (LC) and raw winrate (WR) are computed as metrics. For Arena Hard benchmark, we use the GPT-4-0314 as the baseline model, and GPT-4 as the judge model, and the reported metric is winrate againts the baseline model.

**Decoding hyperparameters.** We follow Meng et al. (2024) and use a sampling decoding strategy with a temperature 0.9 on AlpacaEval 2.0 for all methods. For Arena Hard, we use the default approach of greedy generation.

Table 3: Different preference optimization objectives and hyperparameter search range.

| Method | Objective (Min) | Hyperparameter Range |
|---|---|---|
| DPO | $-\log \sigma \left( \beta \log \frac{\pi_\theta(y^+|x)}{\pi_{\theta_0}(y^+|x)} - \beta \log \frac{\pi_\theta(y^-|x)}{\pi_{\theta_0}(y^-|x)} \right)$ | $\beta \in \{0.001, 0.005, 0.01, 0.05, 0.1\}$ |
| DPO+SFT | $-\log \sigma \left( \beta \log \frac{\pi_\theta(y^+|x)}{\pi_{\theta_0}(y^+|x)} - \beta \log \frac{\pi_\theta(y^-|x)}{\pi_{\theta_0}(y^-|x)} \right) - \lambda \log \pi_\theta(y^+|x)$ | $\beta \in \{0.001, 0.005, 0.01, 0.05, 0.1\}$ 
 $\lambda \in \{0.0005, 0.001, 0.01, 0.1, 1\}$ |
| cDPO | $-(1-c) \log \sigma \left( \beta \log \frac{\pi_\theta(y^+|x)}{\pi_{\theta_0}(y^+|x)} - \beta \log \frac{\pi_\theta(y^-|x)}{\pi_{\theta_0}(y^-|x)} \right)$ 
 $-c \log \sigma \left( \beta \log \frac{\pi_\theta(y^-|x)}{\pi_{\theta_0}(y^-|x)} - \beta \log \frac{\pi_\theta(y^+|x)}{\pi_{\theta_0}(y^+|x)} \right)$ | $\beta \in \{0.001, 0.005, 0.01, 0.05, 0.1\}$ 
 $c \in \{0.05, 0.1, 0.15, 0.2, 0.3\}$ |
| R-DPO | $-\log \sigma \left( \beta \log \frac{\pi_\theta(y^+|x)}{\pi_{\theta_0}(y^+|x)} - \beta \log \frac{\pi_\theta(y^-|x)}{\pi_{\theta_0}(y^-|x)} - (\alpha|y^+| - \alpha|y^-|) \right)$ | $\alpha \in \{0.005, 0.01, 0.05, 0.1, 0.5\}$ 
 $\beta \in \{0.001, 0.005, 0.01, 0.05, 0.1\}$ |
| IPO | $\left( \log \frac{\pi_\theta(y^+|x)}{\pi_{\theta_0}(y^+|x)} - \log \frac{\pi_\theta(y^-|x)}{\pi_{\theta_0}(y^-|x)} - \frac{1}{2\tau} \right)^2$ | $\tau \in \{0.001, 0.005, 0.01, 0.1, 1.0\}$ |
| $\chi$PO | $-\log \left( \sigma \left( \mathrm{clip}_{2R} \left[ \beta\phi \left( \frac{\pi_\theta(y^+|x)}{\pi_{\theta_0}(y^+|x)} \right) - \beta\phi \left( \frac{\pi_\theta(y^-|x)}{\pi_{\theta_0}(y^-|x)} \right) \right] \right) \right)$ 
 $\phi(z) := z + \log(z)$ | $\beta \in \{0.001, 0.01, 0.1\}$ 
 $R \in \{0.1, 0.5, 1, 5, 10\}$ |
| SPPO (offline) | $\left( \beta \log \frac{\pi_\theta(y^+|x)}{\pi_{\theta_0}(y^+|x)} - \frac{1}{2} \right)^2 + \left( \beta \log \frac{\pi_\theta(y^-|x)}{\pi_{\theta_0}(y^-|x)} + \frac{1}{2} \right)^2$ | $\beta \in \{0.1, 1, 10, 100, 1000, 10000\}$ |
| CPO | $-\log \sigma \left( \beta \log \pi_\theta(y^+|x) - \beta \log \pi_\theta(y^-|x) \right) - \lambda \log \pi_\theta(y^+|x)$ | $\lambda = 1.0$ 
 $\beta \in \{0.001, 0.01, 0.1, 1, 10\}$ |
| RRHF | $\max \left\{ 0, -\frac{1}{|y^+|} \log \pi_\theta(y^+|x) + \frac{1}{|y^-|} \log \pi_\theta(y^-|x) \right\}$ 
 $-\lambda \log \pi_\theta(y^+|x)$ | $\lambda \in \{0.01, 0.05, 0.1, 0.5, 1, 10\}$ |
| SLiC-HF | $\max \left( 0, \beta - \log \pi_\theta(y^+|x) + \log \pi_\theta(y^-|x) \right) - \lambda \log \pi_\theta(y^+|x)$ | $\lambda \in \{0.01, 0.05, 0.1, 0.5, 1, 10\}$ 
 $\beta \in \{0.1, 0.5, 1.0, 2.0\}$ |
| ORPO | $-\log p_\theta(y^+|x) - \lambda \log \sigma \left( \log \frac{p_\theta(y^-|x)}{1 - p_\theta(y^-|x)} - \log \frac{p_\theta(y^+|x)}{1 - p_\theta(y^+|x)} \right)$ 
 $p_\theta(y|x) := \exp \left( \frac{1}{|y^-|} \log \pi_\theta(y^-|x) \right)$ | $\lambda \in \{0.1, 0.5, 1.0, 2.0\}$ |
| SimPO | $-\log \left( \frac{\beta}{|y^-|} \log \pi_\theta(y^+|x) - \frac{\beta}{|y^-|} \log \pi_\theta(y^-|x) - \gamma \right)$ | $\beta \in \{1, 2, 10, 20\}$ 
 $\gamma \in \{0.3, 0.5, 0.8, 1.0\}$ |
| ROPO | $-\alpha \log \sigma \left( \beta \log \frac{\pi_\theta(y^+|x)}{\pi_{\theta_0}(y^+|x)} - \beta \log \frac{\pi_\theta(y^-|x)}{\pi_{\theta_0}(y^-|x)} \right)$ 
 $+\gamma \sigma \left( \beta \log \frac{\pi_\theta(y^-|x)}{\pi_{\theta_0}(y^-|x)} - \beta \log \frac{\pi_\theta(y^+|x)}{\pi_{\theta_0}(y^+|x)} \right)$ | $\beta \in \{0.001, 0.005, 0.01, 0.05, 0.1\}$ 
 $\gamma = 0.1, \alpha \in \{0.2, 2, 20, 200, 2000\}$ |

## H.3 TRAINING AND HYPERPARAMETER DETAILS

**Hyperparameters for training reference models in the base setting.** We train initial reference models in the base setups. In the Helpsteer2 setting (Wang et al., 2024e), we train a model through supervised instruction finetuning on the (English only) OpenAssistant2 dataset (Köpf et al., 2024). In the Zephyr setting (Tunstall et al., 2023), we conduct supervised finetuning on the UltraChat-200K dataset (Ding et al., 2023). We use the following hyperparameters for both cases: a train batch size of 256, learning rate of 2e-5 with a cosine learning rate schedule with 10% warmup, right padding, and a max sequence length of 2048. We train the models with Adam optimizer for 1 epoch.

**General hyperparameters for preference optimization.** We use a fixed batch size of 128 and a maximum sequence length of 2048 for all methods. For learning rate, we search over {3e-7,5e-7} separately for each method and use a cosine learning rate schedule with a 10% warmup. We use Adam optimizer for all the approaches. In the Helpsteer2 setting and following Wang et al. (2024e), we train the models for up to 7 epochs, and in the Zephyr setting, we train the models for up to 3 epochs, and select the best number of epochs for each method according to validation. We use right padding for preference optimization following the recommendation of Hu et al. (2024).

**Specific hyperparameters for preference optimization.** For the hyperparameters specific to each preference optimization objective, we conduct hyperparameter search according to the values in Table 3. In each setting, we select the best model according to the ranking performance on the validation set. For POWER-DL, we conduct hyperparameter search over $\beta \in \{1, 2, 10, 20\}$, $\eta \in$

$\{0.0005, 0.001\}$, and $\gamma \in \{0.1, 0.3\}$. For Helpsteer2, we select $\beta = 10, \eta = 0.001, \gamma = 0.1$, 5 epochs, and learning rate of 5e-7 in the base setting and $\beta = 20, \eta = 0.0005, \gamma = 0.1$, 4 epochs and learning rate of 3e-7 in the instruct setting. For Zephyr, we select $\beta = 1, \eta = 0.001, \gamma = 0.1$, 2 epochs, and learning rate of 3e-7 in the base setting, and $\beta = 2, \eta = 0.0005, \gamma = 0.3$, 2 epochs, and learning rate of 3e-7 in the instruct setting.

**Computation environment.** All experiments are conducted on $8 \times$A100 GPUs based on the Open-RLHF repository (Hu et al., 2024).

# I  ADDITIONAL EXPERIMENTAL RESULTS ON THE LLAMA FAMILY

## I.1  PERFORMANCE ON ACADEMIC BENCHMARKS

Tables 4 and 5 present the benchmark scores for the Helpsteer2 and Zephyr pipelines across a variety of downstream tasks. Considering the average benchmark score, POWER-DL consistently improves over the initial model and ranks within the top two or three methods across all four settings, despite significantly outperforming other methods in alignment benchmarks AlpacaEval 2.0 and Arena-Hard as detailed in Table 2. Achieving a high score on instruction-following benchmarks AlpacaEval 2.0 and Arena-Hard while maintaining a good performance on downstream tasks is considered key empirical evidence for mitigating reward hacking in practice (Xu et al., 2024b).

Table 4: Downstream task evaluation results of the models trained on the Helpsteer2 pipeline. The arrows show improvement or degradation of performance with respect to the initial model. The top three average scores are shown in bold.

| Task | MMLU | ARC | HellaSwag | TruthfulQA | Winogrande | GSM8K | IFEval | MMLU-PRO | Average |
|---|---|---|---|---|---|---|---|---|---|
| **Helpsteer2 Llama3-8B-Base** | | | | | | | | | |
| Initial Model | 62.6 | 58.4 | 80.0 | 49.5 | 77.4 | 37.1 | 33.6 | 29.8 | 53.5 |
| DPO | 61.2 ↓1.4 | 57.5 ↓0.9 | 81.1 ↑1.1 | 51.0 ↑1.5 | 75.6 ↓1.8 | 32.5 ↓4.6 | 41.1 ↑7.5 | 29.8 | 53.7 ↑0.2 |
| DPO+SFT | 62.1 ↓0.5 | 60.5 ↑2.1 | 81.9 ↑1.9 | 52.9 ↑3.4 | 77.0 ↓0.4 | 42.1 ↑5.0 | 42.5 ↑8.9 | 30.5 ↑0.7 | **56.2** ↑2.7 |
| cDPO | 62.0 ↓0.6 | 60.9 ↑2.5 | 82.9 ↑2.9 | 53.9 ↑4.4 | 76.6 ↓0.8 | 37.9 ↑0.8 | 42.6 ↑9.0 | 30.5 ↑0.7 | 55.9 ↑2.4 |
| R-DPO | 62.8 ↑0.2 | 58.2 ↓0.2 | 80.3 ↑0.3 | 52.9 ↑3.4 | 77.4 | 43.2 ↑6.1 | 38.7 ↑5.1 | 30.1 ↑0.3 | 55.5 ↑2.0 |
| IPO | 62.7 ↑0.1 | 60.6 ↑2.2 | 81.7 ↑1.7 | 52.5 ↑3.0 | 78.1 ↑0.7 | 43.8 ↑6.7 | 42.3 ↑8.7 | 30.3 ↑0.5 | **56.5** ↑3.0 |
| $\chi$PO | 62.9 ↑0.3 | 59.0 ↑0.6 | 81.0 ↑1.0 | 51.7 ↑2.2 | 78.1 ↑0.7 | 41.7 ↑4.6 | 38.0 ↑4.4 | 30.1 ↑0.3 | 55.3 ↑1.8 |
| SPPO | 62.8 ↑0.2 | 60.9 ↑2.5 | 83.0 ↑3.0 | 54.2 ↑4.7 | 77.1 ↓0.3 | 38.8 ↑1.7 | 42.7 ↑9.1 | 30.6 ↑0.8 | **56.3** ↑2.8 |
| CPO | 62.6 | 59.0 ↑0.6 | 80.2 ↑0.2 | 54.2 ↑4.7 | 77.0 ↓0.4 | 44.2 ↑7.1 | 41.4 ↑7.8 | 30.4 ↑0.6 | 56.1 ↑2.6 |
| RRHF | 62.6 | 57.7 ↓0.7 | 79.0 ↓1.0 | 51.1 ↑1.6 | 77.2 ↓0.2 | 37.2 ↑0.1 | 34.2 ↑0.6 | 29.8 | 53.6 ↓0.1 |
| SLiCHF | 62.6 | 59.0 ↑0.6 | 79.9 ↓0.1 | 55.0 ↑5.5 | 76.8 ↓0.6 | 43.7 ↑6.6 | 40.1 ↑6.5 | 30.2 ↑0.4 | 55.9 ↑2.4 |
| ORPO | 61.5 ↓1.1 | 57.6 ↓0.8 | 79.0 ↓1.0 | 61.4 ↑11.9 | 77.7 ↑0.3 | 15.6 ↓21.5 | 40.4 ↑6.8 | 29.6 ↓0.2 | 52.9 ↓0.7 |
| SimPO | 61.3 ↓1.3 | 59.0 ↑0.6 | 80.6 ↑0.6 | 59.6 ↑10.1 | 77.7 ↑0.3 | 23.4 ↓13.7 | 40.5 ↑6.9 | 30.2 ↑0.4 | 54.0 ↑0.5 |
| ROPO | 61.5 ↓1.1 | 60.9 ↑2.5 | 82.1 ↑2.1 | 52.5 ↑3.0 | 76.6 ↓0.8 | 37.5 ↑0.4 | 41.4 ↑7.8 | 30.1 ↑0.3 | 55.3 ↑1.8 |
| POWER-DL | 62.0 ↓0.6 | 59.6 ↑1.2 | 82.0 ↑2.0 | 61.0 ↑11.5 | 78.1 ↑0.7 | 36.5 ↓0.6 | 40.3 ↑6.7 | 30.5 ↑0.7 | **56.3** ↑2.8 |
| POWER | 61.9 ↓0.7 | 60 ↑1.6 | 82.0 ↑2.0 | 61.0 ↑11.5 | 77.9 ↑0.5 | 35.3 ↓1.8 | 40.1 ↑6.5 | 30.4 ↑0.6 | 56.1 ↑2.6 |
| **Helpsteer2 Llama3-8B-Instruct** | | | | | | | | | |
| Initial Model | 65.7 | 62.0 | 78.8 | 51.7 | 76.0 | 75.3 | 54.4 | 36.0 | 62.5 |
| DPO | 65.9 ↑0.2 | 63.4 ↑1.4 | 79.5 ↑0.7 | 52.7 ↑1.0 | 75.9 ↓0.1 | 76.4 ↑1.1 | 54.1 ↓0.3 | 36.3 ↑0.3 | 63.0 ↑0.5 |
| DPO+SFT | 65.9 ↑0.2 | 61.0 ↓1.0 | 73.6 ↓5.2 | 54.7 ↑3.0 | 71.2 ↓4.8 | 74.7 ↓0.6 | 48.2 ↓6.2 | 37.0 ↑1.0 | 60.8 ↓1.7 |
| cDPO | 66.0 ↑0.3 | 65.4 ↑3.4 | 79.5 ↑0.7 | 57.0 ↑5.3 | 74.6 ↓1.4 | 76.9 ↑1.6 | 49.9 ↓4.5 | 36.9 ↑0.9 | **63.3** ↑0.8 |
| R-DPO | 65.8 ↑0.1 | 62.8 ↑0.8 | 74.8 ↓4.0 | 54.6 ↑2.9 | 72.6 ↓3.4 | 77.7 ↑2.4 | 51.0 ↓3.4 | 36.6 ↑0.6 | 62.0 ↓0.5 |
| IPO | 66.0 ↑0.3 | 64.9 ↑2.9 | 79.2 ↑0.4 | 58.4 ↑6.7 | 73.8 ↓2.2 | 75.8 ↑0.5 | 49.8 ↓4.6 | 37.1 ↑1.1 | **63.1** ↑0.6 |
| $\chi$PO | 65.8 ↑0.1 | 63.7 ↑1.7 | 75.6 ↓3.2 | 59.1 ↑7.4 | 72.4 ↓3.6 | 75.2 ↓0.1 | 52.4 ↓2.0 | 37.2 ↑1.2 | 62.7 ↑0.2 |
| SPPO | 66.0 ↑0.3 | 63.0 ↑1.0 | 76.7 ↓2.1 | 55.7 ↑4.0 | 72.9 ↓3.1 | 75.7 ↑0.4 | 49.6 ↓4.8 | 37.1 ↑1.1 | 62.1 ↓0.4 |
| CPO | 65.7 | 62.2 ↑0.2 | 78.0 ↓0.8 | 52.2 ↑0.5 | 74.1 ↓1.9 | 75.7 ↑0.4 | 49.9 ↓4.5 | 36.0 | 61.9 ↓0.6 |
| RRHF | 65.9 ↑0.2 | 62.0 | 77.5 ↓1.3 | 51.5 ↓0.2 | 73.8 ↓2.2 | 76.7 ↑1.4 | 51.2 ↓3.2 | 36.4 ↑0.4 | 61.9 ↓0.6 |
| SLiCHF | 65.7 | 62.9 ↑0.9 | 78.0 ↓0.8 | 53.9 ↑2.2 | 74.2 ↓1.8 | 76.7 ↑1.4 | 47.6 ↓6.8 | 36.1 ↑0.1 | 62.3 ↓0.2 |
| ORPO | 65.7 | 62.1 ↑0.1 | 74.0 ↓4.8 | 56.7 ↑5.0 | 71.5 ↓4.5 | 75.9 ↑0.6 | 51.3 ↓3.1 | 37.2 ↑1.2 | 61.9 ↓0.6 |
| SimPO | 65.8 ↑0.1 | 61.9 ↓0.1 | 75.0 ↓3.8 | 58.3 ↑6.6 | 72.3 ↓3.7 | 74.3 ↓1.0 | 54.1 ↓0.3 | 37.2 ↑1.2 | 62.3 ↓0.2 |
| ROPO | 65.4 ↓0.3 | 62.0 | 76.9 ↓1.9 | 54.1 ↑2.4 | 73.7 ↓2.3 | 73.9 ↓1.4 | 50.2 ↓4.2 | 36.2 ↑0.2 | 61.8 ↓0.7 |
| POWER-DL | 66.0 ↑0.3 | 64.3 ↑2.3 | 79.5 ↑0.7 | 53.1 ↑1.4 | 76.0 | 76.3 ↑1.0 | 53.5 ↓0.9 | 36.6 ↑0.6 | **63.2** ↑0.7 |
| POWER | 65.8 ↑0.1 | 63.9 ↑1.9 | 79.6 ↑0.8 | 53.1 ↑1.4 | 76.1 ↑0.1 | 76.6 ↑1.3 | 52.5 ↓1.9 | 36.4 ↑0.4 | 63.0 ↑0.5 |

Table 5: Downstream task evaluation results of the model trained on the Zephyr pipeline. The arrows show improvement or degradation of performance with respect to the initial model. The top three average scores are shown in bold.

| Task | MMLU | ARC | HellaSwag | TruthfulQA | Winogrande | GSM8K | IFEval | MMLU-PRO | Average |
|---|---|---|---|---|---|---|---|---|---|
| **Zephyr Llama3-8B-Base** | | | | | | | | | |
| Initial Model | 61.6 | 58.2 | 78.6 | 52.1 | 75.9 | 47.3 | 38.1 | 29.4 | 55.2 |
| DPO | 61.9 ↑0.3 | 62.4 ↑4.2 | 81.6 ↑3.0 | 63.0 ↑11.0 | 74.4 ↓1.4 | 52.8 ↑5.5 | 50.6 ↑12.5 | 31.2 ↑1.7 | **59.7 ↑4.6** |
| DPO+SFT | 62.0 ↑0.3 | 62.0 ↑3.8 | 80.9 ↑2.3 | 61.6 ↑9.5 | 75.1 ↓0.7 | 55.0 ↑7.7 | 48.7 ↑10.6 | 31.0 ↑1.6 | 59.5 ↑4.4 |
| cDPO | 61.9 ↑0.3 | 61.9 ↑3.7 | 81.4 ↑2.7 | 62.3 ↑10.2 | 75.3 ↓0.5 | 54.7 ↑7.4 | 49.3 ↑11.2 | 31.0 ↑1.6 | **59.7 ↑4.6** |
| R-DPO | 61.9 ↑0.2 | 59.5 ↑1.3 | 80.1 ↑1.5 | 62.0 ↑9.9 | 75.0 ↓0.9 | 52.8 ↑5.5 | 45.3 ↑7.2 | 30.5 ↑1.1 | 58.4 ↑3.2 |
| IPO | 61.8 ↑0.1 | 58.3 ↑0.1 | 78.7 ↑0.1 | 53.5 ↑1.4 | 76.7 ↑0.9 | 46.8 ↓0.5 | 39.5 ↑1.3 | 29.7 ↑0.3 | 55.6 |
| χPO | 62.1 ↑0.4 | 60.8 ↑2.6 | 80.2 ↑1.6 | 57.9 ↑5.8 | 75.3 ↓0.5 | 54.8 ↑7.5 | 48.8 ↑10.7 | 30.2 ↑0.7 | 58.7 ↑3.6 |
| SPPO | 61.5 ↓0.1 | 61.9 ↑3.7 | 81.2 ↑2.6 | 62.0 ↑10.0 | 73.4 ↓2.4 | 51.9 ↑4.6 | 50.7 ↑12.6 | 30.7 ↑1.3 | 59.2 ↑4.0 |
| CPO | 61.6 | 56.1 ↓2.1 | 77.8 ↓0.8 | 51.5 ↓0.6 | 75.9 | 40.2 ↓7.1 | 37.3 ↓0.8 | 29.3 ↓0.2 | 53.4 ↓1.8 |
| RRHF | 61.7 ↑0.1 | 55.8 ↓2.4 | 77.7 ↓0.9 | 51.3 ↓0.8 | 75.6 ↓0.2 | 39.1 ↓8.2 | 36.6 ↓1.6 | 29.3 ↓0.2 | 53.4 ↓1.8 |
| SLiCHF | 61.8 ↑0.2 | 57.9 ↓0.3 | 79.4 ↑0.8 | 60.2 ↑8.2 | 75.6 ↓0.2 | 49.4 ↑2.1 | 42.3 ↑4.2 | 30.3 ↑0.9 | 57.1 ↑2.0 |
| ORPO | 61.8 ↑0.1 | 62.5 ↑4.3 | 81.2 ↑2.6 | 63.9 ↑11.9 | 76.7 ↑0.9 | 48.9 ↑1.6 | 56.1 ↑18.0 | 30.8 ↑1.4 | **60.2 ↑5.1** |
| SimPO | 61.9 ↑0.3 | 62.4 ↑4.2 | 81.6 ↑3.0 | 63.0 ↑11.0 | 74.4 ↓1.4 | 42.8 ↓4.5 | 50.6 ↑12.5 | 31.2 ↑1.7 | 58.5 ↑3.3 |
| ROPO | 61.5 ↓0.2 | 61.7 ↑3.5 | 81.4 ↑2.8 | 64.8 ↑12.8 | 73.8 ↓2.1 | 54.4 ↑7.1 | 49.9 ↑11.8 | 30.9 ↑1.4 | **59.8 ↑4.6** |
| POWER-DL | 61.5 ↓0.2 | 61.7 ↑3.5 | 81.4 ↑2.8 | 64.8 ↑12.8 | 73.8 ↓2.1 | 54.4 ↑7.1 | 49.9 ↑11.8 | 30.9 ↑1.4 | **59.8 ↑4.6** |
| POWER | 61.8 ↑0.2 | 61.8 ↑3.6 | 80.6 ↑2.0 | 59.8 ↑7.7 | 76.6 ↑0.8 | 46.3 ↓1.1 | 53.7 ↑15.6 | 30.4 ↑1.0 | 58.9 ↑3.7 |
| **Zephyr Llama3-8B-Instruct** | | | | | | | | | |
| Initial Model | 65.7 | 62.0 | 78.8 | 51.7 | 76.0 | 75.3 | 54.4 | 36.0 | 62.5 |
| DPO | 66.0 ↑0.3 | 63.0 ↑0.9 | 76.7 ↓2.1 | 55.7 ↑4.0 | 72.9 ↓3.1 | 75.7 ↑0.4 | 49.6 ↓4.8 | 37.1 ↑1.1 | 62.1 ↑0.5 |
| DPO+SFT | 66.0 ↑0.3 | 66.6 ↑4.5 | 79.2 ↑0.5 | 59.9 ↑8.2 | 75.0 ↓1.0 | 75.7 ↑0.4 | 51.8 ↓2.6 | 37.3 ↑1.3 | **64.2 ↑1.7** |
| cDPO | 66.0 ↑0.3 | 67.8 ↑5.8 | 80.5 ↑1.8 | 59.0 ↑7.3 | 75.1 ↓1.0 | 76.9 ↑1.6 | 51.2 ↓3.2 | 37.3 ↑1.4 | **64.2 ↑1.7** |
| R-DPO | 65.7 ↑0.0 | 66.0 ↑4.0 | 78.2 ↓0.6 | 58.9 ↑7.2 | 74.5 ↓1.5 | 75.8 ↑0.5 | 50.7 ↓3.7 | 36.9 ↑0.9 | 63.3 ↑0.8 |
| IPO | 65.8 ↑0.1 | 62.1 ↑0.1 | 78.7 ↓0.0 | 51.8 ↑0.1 | 75.9 ↓0.2 | 75.8 ↑0.5 | 54.0 ↓0.5 | 35.8 ↓0.1 | 62.2 ↑0.3 |
| χPO | 66.2 ↑0.5 | 65.4 ↑3.3 | 80.5 ↑1.7 | 54.2 ↑2.6 | 76.2 ↑0.1 | 76.8 ↑1.5 | 54.6 ↑0.1 | 37.1 ↑1.1 | **64.0 ↑1.2** |
| SPPO | 66.1 ↑0.4 | 65.8 ↑3.8 | 78.7 ↓0.1 | 58.6 ↑6.9 | 74.1 ↓1.9 | 74.0 ↓1.3 | 55.2 ↑0.7 | 37.4 ↑1.4 | **63.7 ↑1.1** |
| CPO | 65.4 ↓0.3 | 61.9 ↓0.2 | 77.8 ↓0.9 | 52.3 ↑0.6 | 75.5 ↓0.5 | 75.4 ↑0.1 | 53.8 ↓0.6 | 35.9 ↓0.1 | 62.2 ↓0.2 |
| RRHF | 65.4 ↓0.3 | 61.9 ↓0.2 | 77.7 ↓1.1 | 52.3 ↑0.6 | 75.3 ↓0.7 | 75.3 | 54.2 ↓0.2 | 35.8 ↓0.2 | 62.2 ↓0.2 |
| SLiCHF | 65.6 | 63.1 ↑1.0 | 79.1 ↑0.3 | 56.0 ↑4.3 | 75.4 ↓0.6 | 76.8 ↑1.5 | 48.8 ↓5.6 | 36.4 ↑0.5 | 62.9 ↑0.4 |
| ORPO | 65.9 ↑0.2 | 64.5 ↑2.5 | 78.5 ↓0.3 | 57.6 ↑5.9 | 75.3 ↓0.7 | 77.2 ↑1.9 | 52.3 ↓2.2 | 36.7 ↑0.8 | **63.7 ↑1.1** |
| SimPO | 65.8 ↑0.1 | 62.1 ↑0.1 | 74.2 ↓4.6 | 57.4 ↑5.7 | 71.1 ↓4.9 | 72.4 ↓2.9 | 54.1 ↓0.4 | 37.0 ↑1.0 | 61.8 ↓0.7 |
| ROPO | 66.2 ↑0.5 | 63.6 ↑1.5 | 76.4 ↓2.4 | 58.1 ↑6.4 | 72.9 ↓3.1 | 73.0 ↓2.3 | 55.6 ↑1.2 | 37.5 ↑1.5 | 62.9 ↑0.4 |
| POWER-DL | 65.8 ↑0.1 | 64.7 ↑2.7 | 76.9 ↓1.8 | 59.5 ↑7.9 | 73.6 ↓2.4 | 76.2 ↑0.9 | 55.6 ↑1.2 | 37.2 ↑1.3 | **63.7 ↑1.1** |
| POWER | 65.7 | 63.1 ↑1.1 | 75.0 ↓3.7 | 59.1 ↑7.4 | 71.7 ↓4.3 | 75.7 ↑0.5 | 54.6 ↑0.1 | 37.1 ↑1.1 | 62.9 ↑0.4 |

## I.2 PERFORMANCE ON MT-BENCH

We evaluate the trained models on MT-Bench, which includes 80 questions across 8 categories. We report the average score in Table 6 as evaluated by GPT-4 as the judge model. The highest score in trained models is shown in bold and the second highest score is underlined. POWER-DL consistently outperforms other preference optimization methods. Similar to Meng et al. (2024), we observe that variations across different methods in MT-Bench scores are small compared to AlpacaEval 2.0 and Arena-Hard.

Table 6: Llama3 MT-Bench results on Helpsteer2 and Zephyr settings.

| | Helpsteer2 | | Zephyr | |
| | Llama3-8B-Base | Llama3-8B-Instruct | Llama3-8B-Base | Llama3-8B-Instruct |
| Method | GPT-4 Score | GPT-4 Score | GPT-4 Score | GPT-4 Score |
| --- | --- | --- | --- | --- |
| Initial Model | 4.9 | 8.3 | 6.1 | 8.3 |
| DPO | 5.6 | **8.2** | 7.0 | **8.3** |
| DPO+SFT | 5.5 | 8.1 | 7.0 | 8.0 |
| cDPO | 5.8 | 8.1 | 6.9 | 8.2 |
| R-DPO | 5.4 | **8.2** | 6.9 | 8.1 |
| IPO | 4.9 | 8.0 | 6.3 | 8.0 |
| $\chi$PO | 5.5 | 8.0 | 6.9 | 8.1 |
| SPPO | 5.8 | **8.2** | **7.1** | 8.2 |
| CPO | 5.9 | 7.2 | 6.3 | 8.0 |
| RRHF | 5.7 | 7.9 | 6.2 | 8.1 |
| SLiC-HF | 5.9 | 7.8 | 6.8 | 7.9 |
| ORPO | 5.7 | 8.1 | 6.6 | **8.3** |
| SimPO | **6.0** | 8.0 | 6.8 | 8.2 |
| ROPO | 5.8 | 8.1 | **7.1** | 8.2 |
| POWER-DL | **6.0** | **8.2** | **7.1** | 8.2 |
| POWER | 5.9 | **8.2** | 7.0 | 8.2 |

## I.3 EXPERIMENTS ON ITERATIVE PREFERENCE OPTIMIZATION

The multi-iteration experiments are conducted on the Zephyr pipeline that extends the single-iteration instruct setting into three iteration by splitting the dataset of prompts. We compare our approach with iterative DPO (Dong et al., 2024), XPO (Xie et al., 2024), which adds negative SFT to DPO, and an iterative variant of SimPO. Table 7 presents the results, demonstrating that the benefits of our approach that extend to the multi-iteration setting.

Table 7: Instruction-following benchmark results in the multi-iteration setting.

| Method | AlpacaEval 2.0 (LC%) | AlpacaEval 2.0 (WR%) | Arena-Hard (WR%) |
| --- | --- | --- | --- |
| Initial Model | 33.41 | 32.40 | 23.0 |
| DPO iter=1 | 40.02 | 38.58 | 31.6 |
| DPO iter=2 | 42.15 | **40.71** | 37.80 |
| DPO iter=3 | 42.38 | 40.8 | 38.8 |
| XPO iter=1 | 40.96 | **39.50** | 32.6 |
| XPO iter=2 | 41.33 | 39.66 | 36.3 |
| XPO iter=3 | 42.42 | 39.98 | 35.7 |
| SimPO iter=1 | 38.91 | 36.44 | 28.7 |
| SimPO iter=2 | 41.15 | 37.77 | 28.9 |
| SimPO iter=3 | 41.97 | 38.32 | 32.6 |
| POWER-DL iter=1 | **41.42** | 37.64 | **34.4** |
| POWER-DL iter=2 | **43.48** | 40.17 | **39.8** |
| POWER-DL iter=3 | **46.55** | **43.21** | **42.1** |

## I.4 HYPERPARAMETER ROBUSTNESS RESULTS

We assess the hyperparameter robustness of POWER-DL by examining its performance across a range of values for $\gamma$, $\eta$, and $\beta$ in the Helpsteer2 base setting. Figure 2 shows AlpacaEval winrate and length-controlled winrate for various hyperparameters. POWER-DL exhibits a robust behavior with respect to all three hyperparameters, particularly considering the length-controlled winrate.

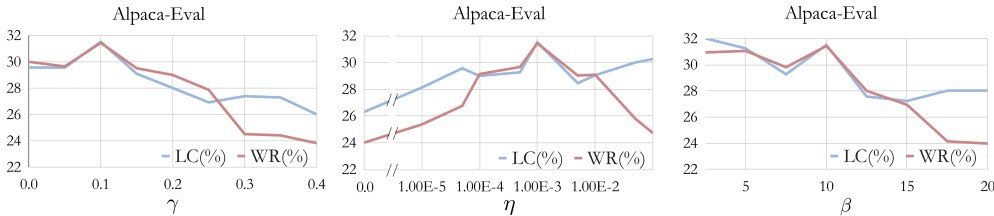

Figure 2: POWER-DL hyperparameter robustness results in the Helpsteer2 base setting.

I.5   EXAMPLES RESPONSES

The following is an example question along with responses from various models from the Helpsteer2 base pipeline. POWER-DL response shows analytical thinking, a deeper reasoning process, and consideration of implicit factors, characteristics reminiscent of System 2 thinking.

---

**Question**

One One was a race horse Two Two was one too. One One won one race. Two Two won one too. How many horses are in this story?

---

**Responses from Different Models**

**Initial Model:** There are two horses in this story, One One and Two Two.

**GPT-4:** There are two horses in this story. Their names are "One One" and "Two Two."

---

**DPO:** Based on the information given in the riddle, there are two horses involved in the story. "The One One" and "The Two Two" are both mentioned as race horses, and each of them has won one race respectively. So, there is one "One One" horse and one "Two Two" horse in this story. Together, they make up the total two horses.

**cDPO:** Based on the information given in the riddle, there are two horses involved in the story. "The One One" and "The Two Two" are both designated as race horses, indicating that there are two horses with those specific monikers. "One One won one race" refers to the horse with the name "One One" winning one race. "And Two Two won one too" refers to the horse with the name "Two Two" winning one race as well, implying that both horses won one race each, not ruling out the possibility of both being race horses. So, there is "One One" horse and "Two Two" horse in this story.

**χPO:** There are two horses in this story.

**SPPO:** Based on the information provided in the riddle, there are two horses involved in the story. "One One" and "Two Two" are both race horses, and the sentence "One One won one race. Two Two won one too." indicates that both horses won one race each. So, there are two distinct horses that won a race each, namely One One and Two Two. Therefore, the number of horses in this story is 2.

**SimPO:** Based solely on the information provided in the riddle, there could be as many as three horses and as few as one. The first two lines introduce two horses, "One-One" and "Two-Two." The next two lines indicate that each can win a race, but the information about the races themselves and the horses that competed in them is missing. If "One-One" and "Two-Two" raced each other, then just those two horses would exist in this story. However, if either or both raced a third horse, the number of horses in the story could be three. Therefore, the actual number of horses in this riddle's story is unknown.

---

**POWER-DL (ours):** The story mentions two horses, One One and Two Two. However, it's possible that there could be more horses involved. The statement "One One won one race" does not specify whether the track was crowded with competitors, and it's possible that the horse had to race against multiple opponents. Therefore, the number of horses in the story could be anywhere from two to many. The statement only specifies the race results of the two named horses.

# J EXPERIMENTAL RESULTS ON MISTRAL FAMILY

In this section, we provide experimental results on conducting preference optimization on Mistral-7B (Jiang et al., 2023). We follow the same experimental details described earlier and finetune mistralai/Mistral-7B-v0.1 in the base setting and mistralai/Mistral-7B-Instruct-v0.2 in the instruct setting.

Table 8: AlpacaEval 2 and Arena-Hard results on Helpsteer2 and Zephyr settings for Mistral family.

| | Helpsteer2 | | | | | | Zephyr | | | | | |
| | Mistral-7B-Base | | | Mistral-7B-Instruct | | | Mistral-7B-Base | | | Mistral-7B-Instruct | | |
| | AlpacaEval | | Arena-Hard | AlpacaEval | | Arena-Hard | AlpacaEval | | Arena-Hard | AlpacaEval | | Arena-Hard |
| Method | LC(%) | WR(%) | WR(%) | LC(%) | WR(%) | WR(%) | LC(%) | WR(%) | WR(%) | LC(%) | WR(%) | WR(%) |
|---|---|---|---|---|---|---|---|---|---|---|---|---|
| Initial Model | 5.47 | 4.16 | 1.5 | 27.70 | 22.26 | 14.8 | 2.85 | 2.11 | 0.6 | 27.70 | 22.26 | 14.8 |
| DPO | 11.89 | 10.39 | 4.6 | 36.47 | 29.41 | 17.1 | 15.67 | 13.26 | 5.9 | 36.55 | 36.28 | 24.3 |
| DPO+SFT | 10.57 | 8.21 | 3.4 | 35.28 | 27.88 | 16.9 | 12.30 | 10.68 | 5.4 | 35.31 | 35.70 | 25.1 |
| cDPO | 12.12 | 10.52 | 3.5 | 32.07 | 29.22 | 16.6 | 13.57 | 12.24 | 5.4 | 31.42 | 30.43 | 20.9 |
| $\chi$PO | 10.88 | 8.64 | 4.1 | 38.90 | **37.02** | 22.5 | 9.80 | 8.35 | 3.4 | 34.79 | 35.53 | 17.0 |
| SimPO | 14.56 | 13.97 | 7.9 | 38.28 | 28.89 | 14.1 | 16.08 | 16.50 | 7.3 | 36.23 | 29.11 | 22.5 |
| POWER-DL | **19.83** | 15.40 | **8.2** | **42.26** | 34.72 | **23.7** | 20.34 | 19.50 | 12.1 | 42.57 | 42.53 | **28.0** |
| POWER | 19.72 | **16.04** | 6.5 | 39.23 | 33.06 | 20.1 | 17.09 | 15.25 | 10.2 | 38.13 | 36.85 | 26.2 |

Table 9: Mistral MT-Bench results on Helpsteer2 and Zephyr settings.

| | Helpsteer2 | | Zephyr | |
| | Mistral-7B-Base | Mistral-7B-Instruct | Mistral-7B-Base | Mistral-7B-Instruct |
| Method | GPT-4 Score | GPT-4 Score | GPT-4 Score | GPT-4 Score |
|---|---|---|---|---|
| Initial Model | 3.1 | 6.6 | 3.6 | 6.6 |
| DPO | 3.4 | 6.3 | 5.0 | **6.6** |
| DPO+SFT | 3.4 | 6.3 | **5.2** | 6.3 |
| cDPO | **3.5** | 6.1 | **5.2** | 6.5 |
| $\chi$PO | 3.2 | **6.6** | 5.0 | 6.2 |
| SimPO | 2.9 | 6.3 | 4.5 | 6.0 |
| POWER-DL | 3.4 | **6.6** | **5.2** | **6.6** |
| POWER | 3.2 | **6.6** | **5.2** | 6.5 |

