# OpenReview forum: "Sail into the Headwind: Alignment via Robust Rewards and Dynamic Labels against Reward Hacking"
_ICLR.cc/2025/Conference — ICLR 2025 Poster_

### Official Review · Reviewer_fMHu · 2024-10-20

**Soundness:** 3
**Presentation:** 3
**Contribution:** 3
**Rating:** 8
**Confidence:** 3

**Summary:**

This paper proposed two types of reward hacking in offline preference optimization . The Type I is when subpar choices appear more favorable due to sparse data, and the Type II is when good choices appear worse. They propose POWER(-DL), a new offline preference optimization method that (1) integrates weighted entropy with robust reward maximization to address Type I hacking and (2) use dynamically updating preference labels to resolve Type II hacking. Empirical results show that it outperforms state-of-the-art methods on benchmarks.

**Strengths:**

The use of weighted entropy and dynamic label is interesting and new as far as I am aware.

The reasoning is sound and convincing. For instance, for the first type of reward hacking, they established a lower bound (proposition 1) showing that regardless of the number of samples, certain existing algorithms will suffer from a constant suboptimality with probability of at least some constant. Then they show that their proposed algorithm avoids this issue (Theorem 1).

The experimental results are impressive. The proposed method consistently outperforms baselines across multiple models.

**Weaknesses:**

Both types of reward hacking issues arise from insufficient data coverage, a well-explored topic in bandits and reinforcement learning. Therefore, to me, the two new definitions of reward hacking don’t feel entirely novel to me. Although I admit that most prior work hasn’t focused on learning-from-preference settings. This consideration actually leads to some questions that I show in the Question section.


It might be beneficial to merge Theorem 2 with Theorem 1 to provide a unified theoretical result for the proposed algorithm. Currently, they are presented separately, so I wonder how the use of dynamic labels will affect the suboptimality bound shown in Theorem 1.

**Questions:**

Compared to Zhan et al., 2023a, what is the advantage of your method? I think the settings are pretty similar and the concentrability coefficient is defined in the same way (please correct me if I missed any subtitles). Since you claim to address both reward hacking issues by proving a finite-sample bound, as in Theorem 1, do Zhan et al. (2023a) address them as well?

If I simply set $w(\cdot)=0$, what would go wrong in Theorem 1? I believe that you have hidden some $w$-dependence term in the bound, and I am curious to know what that is.

Can you provide some intuitive explanations on why choosing $w(y)=1/|y|$ leads to $\log|\mathcal V|$ in Theorem 1?

Minor problem: I don't see how "Sail into the Headwind" is informative in the title

---

> ### Author Response · Authors · 2024-11-24
> **Response (1/2)**
>
> Thank you for the time spent reviewing our work and your positive feedback. We have updated our manuscript to reflect your feedback and respond to your questions and comments below.
>
> > Both types of reward hacking issues arise from insufficient data coverage, a well-explored topic in bandits and reinforcement learning. Therefore, to me, the two new definitions of reward hacking don’t feel entirely novel to me. Although I admit that most prior work hasn’t focused on learning-from-preference settings. This consideration actually leads to some questions that I show in the Question section.
>
> We clarify that the existing literature on insufficient data coverage in offline RL is only related to I reward hacking, and the notion of type II reward hacking and our classification are new. We highlight the differences between the setting we considered in this paper with offline RL, explaining the usefulness and novelty in defining the two types.
>
> In the practice of RLHF finetuning of LLMs, we typically have access an initial model, which already has a decent performance on many downstream tasks, and a previously-collected preference data, which may not have been sampled from initial model (Wang et al. 2024). In this setting, we face two sources of distribution shift: one between the final model and data distribution, and the other between the initial model and data distribution. This setting is different from conventional offline RL, that considers access to an offline dataset (with possibly known data collection policy) and is only concerned with distribution shift between final model and data distribution (Levine et al. 2020). Due to the existence of two sources of distribution shift, we find it useful to define Type I and Type II reward hacking. These definitions motivate the design of our algorithm that achieves theoretical guarantees and a strong empirical performance.
>
> > It might be beneficial to merge Theorem 2 with Theorem 1 to provide a unified theoretical result for the proposed algorithm. Currently, they are presented separately, so I wonder how the use of dynamic labels will affect the suboptimality bound shown in Theorem 1.
>
> We agree that it would be beneficial to merge the two results; however, such a merge is challenging as we now explain. Theoretical guarantee for POWER in Theorem 1 hold for general function classes and under a weak form of single-policy concentrability. However, in Theorem 2, we analyze convergence properties of a system of coupled differential equations, reflecting the learning dynamics of preference optimization combined with dynamic labels. Theorem 2 focuses on the softmax policy instead of general function class, as such learning dynamic analysis for general function classes is challenging. Additionally, merging Theorems 1 and 2 may need a new problem formulation that extends the notion of single-policy concentrability so that it depends on both the initial model and data coverage. We think these two challenges are nontrivial and an important direction for future work, perhaps drawing connections to recent results in analyzing policy gradient methods.

---

> ### Author Response · Authors · 2024-11-24
> **Response (2/2)**
>
> > Compared to Zhan et al., 2023a, what is the advantage of your method? I think the settings are pretty similar and the concentrability coefficient is defined in the same way (please correct me if I missed any subtitles). Since you claim to address both reward hacking issues by proving a finite-sample bound, as in Theorem 1, do Zhan et al. (2023a) address them as well?
>
> We indeed use the concentrability coefficient definition by Zhan et al. 2023a. However, our method and guarantees have important differences that we highlight below.
>
> **1. Our algorithm is highly practical with strong empirical performance whereas the algorithm by Zhan et al. 2023a algorithm is computationally intractable.** Zhan et al. 2023a presents an algorithm that, although comes with sample complexity guarantees against type I reward hacking, involves computationally intractable components and cannot be implemented. In contrast, our approach POWER comes with sample complexity guarantees and is highly practical and achieves strong empirical performance. One key element is the weighted entropy which allows for simplifying the maximin objective into a single-step optimization problem.
>
> **2. Our method and analysis considers access to an initial model while Zhan et al. 2023a does not. Our approach mitigates type II reward hacking while their approach suffers from type II reward hacking.** The work of Zhan et al. 2023a focuses on purely learning a policy from the preference dataset and does not consider access to an initial model. The setting considered by Zhan et al. 2023a is different from LLM practice where the RLHF step is applied to the initial model. As we explained earlier, we define type I and type II reward hacking as we are dealing with two sources of distribution shift, one between the initial model and data, and the other between the final model and data.
>
> Due to the setting considered by Zhan et al. 2023a, their guarantees only hold for type I reward hacking and since their algorithm does not incorporate the initial model, it remains susceptible to type II reward hacking. In contrast, dynamic label procedure in our work mitigates type II reward hacking. (Minor note that reference policy/distribution in Zhan et al. 2023a is not the same as initial model, e.g., they suggest setting it based on the data distribution, and is related to our baseline policy $\pi'$.)
>
> >  If I simply set $w(.)=0$, what would go wrong in Theorem 1? I believe that you have hidden some w-dependence term in the bound, and I am curious to know what that is.
>
> That is an interesting question. If one intends to solve the maximin objective in equation (6), it is indeed possible to set $w(.) = 0$. Note that Theorem 1 provides guarantee on the maximin objective (6) and does not require any additional assumption on $w$ except for $w(x)>=0$ as defined in the Weighted Entropy Definition 1. We highlight that the *dependency on $w$ indeed directly appears in the term $H_w(\pi)$ in the first inequality in Theorem 1* and the second inequality sets $w(y) = 1/|y|$.
>
> In addition to maximin objective, guarantee of Theorem 1 also extends to the POWER objective in light of Proposition 3, where we prove equivalence of the POWER objective to the maximin objective (6) under certain regularity conditions (Assumption 1) and $w(y)> 0$. Intuitively, $w(y)>0$ and $\beta>0$ introduce a strict concavity property which allow for swapping order of max and min.
>
> > Can you provide some intuitive explanations on why choosing $w(y)=1/|y|$  leads to  $\log V$ in Theorem 1?
>
> The first bound for general $w$ in Theorem 1 depends on the weighted entropy. We show that for $w(y) = 1/|y|$, the bound on weighted entropy leads to $\log V$. For simplicity, consider the case where all responses have length $Y$. Shannon entropy is maximized at uniform distribution which assigns probability of $1/|V|^Y$, reflecting the uniform probability of $1/|V|$ for selecting a token at step 1 up to step $Y$. Maximum of Shannon entropy in this case is $- \log 1/|V|^Y = Y \log V$. However, considering the weights $1/Y$, the entropy becomes $1/Y \times - \log 1/|V|^Y = \log V$.
>
> > Minor problem: I don't see how "Sail into the Headwind" is informative in the title
>
> Thank you for your feedback. We consider revising for a better title.
>
> ------
>
> Thank you again for your feedback on our work. We hope that we have answered your questions and clarified the differences with Zhan et al. 2023. We are happy to answer any more questions and discuss further in case you believe there are any missing details.
>
> **References.**
>
> Sergey Levine, Aviral Kumar, George Tucker, and Justin Fu. Offline reinforcement learning: Tutorial, review, and perspectives on open problems. arXiv preprint arXiv:2005.01643, 2020.

---

> > ### Comment · Reviewer_fMHu · 2024-11-25
> >
> > I appreciate your response and found it satisfactory. So I am raising my score to support acceptance.

---

> > > ### Author Response · Authors · 2024-11-28
> > > **Thank you for your response**
> > >
> > > Thank you for your response and for raising your score. We are happy that you found our response satisfactory and are truly grateful for your support in accepting our paper.

---

### Official Review · Reviewer_w7RZ · 2024-10-25

**Soundness:** 3
**Presentation:** 2
**Contribution:** 3
**Rating:** 8
**Confidence:** 2

**Summary:**

The paper introduces a new approach to address two forms of reward hacking due to low-coverage pairs. First, they propose a robust reward maximization objective, which enables a regret guarantee that all(?) previous methods lack. Second, they utilize a gradient freeze technique to prevent overreliance on low-coverage pairs.

**Strengths:**

I believe it is a strong paper with both convincing theoretical analysis and great empirical results.

**Weaknesses:**

Overall I think it's a very nice paper. But I do have some conceptual problems awaiting to be clarified by the authors:
1. I am not sure the two components of your method are defending against each hacking type separately ( based on your paper construction, you are designing the approach this way).
It seems that dynamic labelling simply gives low trust to low-coverage regions. Then it should help both stop the less-desired outcomes from being pushed to a highly-desired position; AND stop highly-desired outcomes from being pushed to a lowly-desired position where the pair has low coverage.  Is that correct? Then should it be a technique for both hacking types?
2. It is nice that you have theoretical guarantee for POWER itself. And empirically, POWER-DL seems to work even better. But I feel the two moves could also interfere with each other due to the fact that highly-covered pairs will still influence the poorly-covered pairs and ultimately many outcomes are competing against each other --- hence even though your set the gradient to zero for poorly covered pairs, the final distribution of the actions can still be changed by other pairs. What are your thoughts on this?
3. Ps. typo on line 092: Should be POWER-DL right?

**Questions:**

1. Could you pls comment on weakness 1?
2. Could you pls comment on weakness 2?
3. What is the main reason you introduced the $\textbf{weighted}$ entropy term? Did you have other weight in mind other than the inverse response length?

---

> ### Author Response · Authors · 2024-11-24
> **Response (1/2)**
>
> Thank you for the time spent reviewing our work and your positive feedback. We have provided our response below.
>
> > I am not sure the two components of your method are defending against each hacking type separately ( based on your paper construction, you are designing the approach this way). It seems that dynamic labelling simply gives low trust to low-coverage regions. Then it should help both stop the less-desired outcomes from being pushed to a highly-desired position; AND stop highly-desired outcomes from being pushed to a lowly-desired position where the pair has low coverage. Is that correct? Then should it be a technique for both hacking types?
>
> We clarify that since POWER solves type I, backed by strong guarantees in Theorem 1, but does not mitigate type II reward hacking (e.g. Proposition 4), we design dynamic labels. We agree with you that dynamic labels do not solely mitigate type II; yet, we provide intuition on why we think both techniques are needed to mitigate reward hacking. If we only use dynamic labels: In the high-coverage regions, we converge to empirical rewards $\approx$ robust rewards due to high coverage. In the low-coverage regions, we remain close to the initial model. In regions with medium coverage, using dynamic labels alone interpolate between initial model and empirical rewards. Empirical rewards remain susceptible to statistical noise in medium coverage and thus susceptible to reward hacking. However, if we combine POWER with dynamic labels, we interpolate between the initial model and robust rewards, resulting in better mitigation of reward hacking.
>
> We verify that empirically, the combined method POWER-DL works better than POWER alone and works better then POWER-DL with $\eta = 0$. In Appendix I.3 of the revised paper, we added empirical evaluation showing that if we only use dynamic labels but remove the robustness component with $\eta=0$, test performance degrades. This suggests both components help achieving the best performance.
>
> As a side note, we highlight the difference between divergence-minimization and dynamic labels. Divergence-based methods aim at keeping the learned model close to the initial model wherever the learned model has a decent probability, regardless of data coverage. Therefore, they may be too pessimistic. However, the dynamic label procedure aims at keeping the learned model close to the initialization only in the untrustworthy, low-coverage region and allows the model to learn from the data in high coverage region.
>
> > It is nice that you have theoretical guarantee for POWER itself. And empirically, POWER-DL seems to work even better. But I feel the two moves could also interfere with each other due to the fact that highly-covered pairs will still influence the poorly-covered pairs and ultimately many outcomes are competing against each other --- hence even though your set the gradient to zero for poorly covered pairs, the final distribution of the actions can still be changed by other pairs. What are your thoughts on this?}
>
> We agree that the poorly-covered and highly-covered pairs interact in the language space; however, we think POWER and dynamic label procedure can automatically operate on the abstract representation space, i.e., they handle poor coverage and high coverage regions in the representation space rather than language space. Coverage across the representation space is impacted by the aggregated information from all relevant samples, e.g. similar to the spectrum of feature covariance matrix that reveals which directions have higher coverage. For POWER, the general function approximation guarantee shows that it should correctly compute robust rewards in the representation space.
>
> For dynamic labels, we presented our analysis focusing on low and high coverage pairs for better clarity; however, we think a similar argument can be extended to the feature space. Intuitively, the preference dataset implies a collection of preferences over features in the abstract space, e.g., if $\theta(y)$ represents response features. In this case, dynamic labels in principle can diminish gradients for untrustworthy regions in the representation space, considering the aggregate impact of preferences implied by the poorly-covered and highly-covered pairs, combined.

---

> > ### Author Response · Authors · 2024-11-24
> > **Response (2/2)**
> >
> > > Ps. typo on line 092: Should be POWER-DL right?
> >
> > Yes, thank you for pointing that out this typo. We have fixed it in the revised manuscript.
> >
> > > What is the main reason you introduced the weighted entropy term? Did you have other weight in mind other than the inverse response length?
> >
> > **Reason to introduce weighted entropy.** We believe that weighted entropy provides a flexible and general way to incorporate conceptual preferences about the responses, guard against biases, and mitigate specific manifestations of reward hacking. Such benefits are reflected in the original definition of weighted entropy by Guiaşu (1971). Our theoretical derivation in Proposition 3 provides a sound way to incorporate the weights in the preference optimization objective, through multiplying log probabilities by weights as well as adding a weight gap $w(y^+) - w(y^-)$.
> >
> > **Choices for weights.** Inverse response length is particularly appealing choice as it relates to sample complexity and is one of the common manifestations of reward hacking. Another choice for weights are preference scores from datasets (such as those included in the Helpsteer2 dataset) or those obtained from reward models. We did not include such weights for a fair comparison with the baselines since they do not readily use such information. Another possibility is a metric that evaluates undesirable characteristics of responses pertaining to reward hacking, such as repetition, hedging, etc. Lastly, inspired by the literature in contrastive learning, another interesting choice is to view the entropy weights as importance weights designed based on gradients, aiming at improving optimization and adaptively focusing on more important pairs, in the style of $\alpha$CL framework (Tian, 2022).
> >
> > ------
> >
> > Thank you again for your feedback on our work. We hope that we have answered your questions and integrated your feedback in the revised draft. We are happy to answer any more questions and discuss further in case you believe there are any missing details.
> >
> > **References**
> >
> > Tian, Y. (2022). Understanding deep contrastive learning via coordinate-wise optimization. Advances in Neural Information Processing Systems, 35, 19511-19522

---

> > > ### Comment · Reviewer_w7RZ · 2024-11-25
> > >
> > > Thanks for your response. In general, I still feel it is a strong paper. But the current presentation (theoretical parts especially) is not so clear. I will retain my score but lower my confidence ---could you please make more modifications based on the reviewers' comments.

---

> > > > ### Author Response · Authors · 2024-11-28
> > > > **Thank you for your response**
> > > >
> > > > Thank you for your response and for acknowledging the strengths of our paper. We greatly appreciate your feedback and have made revisions to the manuscript based on the reviewers' comments. For more details, please refer to our comment titled "Summary of Changes to the Revised Manuscript."

---

### Official Review · Reviewer_3mYa · 2024-11-04

**Soundness:** 3
**Presentation:** 3
**Contribution:** 3
**Rating:** 6
**Confidence:** 4

**Summary:**

This paper addresses reward hacking in offline preference optimization (PO) methods by identifying two types of statistical errors that impact reward reliability:
Type 1: Poorly represented, suboptimal choices in the dataset appear more favorable than they are due to statistical inaccuracies.
Type 2: Due to poor statistical coverage, good choices are misrepresented as less favorable than their actual value.
The authors demonstrate that many current offline PO methods are vulnerable to both forms of reward hacking, leading to suboptimal policy learning. They propose a novel offline PO method named POWER-DL to counteract these effects. This method integrates robust reward maximization with weighted entropy and dynamic preference labeling. Experimental results show that POWER-DL outperforms state-of-the-art methods on benchmark alignment tasks, such as Arena-Hard and AlpacaEval 2.0.

**Strengths:**

The paper is overall well-written. The related work is covered comprehensively, and theoretical statements are clearly articulated (although I haven’t verified the proofs in the appendix).

The theoretical analysis, particularly Theorem 1, helps guide and justify the design of POWER-DL.

The proposed method, POWER-DL, is both theoretically grounded and easy to implement. Its approach—combining robust reward maximization with dynamic label updating—has practical implications.

The authors present extensive empirical results, with broad baseline coverage, that effectively demonstrate POWER-DL's advantages in addressing reward hacking and achieving better alignment.

**Weaknesses:**

Wu et al. (2024) introduce a dynamic beta-based approach that could be a relevant baseline here, especially given its emphasis on preference data quality.

It would be helpful to see experiments on the robustness of POWER-DL to hyperparameters, particularly \beta and \eta.

(Wu et al., 2024) \beta-DPO: Direct Preference Optimization with Dynamic \beta

**Questions:**

In Proposition 4 (I), POWER-DL is shown to recover the best-in-class policy for the constructive example given in Proposition 1. Does this imply that POWER-DL fully mitigates type 1 reward hacking? Or could there still be instances where type 1 reward hacking persists?

In Table 2, POWER-DL seems to perform better in the base setup (where it uses an existing preference dataset) than in the instruct setup (where a preference dataset is constructed by sampling from the initial model). Does this suggest that POWER-DL is more effective under a distribution shift between the initial model and the model(s) used to collect preference data?

---

> ### Author Response · Authors · 2024-11-24
> **Response**
>
> Thank you for the time spent reviewing our work and your positive feedback. We have updated our manuscript to reflect your feedback and respond to your questions and comments below.
>
> > Wu et al. (2024) introduce a dynamic beta-based approach that could be a relevant baseline here, especially given its emphasis on preference data quality.
>
> Thank you for suggesting this work. We have added a comparison of our work to Wu et al. 2024  to Appendix B.2. We are currently training $\beta$DPO to add it as a baseline to our experiments and will update the paper with the results before the end of the discussion period.
>
> > In Proposition 4 (I), POWER-DL is shown to recover the best-in-class policy for the constructive example given in Proposition 1. Does this imply that POWER-DL fully mitigates type 1 reward hacking? Or could there still be instances where type 1 reward hacking persists?
>
> Guarantees in Theorem 1 show that POWER fully mitigates type I reward hacking, as long as the competing policy (such as the best in class policy) is covered in the dataset. Example in Proposition 1 satisfies assumptions of Theorem 1 such as the coverage assumption. Proposition 4 does not imply full mitigation of type I reward hacking by POWER, but servers as a sanity check and proof details provide some intuition on why POWER objective mitigates this type of reward hacking.
>
> > In Table 2, POWER-DL seems to perform better in the base setup (where it uses an existing preference dataset) than in the instruct setup (where a preference dataset is constructed by sampling from the initial model). Does this suggest that POWER-DL is more effective under a distribution shift between the initial model and the model(s) used to collect preference data?
>
> Yes, we observe the improvement of POWER-DL over other methods is more pronounced in the base setup. This is expected because as you said, the distribution shift between data and the initial model is more significant in the base setup, and this increases the risk for reward hacking---also observed in the empirical study by Rame et al. 2024. POWER-DL mitigates reward hacking and therefore, we observe more gains in the base setup compared to other methods that remain susceptible to reward hacking. This suggests that the improvements by POWER-DL are due to mitigation of reward hacking.
>
> -------
>
> Thank you again for your feedback on our work. We hope that we have integrated your feedback in the revised draft and answered your questions. We will revise the manuscript with the $\beta$DPO numbers before the end of discussion phase. We are happy to answer any more questions and discuss further in case you believe there are any missing details.
>
> **References.**
>
> Alexandre Rame, Nino Vieillard, Leonard Hussenot, Robert Dadashi, Geoffrey Cideron, Olivier
> Bachem, and Johan Ferret. WARM: On the benefits of weight averaged reward models. In
> Forty-first International Conference on Machine Learning, 2024.

---

> > ### Comment · Reviewer_3mYa · 2024-12-03
> >
> > Thank you for addressing my concerns and conducting additional experiments. Understanding the practical limitations of obtaining new results during the discussion phase, I will maintain my positive score for the paper and support acceptance.

---

> > > ### Author Response · Authors · 2024-12-03
> > > **Thank you for your response**
> > >
> > > Thank you for your response and your support for the acceptance of our paper. We are happy that we have addressed your concerns. We have just uploaded the experimental results on $\beta$DPO, in addition to the previous experiments.

---

> > ### Author Response · Authors · 2024-12-03
> > **$\beta$DPO Experimental Results**
> >
> > We are writing to present our experimental result on $\beta$DPO as baseline. In the table below, we provide benchmark results on $\beta$DPO in the Helpsteer2 base and instruct settings as well as results on DPO and our methods POWER and POWER-DL; please refer to Table 1 in our paper for comparisons with additional baselines. Our method outperforms $\beta$DPO in both base and instruct setups.
> >
> > Helpsteer2 Base:
> >
> > | Method    | AlpacaEval LC | AlpacaEval WR | Arena-Hard WR | MT-Bench |
> > |-----------|---------------|---------------|---------------|----------|
> > | DPO       | 18.52         | 14.99         | 10            | 5.6      |
> > | $\beta$DPO| 22.92         | 15.35         | 11.5          | 5.8      |
> > ||||||
> > | POWER-DL  | **31.52**     | **31.44**     | **21.5**      | **6.0**  |
> > | POWER     | 29.57         | 30            | 19            | 5.9      |
> >
> > Helpsteer2 Instruct:
> >
> > | Method    | AlpacaEval LC | AlpacaEval WR | Arena-Hard WR | MT-Bench |
> > |-----------|---------------|---------------|---------------|----------|
> > | DPO       | 40.87         | 39.05         | 29.6          | **8.2**  |
> > | $\beta$DPO| 41.52         | 36.26         | 31.9          | **8.2**  |
> > ||||||
> > | POWER-DL  | **47.16**     | **43.08**     | **34.8**      | **8.2**  |
> > | POWER     | 43.52         | 40.19         | 31.5          | **8.2**  |
> >
> > We hope that we have addressed your questions and included the experiments you requested on $\beta$DPO and robustness (Appendix I.2). Thank you again for your feedback and consideration.

---

### Official Review · Reviewer_sDff · 2024-11-04

**Soundness:** 2
**Presentation:** 3
**Contribution:** 2
**Rating:** 5
**Confidence:** 4

**Summary:**

This paper studies the preference optimization problem for LLMs. This work points out the two types of reward hacking that could lead to inaccurate learned reward models. To solve the problem, the proposed algorithm makes use of the weighted entropy for robust estimation. Also, a dynamic labeling procedure has been illustrated in order to diminish gradients for untrustworthy samples. The theoretical and empirical studies are performed to validate the conjectures and benchmark the performance of the algorithm.

**Strengths:**

- The presentation of the paper is clear, which makes it easy to read.
-  The paper investigates one of the most important problems in preference optimization, i.e., reward hacking.
-  Some theoretical studies are conducted for analyzing the popular preference optimization, under the proposed conjectures.
-  The algorithms are evaluated in some benchmarks and downstream tasks.

**Weaknesses:**

- There are many important and highly related literature that have not been reviewed in the paper. For example,
a. Chen, Zixiang, et al. "Self-Play Fine-Tuning Converts Weak Language Models to Strong Language Models." ICML 2024
b. Rame, Alexandre, et al. "WARM: On the Benefits of Weight Averaged Reward Models." ICML 2024
c. Pan, Alexander, Kush Bhatia, and Jacob Steinhardt. "The Effects of Reward Misspecification: Mapping and Mitigating Misaligned Models."  ICLR 2022
The insufficient review of the related literature on reward-hacking and model alignment limits the credibility of the two types of reward-hacking in the paper.

-  As claimed in the paper, the type 2 reward-hacking is due to the decent choices appearing less desirable. It is reasonable to think about that if the over-pessimism in solving the type 1 reward-hacking is the root cause of not well-leveraging the decent samples. However, in this work, there is not a sufficient discussion on this point. See, Chen, Lichang, et al. "ODIN: Disentangled Reward Mitigates Hacking in RLHF." ICML 2024

- The examples used in propositions 1 and 2 are very simple, which deviated from the real applications in the fine-tuning of the LLMs, e.g., the definition of the reward function and policy classes in the examples. Therefore, from the results of propositions 1 and 2, it is only sufficient to obtain that the discussed popular preference optimization algorithms do not work well in such restrictive settings. However, the conclusion cannot be extended to the general settings in practice.

-  The principle of maximum entropy (principle of the weighted entropy in the paper) has been widely studied in the RL literature and this is not a new idea. In particular, the work by Zhu, Banghua, Michael Jordan, and Jiantao Jiao. "Principled reinforcement learning with human feedback from pairwise or k-wise comparisons."  ICML 2023 has theoretically studied the maximum entropy in RLHF/IRL settings.

-  As shown in Huang, Audrey, et al. "Correcting the mythos of kl-regularization: Direct alignment without overparameterization via chi-squared preference optimization." Arxiv, the KL divergence is not enough to induce pessimism in order to mitigate the reward-hacking. Following this argument, the weighted entropy may also encounter similar problems.

-  The choice of the weights in the weighted entropy needs more details to elaborate; otherwise, the length-normalized technique has been utilized in the existing RLHF algorithms.

-  Intuitively, the weighted entropy tends to yield a more uniform distributed policy. Therefore, this is a natural question about how to balance the exploration and exploitation tradeoff.

-  A more pronounced discussion on ``favoring unbiased policies'' is needed.

-  The \beta controls the strength of the explicit pessimism and \eta controls the strength of implicit pessimism. The two factors make the control over the pessimism complicated. In practice, how to balance the two types of pessimism, and in theory, how to mitigate over-pessimism?

-  In resolving the type 2 reward-hacking problem, is there any theoretical guarantee/regret analysis on the algorithm with the dynamic labeling procedure?

- There is no algorithm table for describing the details of the two proposed algorithms.

-  In experiments, it would be better to present some results of the iterative extension of the current method.

-  As the current method is much more complicated than the existing methods, e.g.,  DPO or SimPO, the latency of the algorithms should be presented. An efficient algorithm should balance performance and computational efficiency.

-  The evaluation over the MT-bench is very important. Also, it would be better to compare the models in the instruction-tuned model Mistral as in SimPO. This could better position the work in the existing literature.

**Questions:**

Please see the questions in Weakness.

---

> ### Author Response · Authors · 2024-11-24
> **Response (1/7)**
>
> Thank you for the time spent reviewing our work and your feedback on our work. We have updated our manuscript to reflect your feedback, extending our related work discussion (1.5 pages additional discussion in Appendix B) and clarifying our contributions. We have added many of the experiments that you suggested including MT-Bench evaluation as well as training and evaluation on the Mistral family. We have multi-iteration extension of our algorithm currently running and we will update the manuscript with those results before the end of discussion phase. We respond to your comments and questions below, and clarify our contributions.
>
> > There are many important and highly related literature that have not been reviewed in the paper. For example, a. Chen, Zixiang, et al. "Self-Play Fine-Tuning Converts Weak Language Models to Strong Language Models." ICML 2024 b. Rame, Alexandre, et al. "WARM: On the Benefits of Weight Averaged Reward Models." ICML 2024 c. Pan, Alexander, Kush Bhatia, and Jacob Steinhardt. "The Effects of Reward Misspecification: Mapping and Mitigating Misaligned Models." ICLR 2022 The insufficient review of the related literature on reward-hacking and model alignment limits the credibility of the two types of reward-hacking in the paper.
>
> Thank you for pointing out these papers. We have revised our paper, adding a thorough discussion and comparison with these papers and other relevant papers to Appendix B. We are happy to add more papers in case anything is missing. We present a comparison with the papers you mentioned here as well for completeness.
>
> - Chen et al. 2024 propose an iterative self-play mechanism to improve LLM performance without additional preference data, that outperforms DPO. This work focuses on improving LLMs in an online, iterative setting with LLM feedback and does not study preference optimization from offline data, reward hacking, robustness to partial data coverage, and statistical learning properties. The focus of our work is the study of reward hacking in offline preference optimization and designing empirically strong methods to mitigate reward hacking that come with strong statistical learning guarantees. The offline setting is considerably different from the online setting in Chen et al. 2024, as the offline setting is concerned with handling partial data coverage, whereas the online is concerned with effective exploration and data collection.
>
> - Rame et al. 2024 consider reward hacking from an empirical perspective, attribute reward hacking to distribution shift and human preference inconsistencies, and propose training multiple reward models and averaging them in the weight space. We study reward hacking from a statistical learning theory perspective and identify two types of reward hacking, both related to partial data coverage---we do not consider human preference inconsistencies in this work. We then propose theoretically founded methods to mitigate reward hacking. Our approach is much simpler and faster as we directly finetune the policy, and do not require training multiple reward models as in Rame et al. 2024.
>
> - Pan et al. 2024 consider reward hacking due to human misspecification of the (proxy) reward model from a purely empirical perspective. They construct synthetic examples of misspecified rewards across various environments. They study impact of model size, optimization, and training on reward hacking given human misspecification. In contrast, we conduct a study of reward hacking in preference optimization that arise due to statistical fluctuations (and not human misspecification) that stem from partial data coverage. We conduct theoretical analysis, showing failure of existing methods, and design robust algorithms with theoretical guarantees. We also conduct experiments using real datasets---as opposed to synthetic designs of reward hacking in Pan et al. 2024---in LLMs showing our robust methods lead to significant improvement in both alignment benchmarks and maintaining performance on academic benchmarks such as mathematical reasoning.

---

> ### Author Response · Authors · 2024-11-24
> **Response (2/7)**
>
> > As claimed in the paper, the type 2 reward-hacking is due to the decent choices appearing less desirable. It is reasonable to think about that if the over-pessimism in solving the type 1 reward-hacking is the root cause of not well-leveraging the decent samples. However, in this work, there is not a sufficient discussion on this point. See, Chen, Lichang, et al. "ODIN: Disentangled Reward Mitigates Hacking in RLHF." ICML 2024''
>
> **The root cause of type II reward hacking.** We clarify that the root cause of Type II reward hacking is statistical fluctuations in the low coverage region, which can cause poorly covered, decent choices to appear less desirable, leading to degradation of initial model. While it is possible that if an algorithm for mitigating Type I is overly pessimistic it can exacerbate Type II reward hacking, importantly over-pessimism is not the root cause of Type II reward hacking. For example, Proposition 2 shows that SimPO, which does not incorporate any form of pessimism, still suffers from Type II reward hacking. Intuitively, even if we do not use any pessimism, learned reward of poorly covered choices might still *underestimate* the true rewards by mere chance and due to high statistical fluctuations in the low coverage region.
>
> POWER is a pessimistic method and the level of pessimism can be adjusted through $\eta$. POWER-DL incorporates dynamic labels to keep the learned model close to the initial model in untrustworthy regions, and it enables a tradeoff between robust rewards and remaining close to the model in untrustworthy regions. This can lead to a better balance and less overall pessimism, as supported by our empirical evaluations in Section 6. We also added hyperparameter robustness evaluations to Appendix I.3 that shows both components achieve the best performance.
>
> **Our method avoids over-pessimism of divergence-based methods.** We highlight a benefit of our approach over divergence-minimization in terms of avoiding over-pessimism. Divergence-based methods aim at keeping the learned model close to the initial model everywhere that the learned model has a decent probability, regardless of data coverage. Therefore, they may be too pessimistic by keeping the model close to initialization even in places with high data coverage. However, our dynamic label procedure aims at keeping the learned model close to initialization only in the untrustworthy, low-coverage region and allows the model to learn from the data in high coverage region.

---

> ### Author Response · Authors · 2024-11-24
> **Response (3/7)**
>
> > The examples used in propositions 1 and 2 are very simple, which deviated from the real applications in the fine-tuning of the LLMs, e.g., the definition of the reward function and policy classes in the examples. Therefore, from the results of propositions 1 and 2, it is only sufficient to obtain that the discussed popular preference optimization algorithms do not work well in such restrictive settings. However, the conclusion cannot be extended to the general settings in practice.
>
> **1. Our examples are reasonable special case of practical setting and closer to practice than prior theoretical works.** The examples we construct in Propositions 1 and 2 are natural and reasonable, special cases of the general setting used in practice. In particular, we analyze the continuous softmax policy class applied to tabular features, and assume that the rewards are bounded. In contrast, previous work by Huang et al. 2024 shows insufficiency of KL divergence and failure of DPO by constructing a reward function class that is a discrete set and applying updates to the model despite receiving no samples. As a result, their argument breaks for continuous function classes or gradient-based optimization.
>
> **2. We analyze simpler examples to obtain a better understanding and intuition of design flaws in prior methods, which turns out to be that they overfit poorly-covered samples, leading to reward hacking.** This motivate the design of our methods that are both theoretically sound and empirically strong. Furthermore, we prove algorithms such as DPO, IPO, and SimPO suffer from reward hacking in the basic softmax policy with bounded rewards, thus, they *provably cannot achieve theoretical guarantees* like Theorem 1 achieved by our approach under general function approximation.
>
> **3. Conclusion can be extended to more general theoretical settings and is supported by experiments.** Theoretically, the conclusion that these methods suffer from reward hacking can be extended in a straightforward manner to more general setting such as linear parameterization and considering the spectrum of feature covariance matrix. Furthermore, our experiments corroborate this finding that methods like DPO and SimPO suffer from reward hacking, as evidenced by degradation of performance on downstream tasks like mathematical reasoning, which is a manifestation of reward hacking (Xu et al. 2024). Empirical study by Rafailov et al. 2024 also shows DPO suffers from reward hacking.
>
> **4. Analyzing simple examples of this kind are standard practice in statistical learning theory, and take the necessary first step to understand the more general setting.** It is a challenging open problem to analyze the non-convex optimization problem of the transformer model. Therefore, we take the first steps toward understanding the algorithms in special case of softmax policy. This is a standard practice on theory of machine learning/RL and statistical learning theory; see e.g., Jin et al. 2021; Zhu et al. 2023; Lee et al. 2016.
>
> > The principle of maximum entropy (principle of the weighted entropy in the paper) has been widely studied in the RL literature and this is not a new idea. In particular, the work by Zhu, Banghua, Michael Jordan, and Jiantao Jiao. "Principled reinforcement learning with human feedback from pairwise or k-wise comparisons." ICML 2023 has theoretically studied the maximum entropy in RLHF/IRL settings.
>
> We clarify that Zhu et. al. 2023 *does not* study maximum entropy (in a similar sense to us) in RLHF. We have confirmed with the authors Zhu et al. 2023 that they show MaxEnt IRL is similar to the maximum likelihood under Bradley-Terry/Plackett-Luce models, i.e. the standard reward objective $L_{BT}(r)$. Intuitively, the maximum entropy in MaxEnt IRL leads to a softmax similar to BT/PT models. Page 16 of their paper: “the algorithm of max entropy IRL also reduces to the MLE”.
>
> Our work and our use of entropy are completely different from Zhu et al. 2023 and MaxEnt IRL. First, MaxEnt IRL *does not* have the policy entropy that we have in equation (5). Second, Zhu et al. 2023 propose an RLHF algorithm that leverages uncertainty estimation in linear models. Our approach uses weighted entropy, robust objective, and dynamic labels, and achieves guarantees under general function classes and is completely different from both MaxEnt IRL and Zhu et al. 2023.
>
> Additionally, we do not claim that maximum entropy RL is a new idea and we cite several sources that maximum entropy in RL is well-established and widely used. We simply build our approach, that includes robust formulation and dynamic labels, upon its extension, maximum weighted entropy reward maximization. Also, in the context of RLHF, KL minimization is widely used with the goal of mitigating reward hacking. We prove that algorithms with KL minimization remain susceptible to reward hacking so our proposal is to use weighted entropy instead, which also helps with underoptimization.

---

> ### Author Response · Authors · 2024-11-24
> **Response (4/7)**
>
> > As shown in Huang, Audrey, et al. "Correcting the mythos of kl-regularization: Direct alignment without overparameterization via chi-squared preference optimization." Arxiv, the KL divergence is not enough to induce pessimism in order to mitigate the reward-hacking. Following this argument, the weighted entropy may also encounter similar problems.
>
> Theorem 1 guarantees that our robust maximin objective (6) mitigates reward hacking for general function classes and it provably cannot fail in the example constructed in Huang et al. 2024. For our practical algorithm, we make regularity assumptions on the function class (Assumption 1 e.g. satisfied for Lipschitz continuous classes), allowing us to convert it to a minimax objective leading to the simple objective of POWER (Proposition 3). Huang et al. 2024 provides a construction that relies on (1) a function class that is a discrete set rather than a continuous function, (2) updating parameters despite receiving samples. Hence, the failure argument in Huang et al. 2024 breaks in continuous function classes or gradient-based optimization. In summary, our maximin objective provably cannot fail and our simplified POWER objective provably cannot fail for function classes that satisfy regularity conditions.
>
>  On the other hand, we proved (Propositions 1 and 2) that $\chi$PO (Huang et al. 2024) suffers from reward hacking in a reasonable setting with a continuous function class, softmax policy, and bounded rewards. The reason that theoretical guarantees of $\chi$PO do not extend is that it assumes data distribution on responses is equal to the initial model. This assumption is not satisfied in neither of the two common RLHF settings in practice: (1) using previously-collected data from other models, (2) collecting multiple responses from the model and using rankings to keep a subset of them.
>
> > The choice of the weights in the weighted entropy needs more details to elaborate; otherwise, the length-normalized technique has been utilized in the existing RLHF algorithms.
>
> **1. Our choice of weight in rooted in the sample complexity analysis of Theorem 1.** We explained our choice of weight in the paragraph "Benefits of weighted entropy and choice of weights". Our theory shows that this choice of weights lead to removing a linear dependency to the response length in the sample complexity, which is a new theoretical justification for length-normalization.
>
> **2. Our Proposition (3) offers a theoretically-sound way of incorporating weights such as length-normalization in preference optimization, which is different from previous methods.** Existing methods such as SimPO and RRHF apply length-normalization in the following way: $g(1/|y^+| \log \pi(y^+|x) - 1/|y^-| \log \pi(y^-|x)) + \eta \log \pi(y^+|x)$. However, our derivation suggest a theoretically-sound way of adding weights such as length-normalization, leads to $g(w(y^+) \log \pi(y^+|x) - w(y^-) \log \pi(y^-|x) + w(y^+) - w(y^-)) + \eta w(y^+) \log \pi(y^+|x)$. This leads to two differences to prior work: a weight coefficient in the SFT term and a weight gap in the PO objective.
>
> **3. Other choices for weights.** The general weight-based objective in Proposition 3 provides a flexible framework for exploring different choices of weight that reflect preferences and other properties of responses. Here, we present other several alternatives for weights. One choice for weights are preference scores from datasets (such as those included in the Helpsteer2 dataset) or those obtained from the reward models. We did not include such weights for a fair comparison with the baselines since they do not readily use such information. Another possibility is a metric that evaluates undesirable characteristics of responses pertaining to reward hacking, such as repetition, hedging, etc. Lastly, inspired by the literature in contrastive learning, another interesting choice views the entropy weights as importance weights designed based on gradients, which aim at improving optimization and training and allow for adaptively focusing on more important preference pairs in the style of $\alpha$CL framework (Tian, 2022).

---

> ### Author Response · Authors · 2024-11-24
> **Response (5/7)**
>
> > Intuitively, the weighted entropy tends to yield a more uniform distributed policy. Therefore, this is a natural question about how to balance the exploration and exploitation tradeoff.
>
> Indeed using a weighted entropy indeed encourages a more stochastic distribution. An potential benefit of our weighted entropy approach is mitigating the notorious *overconfidence* challenge, in which RLHF-finetuned models are overconfident and have sharpened output probability (Leng et al. 2024, Kadavath et al. 2022). It would be interesting to study exploration/exploitation trade-off of our weighted-entropy approach in the online setting. However, the focus of our paper is on offline setting and this question is nontrivial and out of scope of this paper.
>
> > A more pronounced discussion on ``favoring unbiased policies'' is needed.
>
> By favoring unbiased policies, we are referring to the main premise of The Principle of Maximum Entropy, which asserts that one should ``select the distribution the leaves the largest remaining uncertainty (i.e., the maximum entropy) consistent with your constraints. That way you have not introduced any additional assumptions or biases into your calculations.'' Source: https://mtlsites.mit.edu/Courses/6.050/2003/notes/chapter10.pdf ; also see Guiasu and Shenitzer 1985.
>
> > The $\beta$ controls the strength of the explicit pessimism and $\eta$ controls the strength of implicit pessimism. The two factors make the control over the pessimism complicated. In practice, how to balance the two types of pessimism, and in theory, how to mitigate over-pessimism?
>
> **Summary of hyperparameters.** We first summarize our hyperparameters with theoretical values and experimental values to mitigate over-pessimism.
>
>
> | Parameter | Description                                                                 | Theoretical Value                  | Range in Experiments          |
> |-----------|-----------------------------------------------------------------------------|------------------------------------|------------------------------------|
> | $\beta$   | Coefficient of weighted entropy                                             | $\beta \asymp 1/\sqrt{N}$          | $\beta \in \{2, \dots, 10\}$       |
> | $\eta$    | Coefficient of weighted SFT term                | $\eta \asymp 1/\sqrt{N}$           | $\eta \in \{0.0005, 0.001\}$       |
> | $\gamma$  | Dynamic labels update rate                                                  | $\gamma \asymp 1/N^c$              | $\gamma \in \{0.1, \dots, 0.3\}$   |
>
> **Clarification on the role of hyperparameters.** We clarify that in our objective, $\beta$ is the weighted entropy coefficient that mainly impact the level of policy stochasticity and optimization landscape. This is different from $\beta$ in DPO that is the KL coefficient and impacts pessimism. In our algorithms, parameters against reward hacking are $\eta$ and $\gamma$, where $\eta$ leads to robust rewards and $\gamma$ keeps the learned model close to the initial model in untrustworthy regions. These two parameters allow preventing over-pessimism better than divergence-based methods. First, our framework allows to turn pessimism off completely by $\eta = \gamma = 0$, while it's not possible to set $\beta = 0$ in DPO. Second, $\eta$ and $\gamma$ provide *data-dependent* pessimism, leading to pessimism in regions with low data coverage, whereas divergence-based methods remain close to initial model everywhere that final model has a decent likelihood. Finally, two parameters $\eta$ and $\gamma$ allow us to better control type I and type II reward hacking, leading to a better final performance as suggested by our experiments. Intuitively, these parameters can reflect our beliefs about the quality of initial model relative to statistical errors in the data. For example, a larger $\gamma$ suggests more trust in the initial model, and a larger $\eta$ suggests noisier data.
>
> **How to select pessimism parameters and mitigate over-pessimism in theory.** Our Theorems 1 and 2 set the hyperparameters to mitigate over-pessimism as achieve optimal statistical rate. Theorem 1 sets $\eta \asymp 1/\sqrt{N}$ by optimizing inequality (40), avoiding over-pessimism in the worst case. Theorem 1 also finds $\beta \asymp 1/\sqrt{N}$ and in light of Theorem 2, $\gamma$ can be set to $1/N^c$.
>
> **How to balance the two types of pessimism in practice.** In practice, we treat these parameters as hyperparameters and select the ones that perform best in ranking implicit rewards on validation set. We also tested the robustness of our approach to hyperparameters and added the results to Appendix I.3, showing the performance of our approach is robust to hyperparameters. We note that several other RLHF objectives also have multiple hyperparameters. For example, DPO+SFT has two pessimism parameters $\beta$ is the KL coefficient and $\eta$ is the SFT coefficient, which are selected in a similar manner.

---

> > ### Author Response · Authors · 2024-11-24
> > **Response (6/7)**
> >
> > >In resolving the type 2 reward-hacking problem, is there any theoretical guarantee/regret analysis on the algorithm with the dynamic labeling procedure?
> >
> > Our Theorem 2 provides theoretical guarantee for Type II reward hacking in preference optimization with dynamic labeling procedure, showing that preferences remain close to initialization in the low-coverage regions and converge to empirical preferences to high coverage region. We note that in Theorem 2, we analyze convergence properties of a system of coupled differential equations that reflect the learning dynamics of preference optimization combined with dynamic labels. The theoretical guarantees for POWER in Theorem 1 are stronger and hold for general function approximation, and it is difficult to combine Theorem 1 and Theorem 2 as analyzing the learning dynamics for general function approximation is challenging and requires future research.
> >
> > >There is no algorithm table for describing the details of the two proposed algorithms.
> >
> > Thank you for your feedback. We added a pseudocode for our algorithm to Appendix G.
> >
> > > In experiments, it would be better to present some results of the iterative extension of the current method.
> >
> > We are currently training models and plan to post the multi-iteration results of our method before the end of discussion period. We clarify the following:
> >
> > **1. Our instruct setting is close to the iterative setting.** We already have included results in the instruct setting, which is close to the iterative setting. The instruct setting is exactly the same as SimPO, which as Meng et al. 2024 explain closely follows the iterative DPO paper Tran et al. 2024 by sampling preference data from the initial model and ranking it with a reward model. As Meng et al. 2024 explain: "This makes our Instruct setup closer to an on-policy setting" and the only difference is that ``for simplicity, we generate data in a single iteration instead of three iterations''.
> >
> > **2. The focus of our work is offline setting.** Our theoretical developments focus on the offline setting and tackling partial data coverage (as stated in the abstract). Proper extension of our algorithm (as well as other preference optimization methods) to the iterative setting is non-trivial as one needs to design an effective exploration strategy. In general, algorithms developed for offline and online RL are different as the former requires pessimism for handling partial data coverage while the latter requires optimism to encourage exploration.
> >
> > > As the current method is much more complicated than the existing methods, e.g., DPO or SimPO, the latency of the algorithms should be presented. An efficient algorithm should balance performance and computational efficiency.
> >
> > **Our approach requires only changing two lines of code compared to DPO.** We clarify that, while we present extensive theoretical analysis, our algorithm is very easy to implement and only requires changing two lines of code in DPO/SimPO. This includes (1) modifying the objective (line 4 in Algorithm 1 in the revised manuscript), and (2) adding one line for updating the dynamic labels (line 6 in Algorithm 1 in the revised manuscript), which is easily computed using the log probabilities.
> >
> > **Computational overhead of our approach is less than 10 minutes.** In implementation, our approach results in very little change in computation. For example, our approach increases computation by less than 10 minutes when training on 8xA100 GPUs---as a reference, DPO training takes from 3 to 8 hours depending on the size of dataset.

---

> > > ### Author Response · Authors · 2024-11-24
> > > **Response (7/7)**
> > >
> > > > The evaluation over the MT-bench is very important. Also, it would be better to compare the models in the instruction-tuned model Mistral as in SimPO. This could better position the work in the existing literature
> > >
> > > Thank you for your feedback; we have listed the new experiments added to the revised paper below. We clarify that the experiments in our initial submission were on both base and instruction-tuned Llama3 models (also used in SimPO). We also have experimented over two pipelines (Helpsteer2 and Zephyr).
> > >
> > > **We added experimental results on the Mistral family in all the four settings.** We added new experiments on the Mistral base and instruction-tuned models, comparing our approach with five most relevant baselines SimPO, DPO, DPO+SFT, cDPO, and $\chi$PO, over all four settings of Helpsteer2 and Zephyr as well as instruct and base settings. Our conclusion remains the same for Mistral family, and our approach consistently outperforms other methods.
> > >
> > > **We added MT-Bench evaluation results on all settings to the revised paper.** We have revised the paper and added new Table in Appendix I.2 and J evaluating the trained Llama and Mistral models on MT-Bench, showing that POWER-DL outperforms other methods. We have a similar observation to Meng et al. 2024 that MT-Bench does not separate the algorithms substantially and the scores are rather close.
> > >
> > > **While we have uploaded and are running most of the requested experimental results, given limited time and resources we are unable to include all the baselines before the end of the discussion phase as all the new experiments need $>75,000$ GPU hours.** As a rough estimate, the RLHF step alone takes at least: 15 methods $\times$ 8 runs for hyperparameter search $\times$ number of settings (4 Mistral + 3 iterative settings $\times$ 3 iterations = 13) $\times$ 8 GPUs $\times$ 6 hours (training) $\approx 75,000$ hours.
> > >
> > > ------
> > >
> > > Thank you again for your feedback on our work. We hope that we have integrated your feedback in the revised draft, addressed your concerns, and answered your questions. We have included your requested experiments on Mistral and MT-Bench and will revise the manuscript with the multi-iteration experiments before the end of discussion phase. We are happy to answer any more questions and discuss further in case you believe there are any missing details.
> > >
> > > **References.**
> > >
> > > Xu, T., Helenowski, E., Sankararaman, K. A., Jin, D., Peng, K., Han, E., ... and Fang, H. (2024). The perfect blend: Redefining RLHF with mixture of judges. arXiv preprint arXiv:2409.20370.
> > >
> > > Rafailov, R., Chittepu, Y., Park, R., Sikchi, H., Hejna, J., Knox, B., ... and Niekum, S. (2024). Scaling laws for reward model overoptimization in direct alignment algorithms. arXiv preprint arXiv:2406.02900.
> > >
> > > Hoang Tran, Chris Glaze, and Braden Hancock. Iterative DPO alignment. Technical report, Snorkel AI, 2023.
> > >
> > > Lee, J. D., Simchowitz, M., Jordan, M. I., and Recht, B. (2016, June). Gradient descent only converges to minimizers. In Conference on learning theory (pp. 1246-1257). PMLR.
> > >
> > > Zhu, Banghua, Michael I. Jordan, and Jiantao Jiao. "Principled reinforcement learning with human feedback from pairwise or K-wise comparisons." Proceedings of the 40th International Conference on Machine Learning. 2023.
> > >
> > > Jin, Ying, Zhuoran Yang, and Zhaoran Wang. "Is pessimism provably efficient for offline rl?." International Conference on Machine Learning. PMLR, 2021.
> > >
> > > Leng, J., Huang, C., Zhu, B., and Huang, J. (2024). Taming Overconfidence in LLMs: Reward Calibration in RLHF. arXiv preprint arXiv:2410.09724
> > >
> > > Kadavath, S., Conerly, T., Askell, A., Henighan, T., Drain, D., Perez, E., ... and Kaplan, J. (2022). Language models (mostly) know what they know. arXiv preprint arXiv:2207.05221.

---

> > > > ### Comment · Reviewer_sDff · 2024-11-27
> > > > **Thank you for the rebuttal!**
> > > >
> > > > I am grateful for the detailed rebuttal and the additional experiments conducted to address the comments. The work has become more comprehensive during the rebuttal phase, and I encourage the authors to ensure that the discussions and new results are incorporated into the revised manuscript.

---

> > > > > ### Author Response · Authors · 2024-11-28
> > > > > **Thank you for your response**
> > > > >
> > > > > Thank you for your response and encouraging comments. We have revised the manuscript, incorporating the discussions and new experimental results---pending the multi-iteration experiment, which we plan to post before the end of discussion period. Please refer to our comment titled "Summary of changes to the revised manuscript" for more details.
> > > > >
> > > > > If you feel we have adequately addressed your concerns, we greatly appreciate it if you could consider adjusting your score accordingly. In our revisions and rebuttal, we highlighted the benefits of our work compared to related work, clarified aspects of our formulation and algorithm---such as the ease of implementation, controlling pessimism, and computational efficiency---, and revised our paper based on the discussions. Additionally, we have included new experiments on MT-Bench, Mistral family, and hyperparameter robustness, all of which provide further support for the benefits of our approach. We are happy to discuss further and answer any remaining questions. Thank you for your consideration.

---

> > > > > > ### Author Response · Authors · 2024-12-03
> > > > > > **Additional Experimental Results**
> > > > > >
> > > > > > We are writing to present additional experimental results.
> > > > > >
> > > > > > **Multi-Iteration Experiments**
> > > > > >
> > > > > > The multi-iteration experiments are conducted using the Zephyr pipeline over three iterations, and we compare our approach with iterative DPO (Tran et al. 2024; Dong et al. 2024), XPO (Xie et al. 2024), and an iterative variant of SimPO.
> > > > > >
> > > > > > | Method          | AlpacaEval LC  | AlpacaEval WR  | Arena-Hard WR  | MT-Bench  |
> > > > > > |-----------------|-----------|-----------|----------|----------|
> > > > > > | DPO iter = 1    | 40.02     | 38.58     | 31.6     | 7        |
> > > > > > | DPO iter = 2    | 42.15     | **40.71**     | 37.8     | 7.2      |
> > > > > > | DPO iter = 3    | 42.38     | 40.8      | 38.8     | 7.2      |
> > > > > > | | | | | |
> > > > > > | XPO iter = 1    | 40.96     | **39.5**      | 32.6     | 6.9      |
> > > > > > | XPO iter = 2    | 41.33     | 39.66     | 36.3     | **7.3**      |
> > > > > > | XPO iter = 3    | 42.42     | 39.98     | 35.7     | **7.3**      |
> > > > > > | | | | | |
> > > > > > | SimPO iter = 1  | 38.91     | 36.44     | 28.7     | **7.2**      |
> > > > > > | SimPO iter = 2  | 41.15     | 37.77     | 28.9     | 7.2      |
> > > > > > | SimPO iter = 3  | 41.97     | 38.32     | 32.6     | 7.2      |
> > > > > > | | | | | |
> > > > > > | POWER-DL iter = 1  | **41.42**    | 37.64     | **34.4**     | **7.2**      |
> > > > > > | POWER-DL iter = 2  | **43.48**  | 40.17     | **39.8**     | 7.2      |
> > > > > > | POWER-DL iter = 3  | **46.55**    | **43.21**     | **42.1**     | **7.3**      |
> > > > > >
> > > > > >
> > > > > > These results demonstrate the benefits of our approach that extend to the multi-iteration setting.
> > > > > >
> > > > > > **Performance on MT-Bench**
> > > > > >
> > > > > > We realized that we forgot to add our MT-Bench Llama experimental results table in the revised paper (though MT-Bench results on the Mistral family are included). The table below provides these results. The highest score among the trained models is shown in **bold** and the second highest score is *italic*. POWER-DL consistently outperforms other methods. Similar to Meng et al. (2024), we observe that variations between different methods in MT-Bench scores are minor compared to those in AlpacaEval and Arena-Hard.
> > > > > >
> > > > > > | Method       | Llama3-8B-Base (Helpsteer2) | Llama3-8B-Instruct (Helpsteer2) | Llama3-8B-Base (Zephyr) | Llama3-8B-Instruct (Zephyr) |
> > > > > > |--------------|-----------------------------|--------------------------------|-------------------------|-----------------------------|
> > > > > > | Initial Model| 4.9                         | 8.3                            | 6.1                     | 8.3                         |
> > > > > > ||||||
> > > > > > | DPO          | 5.6                         | **8.2**                        | *7.0*                   | **8.3**                     |
> > > > > > | DPO+SFT      | 5.5                         | _8.1_                          | _7.0_                   | 8.0                         |
> > > > > > | cDPO         | 5.8                         | _8.1_                          | 6.9                     | _8.2_                       |
> > > > > > | R-DPO        | 5.4                         | **8.2**                        | 6.9                     | 8.1                         |
> > > > > > | IPO          | 4.9                         | 8.0                            | 6.3                     | 8.0                         |
> > > > > > | $\chi$PO          | 5.5                         | 8.0                            | 6.9                     | 8.1                         |
> > > > > > | SPPO         | 5.8                         | **8.2**                        | **7.1**                 | _8.2_                       |
> > > > > > | CPO          | _5.9_                       | 7.2                            | 6.3                     | 8.0                         |
> > > > > > | RRHF         | 5.7                         | 7.9                            | 6.2                     | 8.1                         |
> > > > > > | SLiC-HF      | _5.9_                       | 7.8                            | 6.8                     | 7.9                         |
> > > > > > | ORPO         | 5.7                         | _8.1_                          | 6.6                     | **8.3**                     |
> > > > > > | SimPO        | **6.0**                     | 8.0                            | 6.8                     | _8.2_                       |
> > > > > > | ROPO         | 5.8                         | _8.1_                          | **7.1**                 | _8.2_                       |
> > > > > > ||||||
> > > > > > | POWER-DL     | **6.0**                     | **8.2**                        | **7.1**                 | _8.2_                       |
> > > > > > | POWER        | _5.9_                       | **8.2**                        | _7.0_                   | _8.2_                       |
> > > > > >
> > > > > >
> > > > > > Thank you again for your feedback and consideration.

---

### Author Response · Authors · 2024-11-28
**Summary of changes to the revised manuscript**

Dear reviewers,

Thank you for responding to our rebuttal. We have uploaded a new revision of our paper incorporating the discussions and comments from the reviewers. Below is a summary of the modifications compared to the original submission:

**1. Clarifications and further discussions on our formulation and approach:** (based on comments from **all reviewers**)
- **The two types of reward hacking:** We elaborated on our motivation in the definition of the two types of reward hacking, their relationship with each other and with the sources of distribution shift in the offline RLHF setting. We also compared types of reward hacking with partial data coverage and pessimism in offline RL. We made modifications throughout the paper (e.g., Introduction, Sections 3-5) and included further discussion in Appendix B.3.
- **Our algorithm and its components:** We added a discussion on the two components of our algorithm, explaining how they allow to adjust and trade off between the two types of reward hacking, and how they can effectively mitigate over-pessimism compared to previous methods. We added Section 5, paragraph "POWER with Dynamic Labels", Appendix G (POWER-DL pseudocode), a discussion on Theorems 1 and 2 and their unification in Appendix B.3, along with modifications throughout the paper.
- **Benefits of weighted entropy, length normalization, and choice of weights:** We added a discussion to Appendix B.5, highlighting the benefits of weighted entropy, including sample complexity, theoretically-sound objective, and practical advantages such as disentangling stochasticity and pessimism, and potentially alleviating overconfidence. We further discussed a number of other choices for weights. We also expanded Section 4.3 "Benefits of weighted entropy and choice of weights".

**2. Extensive review of and comparison with related work (Appendix B.1-B.6):** We extended our discussion of related work, providing a comprehensive review and comparison that includes RLHF, the origins of reward hacking, approaches for mitigating reward hacking, specific manifestations of reward hacking, and the theory of RLHF. We also compared our work with specific studies mentioned by reviewers **sDff**, **3mYa**, and **fMHu**.

**3. New experiments**
- **MT-Bench evaluations on the all models:** for Llama and Mistral family and all the four settings (Reviewer **sDff**).
- **Results on the Mistral Family in all the four settings:** (Reviewer **sDff**).
- **Hyperparameter robustness results:** (Reviewer **3mYa**).

We are grateful for the reviewers' valuable feedback that has improved our work. We hope that we have integrated the discussions and comments from the reviewers in the revised manuscript.

---

### Meta-Review · Area_Chair_pRPx · 2024-12-20

**Metareview:**

Summary
This work investigates the conditions under which reward hacking issues arise and proposes methods to mitigate them. A key contribution is the introduction of two novel concepts: I-type reward hacking, which focuses on suboptimal data coverage, and II-type reward hacking, which addresses suboptimal initial models. To address these issues, the authors propose two new algorithms with theoretical guarantees. Experimental results demonstrate their effectiveness.

Strengths
- The presentation is clear and well-structured.
- The introduced concepts are novel, to the best of my knowledge.
- The experimental results are comprehensive and robust.

Weaknesses
- The discussion linking the newly introduced reward hacking concepts to existing literature, such as concentrability, could be more thorough and detailed.

Decision
Accept.

**Additional Comments On Reviewer Discussion:**

During the discussion phase, the authors provided additional clarifications, incorporated more related works, and conducted further experiments. These updates effectively addressed the reviewers' comments.

---

### Decision · Program_Chairs · 2025-01-22

Accept (Poster)